# Bacteria invade the brain following intracortical microelectrode implantation, inducing gut-brain axis disruption and contributing to reduced microelectrode performance

Brain-machine interface performance can be affected by neuroinflammatory responses due to blood-brain barrier (BBB) damage following intracortical microelectrode implantation. Recent findings suggest that certain gut bacterial constituents might enter the brain through damaged BBB. Therefore, we hypothesized that damage to the BBB caused by microelectrode implantation could facilitate microbiome entry into the brain. In our study, we found bacterial sequences, including gut-related ones, in the brains of mice with implanted microelectrodes. These sequences changed over time. Mice treated with antibiotics showed a reduced presence of these bacteria and had a different inflammatory response, which temporarily improved microelectrode recording performance. However, long-term antibiotic use worsened performance and disrupted neurodegenerative pathways. Many bacterial sequences found were not present in the gut or in unimplanted brains. Together, the current study established a paradigm-shifting mechanism that may contribute to chronic intracortical microelectrode recording performance and affect overall brain health following intracortical microelectrode implantation.

Intracortical microelectrodes hold promise for studying brain functions and treating neurological disorders by recording neural signals from the brain[1,2]. However, translation of this technology to clinical applications requires long-term reliability of the microelectrodes[3]. The neuroinflammatory response in the brain following implantation has been identified as a major factor influencing microelectrode performance[3–5]. Despite extensive studies on the identification of triggers of neuroinflammation and their related pathways following microelectrode implantation[4,6–10], limited information exists on neuroinflammatory responses to microelectrodes associated with the presence of bacteria at the site of the implant[11].

Bacteria can enter the brain at various stages of device implantation during the surgical procedure, ranging from contamination of the initially sterile device to transport by blood to the implantation site[12,13]. We have previously demonstrated that bacterial contamination from the implant itself can be avoided with rigorous sterilization and proper surgical technique[11], and decades of similarly rigorous sterilization and proper surgical techniques in human participants have reported no adverse related bacterial contamination events[14]. Degradation of blood-brain barrier (BBB) integrity is an appreciable consequence of microelectrode-mediated neuroinflammation and increases the entry of blood-borne components into the brain[6,15,16].

✉e-mail: lxz716@case.edu; jrc35@case.edu

Therefore, it is not inconceivable that other factors circulating in the blood at the time of, or after, microelectrode implantation could also enter the brain through the permeable domain. The integrity of the mucosal lining of the intestines is dysfunctional following traumatic brain injury (TBI)[17]. Upon microelectrode implantation, there is compression of brain tissue of 1–3 mm, which Rennakar et al. suggest shares similar injury characteristics with TBI due to the compression of brain tissue upon insertion[18]. Therefore, a potential pathway for gut-derived bacteria to enter the brain following the trauma associated with microelectrode implantation may exist. Bacteria could enter from the bloodstream through the damaged BBB. The investigation of microbes in diseased and injured brains has recently become an exciting area of research[19,20]. The existence of a brain-specific micro-biome is beyond the scope of this study. However, publicly presented data, which have not appeared in a peer-reviewed publication, suggested that certain gut bacterial constituents could penetrate the BBB, become resident in brain parenchyma in rodents and humans, and play a role in health and disease[20]. In fact, there is robust evidence that gut microbiota can trigger and mediate systemic neuroinflammatory processes that have been implicated in neuropsychiatric conditions such as schizophrenia, depression, anxiety, Alzheimer's, Parkinson's, and stroke[19,21]. Despite such links, there are no reports of how brain responses to gut microbiome infiltration following microelectrode implantation might be explored to improve device tissue integration and performance.

The gut microbiome affects innate and adaptive immunological players, ranging from epithelial cells and antigen-presenting cells to innate lymphoid cells and regulatory T-cells[22,23]. Diverse microbiota-derived bioactive molecules, including bacteria-produced metabolites and even neurotransmitters, have been strongly implicated in inflammatory processes in the gut and the brain[24]. However, the role of gut-resident microorganisms translocated beyond the gut's usual niche remains unclear. There are multiple elements and processes through which the gut microbiome affects brain health that constitute the microbiome-gut-brain axis in both acute and chronic brain disease[25].

Our study explores the role of the microbiome-gut-brain axis and neuroinflammatory response following intracortical microelectrode implantation in a mouse model. The current study was designed to test the hypothesis that microelectrode implantation could disrupt the microbiome-gut-brain axis via changes to the composition of the gut microbiome and/or infiltration of the brain by gut-resident microbes. Utilizing 16S rRNA gene sequencing, we have identified transient populations of bacterial sequences which were previously observed in the gut but not in background samples from the naïve brain and bacterial sequences which were of an undefined anatomical origin observed in neither the gut nor background following microelectrode implantation. We also demonstrated that systemic antibiotic treatment altered bacterial feature abundance and composition in feces, and the composition of sequences from the implanted brain tissue of antibiotic-treated mice was more like background than in the control cohort. Manipulations of the gut microbiome with antibiotic treatment were associated with changes in single-unit recordings using intracortical microelectrodes and temporal changes in the neuroinflammatory response as indicated through spatial proteomics and spatial transcriptomics.

Although the composition of bacterial sequences varied by timepoint and treatment group relative to background in DNA extracted from implanted brain tissue, it is important to indicate that these results do not confirm the presence of live bacteria in the brain. Our initial findings suggest that future studies could explore the connection between the endogenous (example: gut) microbiome and microelectrode performance by modulating the (gut) microbiome or implementing strategies to manipulate the neuroinflammatory response to invading bacteria. While the source of bacterial sequences not resident in the gut or seen at background was not identified in this study, future studies could also investigate sources such as the nasal cavity or oral microbiome, and/or that some portion of the 16S DNA is brought into the brain by infiltrating macrophages that had phagocytosed the bacteria before entering the brain during the inflammatory response to the implanted microelectrodes.

## Results

### Bacteria invade the brain after intracortical microelectrode implantation

Microbiome composition can be profiled by sequencing the 16S rRNA gene from bacterial DNA isolated from feces or other tissue[26]. Here, the V3-V4 region of the gene for the 16S rRNA small subunit was sequenced using total DNA extracted from brain biopsy punches or fecal samples. Measurements were taken from pre-treatment baseline and weekly fecal samples from unimplanted, untreated control mice two weeks after housing separation ($n = 5$). Additionally, samples were collected and measured for intracortical microelectrode-implanted, untreated control mice from the acute ($n = 6$) and chronic time points ($n = 7$), 4- and 12-weeks post-implantation, respectively. We compared within-sample diversity, between-sample diversity, and differential abundance in the implanted brain tissues to samples from the naïve, intact brain as a background condition to characterize any changes in sequence composition following BBB disruption.

To determine the impact, if any, of depleting the fecal (gut) microbiota on bacterial sequences extracted from the brain, an additional cohort of mice were treated with antibiotics: unimplanted ($n = 5$), acute ($n = 5$), and chronic ($n = 6$). Antibiotic-treated mice were provided with an antibiotic cocktail of Ampicillin, Clindamycin, and Streptomycin in their drinking water following established protocols, and antibiotic-treated mice displayed significant alterations to the gut microbiome as early as one week after the start of treatment, which continued throughout the study (Supplementary Fig. S1).

Given the ongoing concern that bacterial sequences recovered from tissue are artifactual (e.g., introduced via sample preparation), an effort was made to computationally evaluate the presence of host-derived bacterial DNA in the brain tissue. 16S sequencing of total DNA from tissue samples is subject to two primary sources of contaminating reads: host organism gDNA and reagent or laboratory artifacts (e.g., the 'kitome')[27]. The contribution of host gDNA and 'kitome' contamination are assumed to be equivalent across samples processed in parallel according to the same protocol, whereas the amount of microbial DNA extracted is expected to vary sample-to-sample (Fig. 1A). As the amount of microbial DNA derived from a sample increases, the proportion of total reads derived from contaminants decreases (i.e., the relative abundance of contaminants is inversely related to read count). This statistical property can help identify and remove contaminants from analyses[28].

Sequences that aligned to the NCBI mouse genome were categorized as contaminating host gDNA. Sequences prevalent in the no-template sequencing blanks or whose abundance was negatively correlated to read count in the fecal samples were categorized as contaminating microbial DNA[28]. The remainder were classified as putative 16S amplicons. The Spearman correlation between the sample's read count and the abundance of the two types of contamination in the sample was compared between the brain and fecal tissues, as a reference (Fig. 1B).

In brain tissues and fecal samples, there is a significant inverse relationship between the relative abundance of contaminant DNA and a sample's read count, which aligns with the statistical assumption and validates the exclusion of these sequences from further analysis. Host gDNA was the primary contaminant in brain tissue samples, while technical contaminants were more abundant in the fecal samples. The abundance of sequences categorized as contaminating microbial DNA was negatively correlated with total read count in fecal samples, but it was not correlated with total read count in brain tissues. Extraction

reagent blanks were not used, so there may have been an incomplete removal of technical contaminants from the lowest read count samples. In contrast, the relative abundance of the remaining sequences increased with increasing sample read count in both brain tissues and

fecal samples, suggesting that these remaining reads are not contaminants and describe host-derived bacterial DNA.

Having established that microbial DNA extracted from brain tissues was not entirely due to contamination, we sought a relative

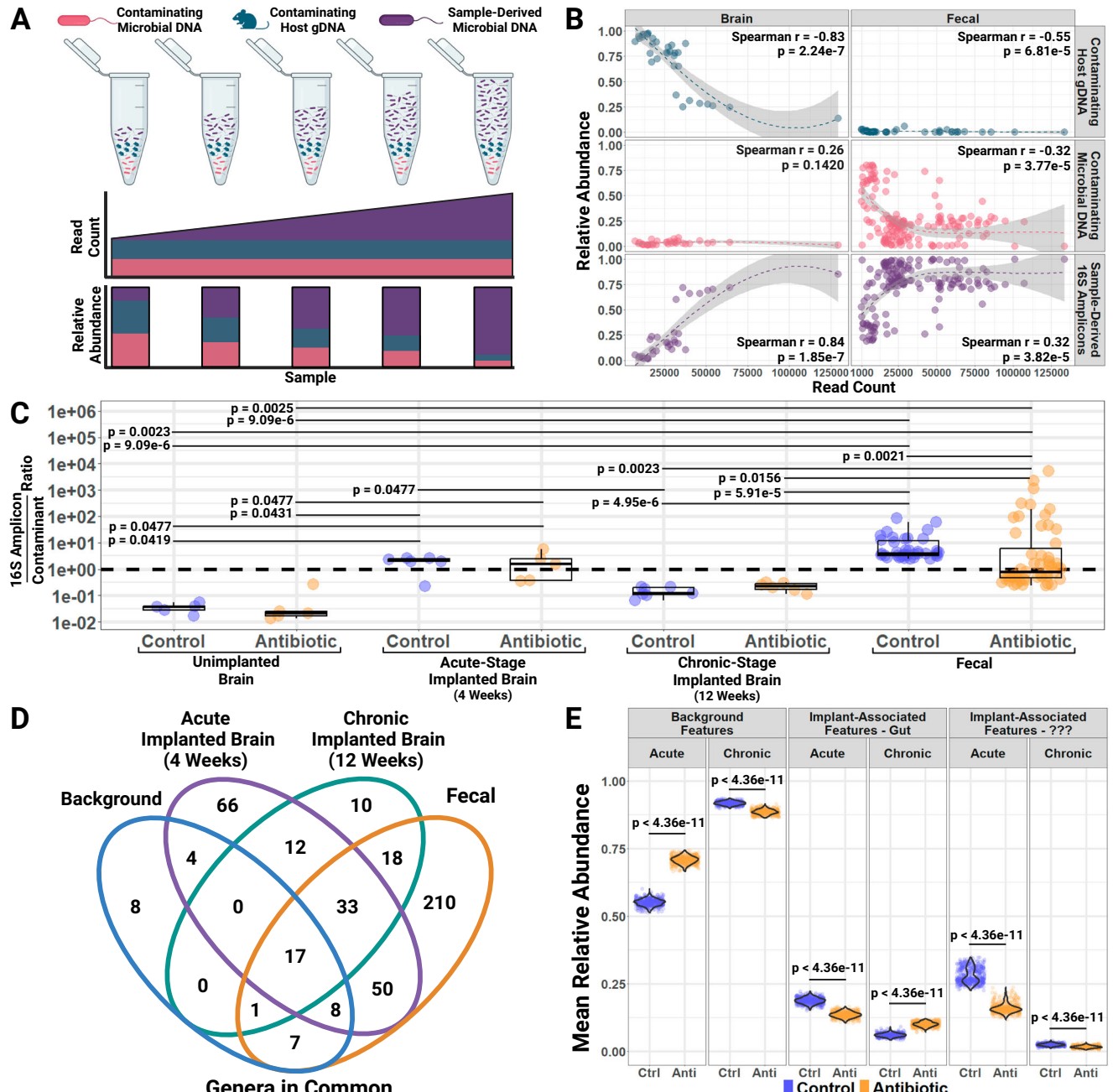

**Fig. 1 | Systemic antibiotic treatment associated with fewer distinct bacterial features in brain following intracortical microelectrode implantation.** Biological replicate sample sizes: unimplanted control brain ($n = 5$), unimplanted antibiotic brain ($n = 5$), acute implanted control brain ($n = 6$), acute implanted antibiotic brain ($n = 5$), chronic implanted control brain ($n = 7$), chronic implanted antibiotic brain ($n = 6$), baseline fecal ($n = 38$), antibiotic fecal ($n = 55$). **A** Schematic of relationship between technical contaminants and sample-derived bacterial sequences in microbiome studies. **B** Scatterplots, Locally Estimated Scatterplot Smoothing (LOESS) regressions, and Spearman correlations of total read count versus relative abundance of contaminating host gDNA, contaminating microbial DNA, and sample-derived 16S amplicons for brain tissues and fecal samples. LOESS curve shading indicates 95% confidence for local weighted least-squares fit, and Spearman correlation $p$-values are 2-sided. **C** Boxplots with raw data points for the ratio

of 16S amplicon reads to contaminant reads for untreated control and antibiotic-treated brain tissues versus untreated baseline and antibiotic-treated fecal samples. Whisker length is 1.5x the interquartile region, lower box boundary the 1st quartile, center line the median, and upper box boundary the 3rd quartile. Pairwise comparisons were made using a 2-sided Dunn test and Benjamini-Hochberg correction for multiple comparisons. **D** Total genera observed in background, acute and chronic brains, or fecal samples. **E** Violin plots over raw data points of mean relative abundance in rarified implanted brain samples of bacterial features shared by background samples or which were implantation-associated, subdivided by whether sequences were also observed in the fecal samples or of unknown anatomical origin. 500 rarefactions performed and pairwise comparisons made using 2-sided Tukey's Honest Significant Differences with adjustment for multiple comparisons. Created in BioRender. Capadona, J. (2025) https://BioRender.com/g57h879.

quantification of host-derived microbial DNA, recognizing that 16S sequencing cannot directly quantify the microbial load of a sample without additional experimental controls. To assess this, we calculated the ratio of 16S amplicon reads to contaminant reads by sample and compared the ratios observed in the brain to those of fecal samples expected to have high microbial biomass (i.e., pre-treatment fecal samples) and low microbial biomass (i.e., antibiotic-treated fecal samples) (Fig. 1C). Samples with similar microbial biomasses were expected to have similar ratios of 16S amplicon reads to contaminant reads, and ratios for samples with low or no microbial biomass were expected to be less than one and significantly different from the ratios for the baseline or antibiotic-treated fecal samples.

First, the unimplanted, untreated control brains were compared to control, implanted brains at the acute and chronic time points to determine if the ratio of 16S reads to contaminant reads varied by implantation status. Then, the implanted control acute and chronic brains were compared to antibiotic-treated brains at the corresponding time points to determine if the ratios observed in the untreated control animals differed following antibiotic-treatment (Fig. 1C). The ratio of 16S amplicon reads to contaminant reads in the untreated control brains was significantly greater in the acute than in the unimplanted or chronic untreated control brains, with the latter two not significantly different. Assuming that technical contamination is proportional between samples, the increase and subsequent decrease in the ratio of 16S reads to contaminant reads suggested a transient rise in host-derived bacterial DNA after implantation.

In the control untreated unimplanted naïve brain tissue, the median ratio of 16S reads to contaminant reads was 0.04, such that 16S reads were 26.39 times less abundant than contaminant reads. This suggests very few bacteria were in the unimplanted brain, with the bulk of reads coming from contaminants. Conversely, 16S reads were 2.10 times more abundant than contaminant reads in the control acute-stage implanted brain tissues, suggesting a large rise in host-derived bacterial DNA compared to the background levels in unimplanted brain samples. In the control chronic-stage implanted brain, 16S reads were 8.20 times less abundant than contaminant reads (median ratio: 0.12), similar to the unimplanted brain, with the bulk of reads coming from contaminants. The antibiotic-treated group was not significantly different from the untreated controls for the unimplanted or acute-stage. However, the median ratio was slightly elevated in the antibiotic-treated chronic-stage brain tissues versus the control group (0.23 vs. 0.12), which indicated the antibiotic treatment did not meaningfully alter the proportion of host-derived bacterial DNA in these samples relative to controls. These ratios suggest that microbial biomass is low or non-existent in the unimplanted brain and is considered background for the sake of this analysis. Still, after implantation, bacterial DNA levels rise significantly by 4 weeks post-implantation before falling back down to levels like that of the unimplanted brain by 12 weeks post-implantation.

The median ratios of 16S reads to contaminant reads of brain tissue samples were also compared to those observed in the antibiotic-treated fecal samples and the corresponding baseline fecal samples as references for low and high expected microbial loads (Fig. 1B). The median ratio of 16S reads to contaminant reads in the baseline fecal samples was 3.72 (min: 2.44, max: 82.50), which as expected was greater than the median ratio of 0.76 in the antibiotic-treated fecal samples (min: 0.24, max: 5133). The ratio of 16S reads to contaminant reads was greater in both baseline and antibiotic-treated fecal samples than in unimplanted or chronic implanted brain tissues of both treatment groups. However, the fecal sample ratios were not significantly different from either the control or antibiotic-treated acute implanted brains, suggesting that sample-derived 16S amplicons make up a similar proportion of the extracted DNA in these brain tissues as they do in the fecal samples.

## Antibiotic treatment reduces abundance of distinct, implant-associated bacterial features

Having removed contaminants and finding evidence of host-derived bacterial DNA in the implanted brain samples, our first step was to identify and compare the composition of bacterial sequences present in both fecal matter and at background in the samples from unimplanted mice to those seen in samples from microelectrode-implanted mice. Given how little bacterial DNA was extracted in the naïve unimplanted brain samples, these samples from both control and antibiotic-treated mice will be referred to as background for their respective cohorts, so as not to imply that these results provide evidence of a native brain microbiome. Analyses were conducted on read counts for both unique amplicon sequence variants (ASVs) and operational taxonomic units (OTUs) of similar sequences.

Variation in microbial communities manifest primarily via differences in prevalence (presence/absence of an OTU in a sample) and/or differences in abundance (proportion of sample reads derived from an OTU). Significant differences in the prevalence and the abundance of bacterial sequences were observed in the brain following microelectrode implantation. Observations at the genus level will also be discussed as changes to the genera. Although we recognize that differences in the observed features between implanted and background could be attributed to their being sequenced separately, for convenience's sake in this analysis, any ASV or OTU that was not originally detected in background samples was considered a distinct feature associated with implantation. At the same time, any taxa observed in the unimplanted brain samples were referred to as common background features.

Across all samples within a group, background samples contained 45 total genera (8 unique), whereas the 4-week brains contained 190 total genera (66 unique, 161 invading, 29 non-invasive) and the 12-week brains contained 91 total genera (10 unique, 73 invading, 18 non-invasive). There was a total of 189 distinct, implantation-associated genera observed across the acute and chronic brains (Fig. 1D). By 12 weeks post-implantation, only 73 of these 189 distinct genera could still be detected in the brain samples. There were 17 common background features also found at both time points post-implantation, and one background genus was observed at the chronic time point after becoming absent during the acute time point. Of the 189 distinct, implantation-associated genera, 101 were also observed in the fecal samples, lending credibility to the hypothesis that gut-derived bacteria invade the brain after IME implantation.

To investigate the connection between the gut microbiome and these distinct, implantation-associated features following IME implantation, the genera observed in the implanted brain were classified as common features, distinct features observed in the gut, and distinct features with unknown anatomical origin (not observed at background or in the gut). The relative abundance of bacterial sequences from these three categories was compared between the implanted brains of both groups after repeatedly rarifying ($n = 500$) to adjust for variation in sequencing depth (Fig. 1E)[29]. Looking at control brain tissues, 18.8% ± 0.01% of 16S reads at 4 weeks post-implantation and 6.1% ± 0.01% at 12 weeks post-implantation were distinct, implantation-associated features also found in the gut. 28.1% ± 0.3% of 16S reads at 4 weeks post-implantation and 2.5% ± 0.01% at 12 weeks post-implantation was from distinct, implantation-associated features of unknown anatomical origin. All implanted brains were processed and sequenced in parallel, so the variation in the abundance of these distinct, implantation-associated features between the acute and chronic time points cannot be explained by a batch effect.

At both the acute and chronic time points, the abundance of both types of distinct, implantation-associated features were also significantly different in antibiotic-treated animals versus untreated controls. Distinct implantation-associated features found in the gut were 27.8% less abundant (95% CI: 26.8–28.8%) in antibiotic-treated

brain tissues than controls at 4 weeks post-implantation but 65.9% more abundant (95% CI: 63.5–68.3%) in antibiotic-treated brain tissues than controls at 12 weeks post-implantation. Distinct, implantation-associated features of unknown anatomical origin was 43.0% less abundant (95% CI: 41.3–44.6%) in antibiotic-treated brain tissues than controls at 4 weeks post-implantation and 33.0% less abundant (95% CI: 31.0–35.0%) in antibiotic-treated brain tissues than controls at 12 weeks post-implantation.

Although antibiotic-treated brains were processed and sequenced in parallel with the corresponding control cohort, this study did not experimentally validate the origin of these bacterial sequences. Therefore, we cannot rule out that these differences are not due to differential contamination of the brain sample with host-derived bacterial DNA from other body sites, rather than indicative of the presence of live bacteria in the brain. If implantation disrupts the microbiome at other body sites, the contamination of samples from implanted brains would differ from the contamination in background samples. Similarly, systemic antibiotic treatment may have altered the microbiome in those mice, potentially contributing distinct contaminants. Any such differences in contamination would likely be reflected in the results, underscoring the need for further experimental validation to clarify the true origin of the bacterial sequences.

## Antibiotic treatment mitigates diversity of implantation-associated sequences

To better understand what types of bacteria the distinct, implantation-associated features are, we utilized Linear discriminant analysis Effect Size (LEfSe), a method used in biomarker discovery to identify taxa most likely to be associated with differences between experimental groups[30]. Higher values of the log-linear discriminant analysis score indicate greater enrichment of taxa within a particular group (Supplementary Fig. S2A). The cladogram displays the taxonomic structure and phylogenetic overlap of the differential features across implantation status, highlighting the distinction between 4 weeks and 12 weeks post-implantation (Supplementary Fig. S2B). The background samples were characterized by sequences from the phyla Bacteroidota, specifically the genus Muribaculaceae, and Firmicutes, specifically the genus Lactobacillus of the class Bacilli. The 4-week post-implantation brain was characterized by sequences from the phylum Firmicutes, specifically of the class Clostridia, and a shift to the genus Streptococcus over Lactobacillus in the class Bacilli. The 12-week post-implantation brain was characterized by sequences from the phyla Bacteroidota, as in the background samples, but specifically the genus Bacteroides over Muribaculaceae. The two phyla identified by LEfSe as important for discriminating between the experimental groups, Bacteroidota and Firmicutes, were the two most abundant phyla in the brain samples across all implant statuses and both treatment groups (Fig. 2A).

The Shannon Diversity Index, a measure of alpha (within a sample) diversity, provides a quantitative assessment of the feature richness and evenness of a bacterial community sample; it is robust to sample composition, with a higher value indicating greater sample diversity[31]. The Shannon Diversity Index varied significantly by implantation status, with the highest values of 3.55 ± 0.07 observed in the 4-week post-implantation control brains as compared to 1.16 ± 0.09 in the 12-week post-implantation control brains and 2.31 ± 0.05 in the background samples (Fig. 2B). The number of observed ASVs varied significantly within treatment group with the largest number of ASVs observed at 4 weeks, in which 72 ± 3 ASVs were observed in rarefied control brains (Fig. 2C). At 12 weeks post-implantation, control brains contained 17 ± 2 ASVs compared to 20 ± 1 ASVs in background samples (Fig. 2C).

Notably, both alpha diversity metrics were significantly different between antibiotic-treated and control animals for not only implanted animals but unimplanted as well. The mean Shannon Diversity Index in the antibiotic-treated group versus the control group was 0.707 points lower for the unimplanted brains (95% CI: 0.699–0.715), 0.820 points lower for the 4 week post-implantation brains (95% CI: 0.813–0.828), and 0.252 points higher for the 12 week post-implantation brains (95% CI: 0.244–0.260, Fig. 2B). On average, there were 7.62 fewer observed ASVs in the unimplanted brains (95% CI: 7.42–7.81), 19.53 fewer observed ASVs in the 4 week post-implantation brains (95% CI: 19.34–19.73), and 5.14 more observed ASVs in the 12 week post-implantation brains (95% CI: 4.94–5.34) between antibiotic-treated samples and control samples (Fig. 2C).

All antibiotic-treated samples were processed in parallel and sequenced with the corresponding untreated controls, so the variation in alpha diversity between the two treatment groups cannot be explained by batch effects. In addition, were these 16S amplicons extracted from whole brain tissues and analyzed here solely artifactual (e.g., residual contaminants from DNA extraction), we would not expect them to vary in response to antibiotic-treatment of the animal. The variation seen here supports the plausibility that we were successful in retrieving sample-derived microbial sequences from whole brain tissues, and the composition of these sample-derived microbial sequences varies in response to systemic antibiotic treatment. However, we again cannot rule out contamination of the brain biopsy samples by bacterial DNA from other body sites, so these results could represent disruptions to the microbiome which do not occur in the brain.

Given the complexity of comparing multiple groups with feature-rich microbial data, we applied the dimension reduction technique of principal coordinates analysis (PCoA) to visualize the relationship between background samples and implanted brain samples (4-weeks and 12-weeks post-implantation) in the two treatment groups (control vs. antibiotic-treated) in 2-dimensional space (Fig. 2D, E). The unweighted UniFrac distance measures Beta (between-groups) diversity by considering the phylogenetic information of observed microbes[32]. We used the UniFrac distance to quantify the degree of genetic difference between samples by calculating the fraction of branch length in the de novo-assembled phylogenetic tree that is unique to either of the two samples being compared and PERMANOVA to test for differences by implantation status. Samples that are more like each other have fewer unique evolutionary relationships and appear closer together in the 2-dimensional ordination space, as will experimental groups.

In the untreated control cohort, we found three significantly distinct clusters, segregated by implantation status, suggesting that the collection of bacterial sequences seen in 4-week post-implanted and 12-week post-implanted animals are genetically distinct from the background samples (Fig. 2D). There was meaningful overlap between the acute and chronic samples, suggesting that there is variation in the rate at which the bacterial environment may stabilize following implantation. Of note, there was little overlap between the implanted samples and background samples, which indicated either a batch effect from sequencing implanted samples separately or that the environment had still not returned to a baseline state after 12 weeks.

However, in the antibiotic-treated cohort, the unimplanted, acute, and chronic brain tissues did not form distinct clusters in any of the rarefied data sets (Fig. 2E). The considerable overlap between the three groups, despite the background samples being sequenced separately, suggests that the implanted brains from antibiotic-treated mice are more like background antibiotic-treated samples than the implanted brains from the control cohort are to each other or untreated background samples. While this result may be due to depletion of other body site microbiomes, resulting in less host-derived sample contamination as compared to controls, it also indicates that systemic antibiotic treatment may mitigate the appearance of distinct, implantation-associated features and promote a return to or maintenance of the baseline, unimplanted brain environment in response to IME.

## Antibiotic treatment impacts intracortical microelectrode performance

Functional, single-shank, silicon 16-channel intracortical microelectrodes were implanted into the primary motor cortex to obtain awake neural recordings. Animals were separated into two cohorts consisting of untreated control and antibiotic-treated groups. Biweekly recordings and analysis indicate that arrays implanted in antibiotic-treated animals performed significantly better than the control group based

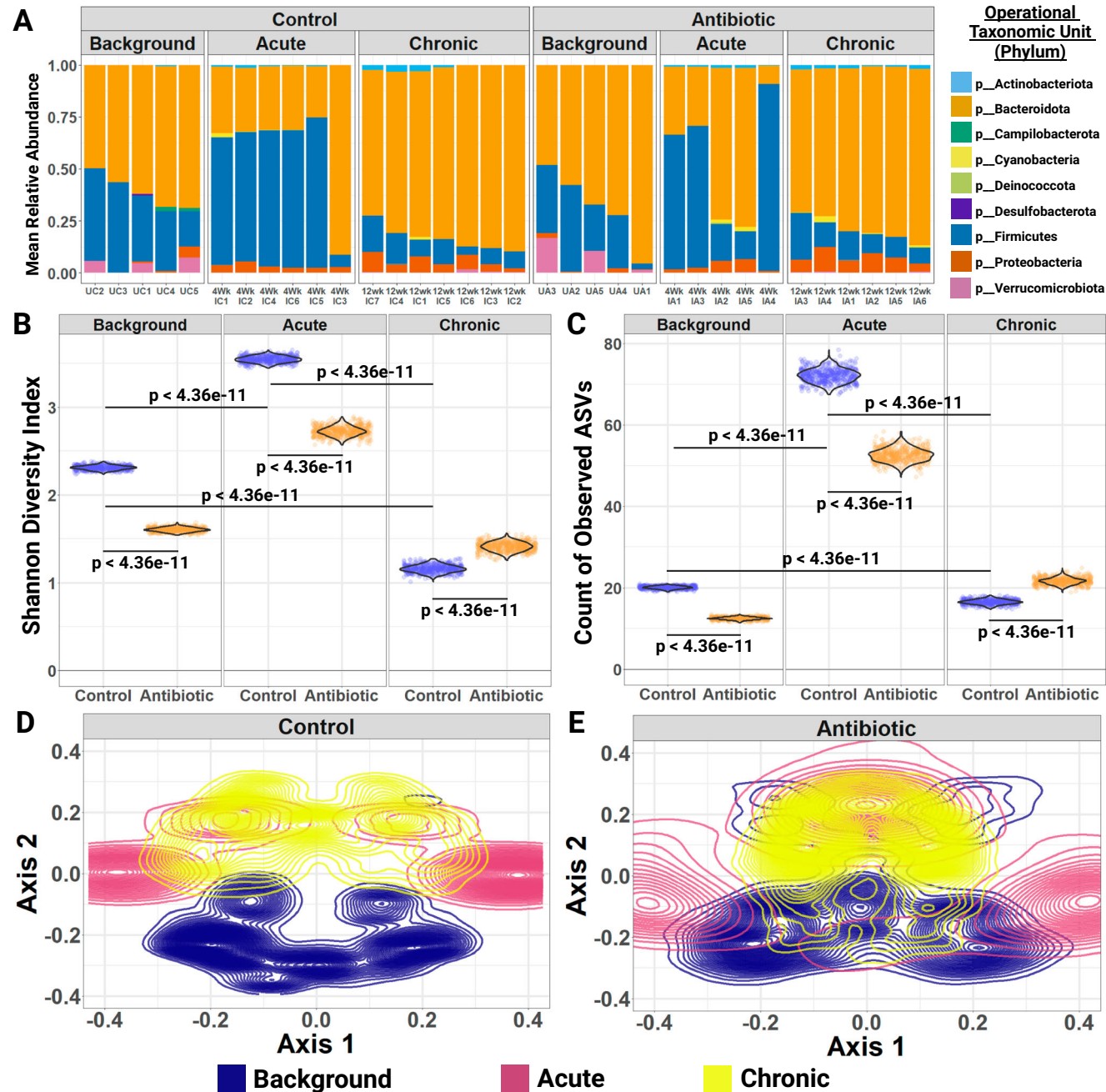

**Fig. 2 | Intracortical microelectrode implantation affects the composition of bacterial sequences extracted from brain tissues, and compositional shifts are reduced in antibiotic-treated animals.** Short-term and long-term changes in 16S sequence prevalence and abundance were observed following IME implantation, which were mitigated by systemic antibiotic treatment. Biological replicate sample sizes: unimplanted control brain ($n = 5$), unimplanted antibiotic brain ($n = 5$), acute implanted control brain ($n = 6$), acute implanted antibiotic brain ($n = 5$), chronic implanted control brain ($n = 7$), chronic implanted antibiotic brain ($n = 6$). **A** Bar plots of the mean relative abundance of bacterial sequences by sample and phylum from 500 rarefactions. **B** Violin plots over raw data points of the mean Shannon Diversity Index in rarefied samples. 500 rarefactions performed and pairwise comparisons made using 2-sided Tukey's Honest Significant Differences with multiple comparisons adjustment. **C** Violin plots over raw data points of the mean number of observed ASVs in rarefied samples. 500 rarefactions performed and pairwise comparisons made using 2-sided Tukey's Honest Significant Differences with multiple comparisons adjustment. **D** Density contour plot of implanted brain and background samples of the untreated control cohort ordinated by Principal Components Analysis (PCoA) on rarefied samples using the unweighted UniFrac distance with 2-sided PERMANOVA test with the Benjamini-Hochberg correction for multiple testing. 10,000 permutations used in the analysis of 500 rarefactions, with the largest adjusted PERMANOVA $p$-value being $p = 0.0025$. **E** Density contour plot of implanted brain and background samples of the antibiotic-treated cohort ordinated by Principal Components Analysis (PCoA) on rarefied samples using the unweighted UniFrac distance with PERMANOVA test. 10,000 permutations used in the analysis of 500 rarefactions, with the smallest unadjusted PERMANOVA $p$-value being $p = 0.2668$. Created in BioRender. Capadona, J. (2025) https://BioRender.com/c43l764.

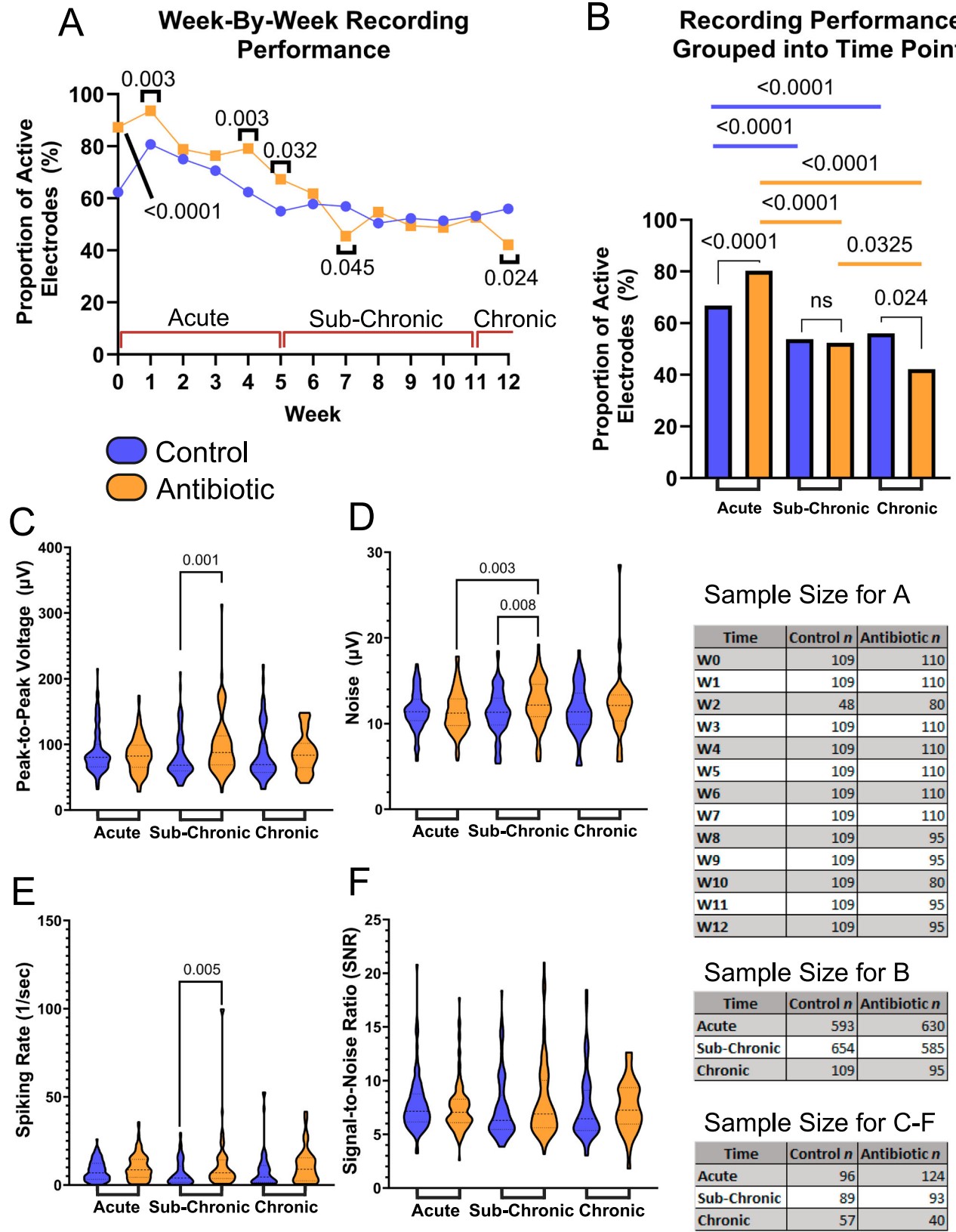

**Sample Size for A**

| Time | Control n | Antibiotic n |
|---|---|---|
| W0 | 109 | 110 |
| W1 | 109 | 110 |
| W2 | 48 | 80 |
| W3 | 109 | 110 |
| W4 | 109 | 110 |
| W5 | 109 | 110 |
| W6 | 109 | 110 |
| W7 | 109 | 110 |
| W8 | 109 | 95 |
| W9 | 109 | 95 |
| W10 | 109 | 80 |
| W11 | 109 | 95 |
| W12 | 109 | 95 |

**Sample Size for B**

| Time | Control n | Antibiotic n |
|---|---|---|
| Acute | 593 | 630 |
| Sub-Chronic | 654 | 585 |
| Chronic | 109 | 95 |

**Sample Size for C-F**

| Time | Control n | Antibiotic n |
|---|---|---|
| Acute | 96 | 124 |
| Sub-Chronic | 89 | 93 |
| Chronic | 57 | 40 |

on measurement of the proportion of active electrodes, or active electrode yield (AEY), at week 0 (day of implantation, $n = 109$ for control, $n = 110$ for antibiotic), week 1 ($n = 109$ for control, $n = 110$ for antibiotic), week 4 ($n = 109$ for control, $n = 110$ for antibiotic), and week 5 ($n = 109$ for control, $n = 110$ for antibiotic) (Fig. 3A). The largest difference in AEY was observed in week 4 (79% for antibiotic-treated animals vs. 62% for the control group).

Antibiotic-treated and control animals declined significantly in performance over time, consistent with historical data (Fig. 3B)[4,33,34]. When grouped into known phases for the maturation of the

**Fig. 3 | Antibiotic Treatment Significantly Improves the Recording Performance of Intracortical Microelectrodes.** Neurophysiological recordings to evaluate the performance of our intracortical microelectrodes and the impact of antibiotic treatment compared to control. Blue indicates control while orange indicates antibiotic groups. Comparisons are made to evaluate **A** the week-by-week proportion of active electrodes and **B** the acute, sub-chronic, and chronic grouped proportion of active electrodes. Additional metrics were evaluated to measure the **C** peak-to-peak voltage ($V_{pp}$), **D** root-mean-squared of the noise, **E** spike-rate of the single units, and **F** signal-to-noise ratio (SNR) of all active channels. The sample size for all comparisons is included as well. A one-tailed proportions z-test was used for

calculating statistical differences in the proportion of active electrodes within and across groups for the acute, sub-chronic, and chronic phases. Peak-to-peak voltage, noise, spiking rate, and signal-to-noise ratio were compared within and across acute, sub-chronic, and chronic neuroinflammatory phases using a Kruskal-Wallis test followed by a Benjamini–Krieger–Yekutieli test to adjust for multiple comparisons for non-normal distributions to increase statistical power and reduce type I errors. Statistically significant p-values are displayed in the figure. No symbol or the abbreviation "ns" indicates a lack of statistical significance. No comparisons were made between antibiotic and control at differing time points. Created in BioRender. Capadona, J. (2025) https://BioRender.com/p93j360.

neuroinflammatory response[33,35], antibiotic-treated mice performed significantly better in the acute (weeks 0–5, $n = 593$ for control, $n = 630$ for antibiotic) phase of implantation (80% AEY for antibiotics vs. 67% AEY for control), exhibited no difference during the sub-chronic (weeks 6–11, $n = 654$ for control, $n = 585$ for antibiotic) phase (52% for antibiotics vs 54% for control), and displayed a significant decline in performance at the chronic (week 12, $n = 109$ for control, $n = 95$ for antibiotic) time period (42% for antibiotic vs 56% for control, Fig. 3B).

During the sub-chronic implant period, the peak-to-peak voltage ($V_{pp}$) (97.9 μV ± 43.7 μV for antibiotic vs. 81.1 μV ± 34.1 μV for control, Fig. 3C), noise levels (12.4 μV ± 2.7 μV for antibiotic vs. 11.3 μV ± 2.7 μV for control, Fig. 3D), and spike rate (12.0 ± 17.8 for antibiotic vs. 6.8 ± 6.9 for control, Fig. 3E), were all significantly higher in the antibiotic group compared to the control. Although there were no significant differences in AEY at the sub-chronic time point, the larger amplitude and spiking rate may indicate healthier and more active neurons in the antibiotic group at that time point. SNR showed no significant change across groups or time points (Fig. 3F).

**Antibiotic treatment impacts the neuroinflammatory response to intracortical microelectrodes**

Neuroinflammation has long been associated with intracortical microelectrode failure[3,15,36]. Here, we utilized one of the most advanced methods reported to date to assess the intracortical microelectrode-tissue interface, both spatial proteomics (with and without cell specificity)[37], and spatially-resolved whole mouse transcriptomics[38,39]. Our goal was to begin to understand the relationship between invasive microbes in the brain, neuroinflammation, and microelectrode recording performance.

Spatial and cell-specific neural proteomic evaluation of the implant site (up to 270 μm from the implant) provides a robust view of the brain tissue's health and the treatment's effect on inflammation. Employing panels for neural health from NanoString, proteins were measured pertaining to neural cell profiling (25 proteins), glial cell subtyping for identification of specific cell types (10 proteins), and autophagy processes (10 proteins) (Table 1). Comparisons between antibiotic and control were made at 4- and 12-weeks post-implantation, along with temporal comparisons within the antibiotic and control groups (4-week antibiotic: $n = 4$, 4-week control: $n = 3$, 12-week antibiotic: $n = 3$, 12-week control: $n = 3$). Fold change, the experimental protein expression divided by the control protein expression, was calculated for each comparison. A fold change of less than 1 (negative log2(fold change) or log2FC) indicates lower expression in antibiotic-treated mice compared to the control group (downregulation). In comparison, greater than 1 (positive log2FC) indicates higher expression in antibiotic-treated mice compared to the control group (upregulation). Across all comparisons, 28 of the 39 tested proteins were significantly differentially expressed in at least one comparison. Table 1 shows the full list of 39 proteins examined and the six proteins used for quality control and normalization. The complete area of interest (AOI) represents the tissue within 270 μm of the implant site. Within the AOI, the inner AOI is the tissue adjacent to the implant site to 90 μm from the implant site; the middle AOI is the tissue 90 μm to 180 μm from the implant site; and the outer AOI is the tissue 180 μm to 270 μm from the

implant site. Neuron (NeuN-positive) and astrocyte (GFAP-positive) cell-specific regions were analyzed for each AOI. All twelve potential combinations of the AOIs used for comparison in this study are summarized in Table 2 and can be visualized in the Methods (See Methods for an in-detail explanation).

The proteomic analysis of antibiotic-treated mice compared to control at 4 weeks post-implantation indicated that in all cases of differential protein expression, the proteins were decreased in expression in tissue from the antibiotic-treated mice compared to the untreated control group (Fig. 4, Table 2). Seven of the twelve AOI comparisons indicated differential protein expression (Fig. 4). Without cell-specific segmentation of the AOI, the complete AOI (0–270 μm), inner, and outer AOI all showed differential protein expression. Specifically, protein expression of our panel showed the downregulation of 18 proteins for the full AOI (ATG12, ATG5, BAG3, CD163, CD31, CD40, CD68, CSF1R, MAP2, NeuN, NfL, OLIG2, P62, PLA2G6, SYP, TMEM119, ULK1, and VIM), six in the inner region (ATG12, CD68, MAP2, SYP, TMEM119, ULK1), none in the middle, and 19 proteins in the outer region (ATG12, CD31, CD40, CD45, CD68, CSF1R, CTSD, GPNMB, ITGAX, MAP2, NeuN, NfL, P62, PLA2G6, SYP, TMEM119, ULK1, VIM, and VPS35) (Fig. 4A–D, Table 2).

For neuron-specific comparisons, there were 15 total proteins downregulated for the full AOI (ATG12, BAG3, CD31, CD68, CSF1R, GPNMB, MAP2, NfL, OLIG2, P62, PLA2G6, SYP, TMEM119, ULK1, and VIM), none in the inner region, 18 downregulated in the middle region (ATG12, BAG3, CD11b, CD31, CD39, CD45, CD68, GPNMB, Ki-67, MAP2, OLIG2, P62, PLA2G6, SYP, TFEB, ULK1, VIM, and VPS35), and eight downregulated in the outer region (BAG3, CD31, CD68, CSF1R, GPNMB, MAP2, SYP, and ULK1) (Fig. 4E–H, Table 2).

In astrocyte-specific comparisons, no proteins were differentially expressed in the full AOI, middle, or outer regions (Fig. 4I–L). However, seven proteins were downregulated in the inner region of the astrocyte-specific AOI (ATG12, CD40, CD68, MAP2, MerTK, TFEB, and ULK1) (Fig. 4I–L, Table 2). Table 2 summarizes comparisons, including all twelve AOIs at 4-weeks post-implantation.

There are several proteomic markers for microglia activation between their pro-inflammatory M1 phenotype (CD14, CD16, CD32, CD40, CD86, MHCII) and their anti-inflammatory M2 phenotype (CD163 and CD206), a few of which were measured in this study[40]. Consistently across comparisons, the antibiotic group shows a downregulation in CD40, indicating reduced M1 microglial activity. Additionally, CD68 is a common protein marker for microglia and macrophages[41], showing consistent downregulation across the comparisons above. Notably, CD163 is also downregulated, but only in a single comparison above.

The proteomic analysis of brain tissue from antibiotic-treated mice compared to control at 12-weeks post-implantation indicated only one differentially expressed protein, CD163, which is a marker that indicates the transition from pro-inflammatory M1 to M2 anti-inflammatory macrophage phenotype (Supplementary Fig. S3)[42]. CD163 was indicated to be upregulated in the antibiotic-treated group, compared to the untreated control group in the astrocyte-specific collection, in the area between 180 and 270 μm from the microelectrode-tissue interface. This upregulation to the M2

**Table 1 | NanoString neural proteomic panel**

| Protein Symbol | Module | Function |
| --- | --- | --- |
| ATG12 | Autophagy | Autophagosome Formation |
| ATG5 | Autophagy | Autophagosome Formation |
| BAG3 | Autophagy | Autophagy Promotion |
| Beclin-1 | Autophagy | Autophagy Promotion |
| LC3B | Autophagy | Autophagosome Formation, Protein Sorting |
| P62 | Autophagy | Protein Sorting |
| PLA2G6 | Autophagy | PD |
| TFEB | Autophagy | Autophagy Promotion, Lysosomal Biogenesis |
| ULK1 | Autophagy | Autophagy Promotion |
| VPS35 | Autophagy | Protein Sorting |
| Aldh1l1 | Glial Cell Subtyping | Astrocyte |
| CD9 | Glial Cell Subtyping | Disease-Associated Microglia |
| CSF1R | Glial Cell Subtyping | Microglia |
| Ctsd | Glial Cell Subtyping | Microglia |
| GPNMB | Glial Cell Subtyping | Disease-Associated Microglia |
| ITGAX | Glial Cell Subtyping | |
| MSR1 | Glial Cell Subtyping | Microglia |
| MerTK | Glial Cell Subtyping | Disease-Associated Microglia |
| SPP1 | Glial Cell Subtyping | Disease-Associated Microglia |
| VIM | Glial Cell Subtyping | Vimentin, an Astrocyte marker |
| CD11b | Neural Cell Profiling | DC, Myeloid |
| CD163 | Neural Cell Profiling | M2 Macrophage, Macrophage, Myeloid, Myeloid Suppression |
| CD31 | Neural Cell Profiling | Endothelial |
| CD39 | Neural Cell Profiling | Myeloid Suppression |
| CD40 | Neural Cell Profiling | Myeloid, Myeloid Activation |
| CD45 | Neural Cell Profiling | Total Immune |
| CD68 | Neural Cell Profiling | M2 Macrophage, Macrophage, Myeloid |
| GFAP | Neural Cell Profiling | Astrocyte, Inflammation |
| IBA1 | Neural Cell Profiling | Microglia |
| Ki-67 | Neural Cell Profiling | Proliferation |
| MAP2 | Neural Cell Profiling | Cytoskeleton, Neuron |
| MHC II | Neural Cell Profiling | Antigen Presentation, MHC2 |
| MBP | Neural Cell Profiling | Myelin basic protein for Oligodendrocytes |
| NeuN | Neural Cell Profiling | Neuron |
| NfL | Neural Cell Profiling | Neurofilament light, a Cytoskeleton, Neuron |
| OLIG2 | Neural Cell Profiling | Oligodendrocytes |
| S100B | Neural Cell Profiling | Antigen, Astrocyte, Inflammation, Melanoma, Tumor |
| SYP | Neural Cell Profiling | Synaptophysin, a Synaptic Vesicle |
| TMEM119 | Neural Cell Profiling | Microglia |
| Rb IgG | Neural Cell Profiling | Background |
| Rt IgG2a | Neural Cell Profiling | Background |
| Rt IgG2b | Neural Cell Profiling | Background |
| GAPDH | Neural Cell Profiling | Housekeepers |
| Histone H3 | Neural Cell Profiling | Housekeepers |
| S6 | Neural Cell Profiling | Housekeepers |

The full list detailing the specific proteins involved in the NanoString mouse neural proteomics panel. The list is split into the Autophagy module, the Glial Cell Subtyping module, and the Neural Cell Profiling core panel. Proteins denoted as "Background" are negative control proteins. Proteins denoted as "Housekeepers" are housekeeping proteins. The negative control and housekeeping proteins were used for checking the success of the assay and normalization only.

phenotype may promote preservation of viable neural tissue near the implant site[43].

Within treatment groups, examination of temporal changes in protein expression from 4-weeks to 12-weeks post-implantation showed that five of the twelve AOI comparisons of antibiotic-treated mice indicated differential protein expression (Supplementary Fig. S4, Supplementary Table S1). Two of the more noteworthy comparisons were identified within examinations of the inner ring. There were eight differentially expressed proteins in the astrocyte-specific inner ring, and ten differentially expressed proteins in the non-specific inner ring, suggesting the largest differential expression between the temporal comparison of the antibiotic-treated animals to be in the tissue closest

**Table 2 | Antibiotic treatment significantly reduces neural proteomic expression at 4-Weeks post-implantation**

**4-Week Antibiotic vs Control**

| Protein Symbol | Protein Name | Category/Function | All AOI | Inner Ring | Middle Ring | Outer Ring | All NeuN | Inner NeuN | Middle NeuN | Outer NeuN | All GFAP | Inner GFAP | Middle GFAP | Outer GFAP |
|---|---|---|---|---|---|---|---|---|---|---|---|---|---|---|
| ATG12 | autophagy related 12 | Autophagy: autophagosome protein (KEGG:04140)[111–113] | - | - | | - | - | | - | - | | - | | |
| ATG5 | autophagy related 5 | Autophagy: autophagosome protein (KEGG:04140)[111–113] | - | | | | | | | | | | | |
| BAG3 | BCL2-associated athanogene 3 | Autophagy: promotes autophagy through increasing glutamine consumption[114] | - | | | | - | | - | - | | | | |
| CD11b | integrin alpha M | Microglia, macrophages, peripheral immune: expressed in activated microglia, macrophages, and dendritic cells; plays a role in cell-cell adhesion during inflammation[115,116] | - | | | | | | - | | | | | |
| CD163 | CD163 antigen | M2 Macrophage, Macrophage, Myeloid, Myeloid Suppression: Marker that indicates transition from pro-inflammatory M1 to M2 anti-inflammatory macrophage phenotype[42] | - | | | | | | | | | | | |
| CD31 | platelet/endothelial cell adhesion molecule 1 | Endothelial cell marker, mediates BBB permeability, specifically permeability to immune cells[116] | - | | | - | - | | - | - | | | | |
| CD39 | ectonucleoside triphosphate diphosphohydrolase 1 | Myeloid Suppression: Expressed by microglia and involved in both innate and adaptive immunity[117] | | | | | | | - | | | | | |
| CD40 | CD40 antigen | General marker for myeloid cells and myeloid activation. Also plays a role in neuronal development, maintenance, and protection[118,119] | - | | | - | | | | | | | | |
| CD45 | protein tyrosine phosphatase, receptor type, C | Microglia, macrophages, peripheral immune: Expressed highly in macrophages, less so in microglia, involved in adhesion and T-cell activation[120] | - | | | - | | | - | | | | | |
| CD68 | CD68 antigen | Macrophage, Microglia, Myeloid: Highly expressed in macrophages and myeloid cells. General marker for inflammation[121] | - | - | | - | - | | - | - | | - | | |
| CSF1R | colony stimulating factor 1 receptor | Microglia: A major regulator of microglial development and homeostasis[122]. | - | | | - | - | | | - | | | | |
| CTSD | Cathepsin D | Astrocytes, microglia, macrophage, peripheral immune: Regulates proteolysis in lysosomes; expressed in neurons, astrocytes, microglia, and macrophages[123–126] | | | | - | | | | | | | | |
| GPNMB | glycoprotein (transmembrane) nmb | Disease-Associated Microglia: Highly expressed in microglia and macrophages during neuroinflammation and neurodegenerative disease states[127] | - | | | - | - | | - | | | | | |
| ITGAX | Integrin alpha x | Microglia, macrophages, peripheral immune: integrin protein found in microglia, macrophages, dendritic cells, and T-cells[116], hypothesized to be neuroprotective[128,129] | | | | - | | | | | | | | |
| Ki-67 | N/A | Astrocytes, microglia, macrophages, peripheral immune: Marker of cell proliferation[116,130] | | | | | | | - | | | | | |

**Table 2 (continued) | Antibiotic treatment significantly reduces neural proteomic expression at 4-Weeks post-implantation**

**4-Week Antibiotic vs Control**

| Protein Symbol | Protein Name | Category/Function | All AOI | Inner Ring | Middle Ring | Outer Ring | All NeuN | Inner NeuN | Middle NeuN | Outer NeuN | All GFAP | Inner GFAP | Middle GFAP | Outer GFAP |
|---|---|---|---|---|---|---|---|---|---|---|---|---|---|---|
| MAP2 | microtubule-associated protein 2 | Neuronal health: Cytoskeletal/microtubule structure[31] | - | - | | - | - | | - | - | | - | | |
| MerTK | MER proto-oncogene tyrosine kinase | Astrocytes, microglia, macrophages, peripheral immune: anti-inflammatory, involved in phagocytosis of apoptotic cells; also plays a role in maintaining BBB integrity[132–134] | - | | | | | | | | | - | | |
| NeuN | RNA binding protein, fox-1 homolog (C. elegans) 3 | Neuronal health: Binds to DNA in mature neurons, serves as a marker for neuronal nuclei[135] | - | | | - | | | | | | | | |
| NfL | neurofilament, light polypeptide | Neuronal health: Cytoskeletal protein found in neurons[136] | - | | | - | - | | | | | | | |
| OLIG2 | oligodendrocyte transcription factor 2 | Neuronal health: Promotes myelin production in oligodendrocytes and differentiation[137] | - | | | | - | | - | | | | | |
| P62 | sequestosome 1 | Autophagy: Tags protein inclusion bodies and delivers them to autophagosomes for degradation[138] | - | | | - | - | | - | | | | | |
| PLA2G6 | phospholipase A2, group VI | Autophagy: Associated with Parkinson's and other neurodegenerative diseases. Essential for metabolism of phospholipids, proper mitochondrial function, inhibition of apoptosis[139,140]. | - | | | - | - | | - | | | | | |
| SYP | synaptophysin | Neuronal health: synaptic vesicle protein[141] | - | - | | - | - | | | - | | | | |
| TFEB | transcription factor EB | Autophagy: promotes transcription of genes that lead to autophagosome formation[142] | | | | | | | - | | | - | | |
| TMEM119 | transmembrane protein 119 | Microglia: Marker for microglia, specifically resting[115] | - | - | | - | - | | | | | | | |
| ULK1 | unc-51 like kinase 1 | Autophagy: forms protein complex that initiates autophagosome formation[143] | - | - | | - | - | | - | | | | | |
| VIM | vimentin | Astrocytes: Intermediate filament protein regulating astrocyte structure during gliosis, also expressed in endothelial cells[144–146] | - | | | - | - | | - | | | | | |
| VPS35 | Vacuolar protein sorting 35 | Autophagy: forms protein complex that initiates autophagosome formation[143] | | | | - | | | - | | | | | |

The list of neural proteomic panels that are differentially expressed between the 4-week antibiotic and the 4-week control groups within the entire AOI (All AOI, 270 µm from the implant site), the inner ring (Inner Ring, 90 µm from the implant site), middle ring (Middle Ring, 90–180 µm from the implant site), and outer ring (Outer Ring, 180–270 µm from the implant site), along with Neuron-specific cells (NeuN) and astrocyte-specific cells (GFAP) for each of those regions. The protein symbol, full name, and function are included in the table. Cells denoted with a "-" indicate downregulation in the 4-week antibiotic group compared to the 4-week control group, while "+" indicates upregulation.

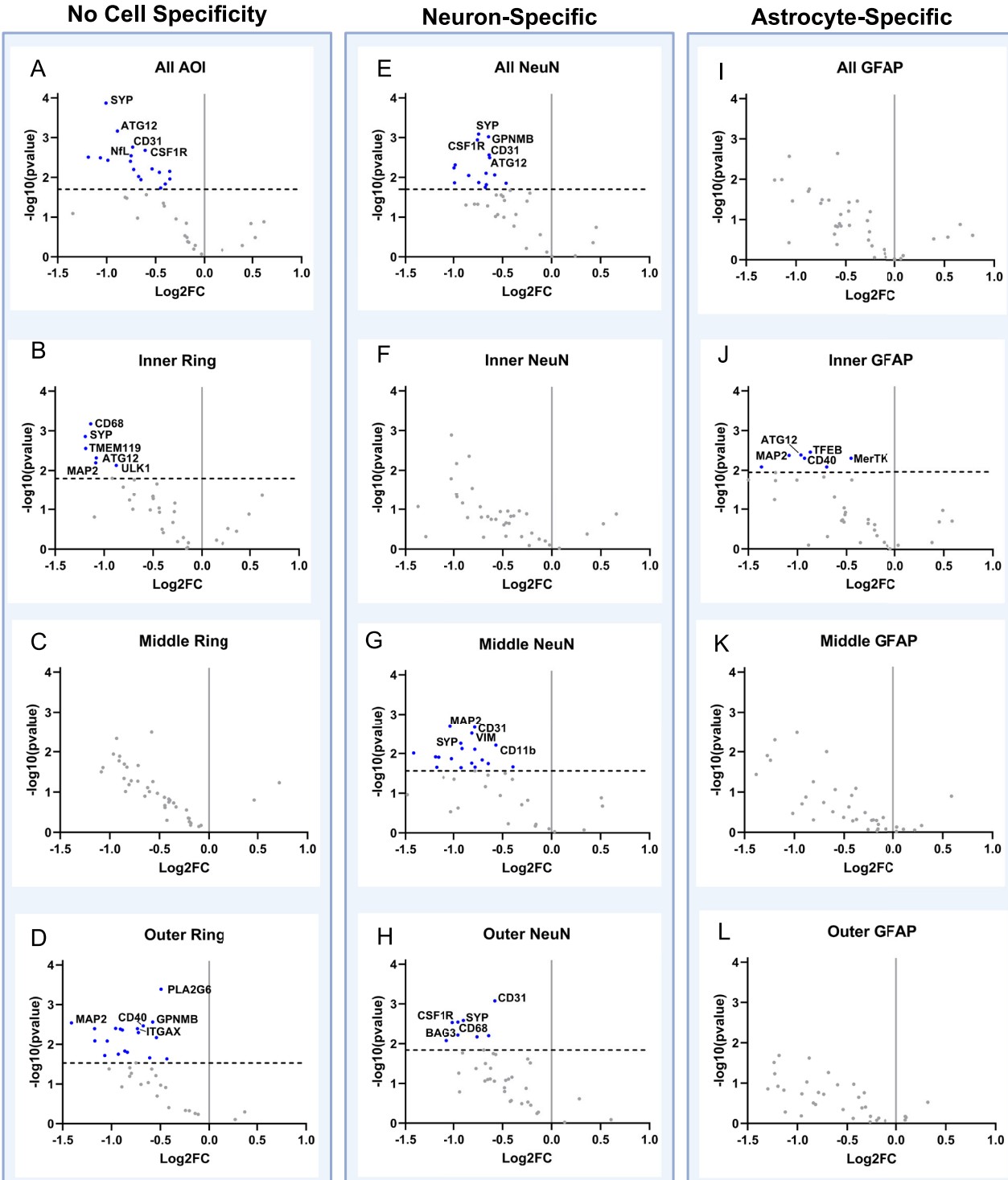

**4-Week Antibiotic Baseline to Control** Antibiotic: *n* = 4 Control: *n* = 3

to the microelectrode implant site. Temporal comparison of the untreated control mice also demonstrated that five of the twelve comparisons indicated differential protein expression (Supplementary Fig. S5, Supplementary Table S2). However, with the untreated control mice, the two groups with the largest differential expression only indicated three or four differentially expressed proteins each, with the remaining comparisons only showing one differentially expressed protein each. The spatial organization of differential protein

expression was evenly distributed between AOIs, one total, two inner, one middle, and one outer AOI each demonstrated differential protein expression.

Spatial transcriptomic evaluation of the implant site was performed to understand how many genes were differentially expressed, and which pathways and molecular processes the genes are involved in. Mouse age and implant status were controlled to allow us to conclude transcriptomic changes based solely on antibiotic treatment

**Fig. 4 | Spatial proteomic response is treatment dependent at 4-Weeks post-implantation.** Volcano plots showing neural proteomic panel evaluation of 4-week antibiotic ($n = 4$) compared to 4-week control ($n = 3$) across the entire AOI (within 270 μm from the implant), the inner ring of the AOI (within 0–90 μm), the middle ring of the AOI (90–180 μm), and the outer ring (180–270 μm) for all cells, all neuron-specific cells (stained using an NeuN antibody), and all astrocyte-specific cells (stained using a GFAP antibody). Proteins with a negative Log2FC indicate downregulation (blue points) in antibiotic compared to control, while a positive Log2FC indicates upregulation (green points) in antibiotic compared to control. Unadjusted $p$-values are plotted and shown, but all statistical comparisons were done using adjusted $p$-values. The black dotted line indicates significance ($p_{adjusted}$ = 0.05). Each point on the volcano plot indicates a singular protein, with select proteins shown in the text. Comparisons with no cell specificity were made on the **A** entire AOI, **B** inner ring, **C** middle ring, and **D** outer ring. Neuron-specific comparisons were made on the **E** entire AOI, **F** inner ring, **G** middle ring, and **H** outer ring. Astrocyte-specific comparisons were made on the **I** entire AOI, **J** inner ring, **K** middle ring, and **L** outer ring. After normalization, an unpaired t-test was performed across respective groups for comparison. Unadjusted $p$-values were corrected using the Benjamini-Hochberg false discovery rate method to account for random significance. A few insignificant proteins were excluded from the plots due to high log2FC values, causing skewing and making visual representation difficult. Five significantly differentially expressed proteins were labeled due to space. Refer to Table 2 for the full list of significantly differentially expressed proteins. Created in BioRender. Capadona, J. (2025) https://BioRender.com/p93j360.

versus no treatment control. Of note, comparisons were not made between implanted (antibiotic and control) and unimplanted, healthy mice. Future studies could incorporate healthy mouse comparisons to bolster analysis and comparisons across groups. Very few studies have been performed with spatial transcriptomic analysis of the intracortical microelectrode-tissue interface to date[38,39]. However, the Nano-String GeoMx system used here is uniquely capable of collecting the entire tissue of interest, rather than orientated circular regions forming a grid within the tissue being analyzed. Here, the whole mouse transcriptome was first filtered using quality control steps in the NanoString GeoMx software (see Methods for details on filtering) leaving a total of 8259 genes. Of the 8259 genes included in our analysis, 490 were differentially expressed at 4-weeks post-implantation ($n = 4$ for antibiotic, $n = 3$ for control), and 1375 genes were differentially expressed at 12-weeks post-implantation ($n = 3$ for antibiotic, $n = 3$ for control) (Fig. 5A, B). Out of all differentially expressed genes, only 52 are shared between the 4- and 12-week time points, indicating consistent temporal changes. No corrections were made for comparisons.

To further our understanding of the physiological responses associated with differences in microbial composition and the implications on neuroinflammation and brain health, we completed a pathway analysis using the Advaita iPathways software. The differential gene expression detected in our study implicated dozens of biological pathways and functions related to neural health. Here, for brevity and focus, we only discuss pathways in which a high proportion of genes associated with the pathway were differentially expressed at either 4-weeks or 12-weeks post-implantation, or pathways in which many differentially expressed genes switched between up- and down-regulated between the 4- and 12-week timepoints. Some pathways discussed were not identified in iPathways as being significantly altered. However, since there are still many differentially expressed genes implicated in the pathway, it was deemed important to discuss the possibility that the altered genes could influence the functions of said pathways.

At 4-weeks post-implantation, there were 20 significantly upregulated genes of 122 genes associated with ribosomal protein function (KEGG: 03010, pathway $p = 0.0002$) in antibiotic-treated animals compared to control (Fig. 5C, Supplementary Fig. 6A). At 12-weeks post-implantation there were 19 differentially expressed genes in the same pathway (pathway $p = 0.696$) with only six genes upregulated and 13 downregulated in the antibiotic-treated animals compared to the control (Fig. 5D, Supplementary Fig. 6B). Ribosomal genes are involved in regulating immune responses, such as *Rps3*, which has multiple functions including regulating the production of inflammatory marker, *NF-Kβ*, and is upregulated at the 4-week time point ($Log_2(FC) = 0.277$, $p = 0.0387$)[44]. Neurodegeneration is an important pathway to consider as it relates to long-term neural health. Healthy, firing neurons can only be detected within ~150 μm from the intracortical microelectrode site[45]. Consequently, evidence of neurodegenerative pathways near the implant site detected by spatial transcriptomic analysis is particularly interesting. At 4-weeks post-implantation, there were 17 differentially expressed genes associated with the neurodegenerative pathway (KEGG: 05022, pathway $p = 0.859$, Fig. 5C), with 49 differentially expressed genes at 12-weeks post-implantation (pathway $p = 0.596$). Although this pathway is not significant, it is important to note how many more genes are differentially expressed that belong to the pathway in the 12-week time point compared to 4-weeks. Looking individually at processes within the neurodegeneration pathway, at 4-weeks post-implantation, four genes are differentially expressed in ubiquitin-proteasome system (UPS) disruption (three upregulated, one downregulated, Supplementary Fig. 7A), and five genes are upregulated in the mitochondrial dysfunction (Supplementary Fig. 7B) (Fig. 5C). At 12-weeks post-implantation, 10 genes associated with UPS disruption were differentially expressed (eight upregulated, one downregulated, Supplementary Fig. 8A), 23 genes associated with mitochondrial dysfunction and mitophagy were differentially expressed (15 downregulated, eight upregulated, Supplementary Fig. 8B), and six associated with tau protein accumulation were differentially expressed (five upregulated, one downregulated, Supplementary Fig. 8C) (Fig. 4D).

Inflammation and the immune response are crucial when characterizing the body's response to an intracortical microelectrode. NOD-like receptors are key regulators of inflammation, especially the innate immune response[46]. At 12-weeks post-implantation, 19 of 69 total genes associated with NOD-like receptor signaling (KEGG: 04621, $p = 0.019$) were differentially expressed (10 upregulated, nine down-regulated). Conversely, at 4-weeks post-implantation, only two genes were upregulated in the NOD-like receptor signaling pathway ($p = 0.354$). The COVID-19 disease pathway (KEGG: 05171) was also impacted at 4-weeks post-implantation where 18 out of 126 genes were differentially expressed (17 upregulated, $p < 0.0001$). Of the effected genes, there was an upregulation of *NF-kB* Inhibitor B (*Nfkbib*, log2FC = 0.387, $p = 0.026$), which inhibits the activity of the common immune and inflammatory initiator, NF-kB[47]. It is also worth noting that 16 of the 18 differentially expressed genes are from the ribosomal dysfunction group, which has previously been noted in Fig. 5C and reinforces their involvement in the inflammatory process. By 12-weeks post-implantation, the number of differentially expressed genes in the COVID-19 pathway increases to 23, however without significance ($p = 0.635$), including a lack of significance of *Nfkbib* ($Log_2(FC) = 0.224$, $p = 0.373$). Like the 4-week timepoint, most genes impacted are from the ribosomal dysfunction group. To further explore the changes to inflammation and neurological processes from antibiotic treatment, the gene ontology (GO) of biological processes was analyzed. GO terms are sets of genes that contribute to a specific biological, cellular, or molecular function. There were 156 significantly impacted GO terms at 4-weeks post-implantation, 227 at 12-weeks post-implantation, and only one GO term was significant and shared across both groups (Cytoplasmic translation, GO: 0002181). G protein-coupled receptor signaling is responsible for many cellular functions, including neuro-transmission, cell metabolism, and immune response[48]. At 4-weeks post-implantation, there are only 12 out of 711 genes differentially

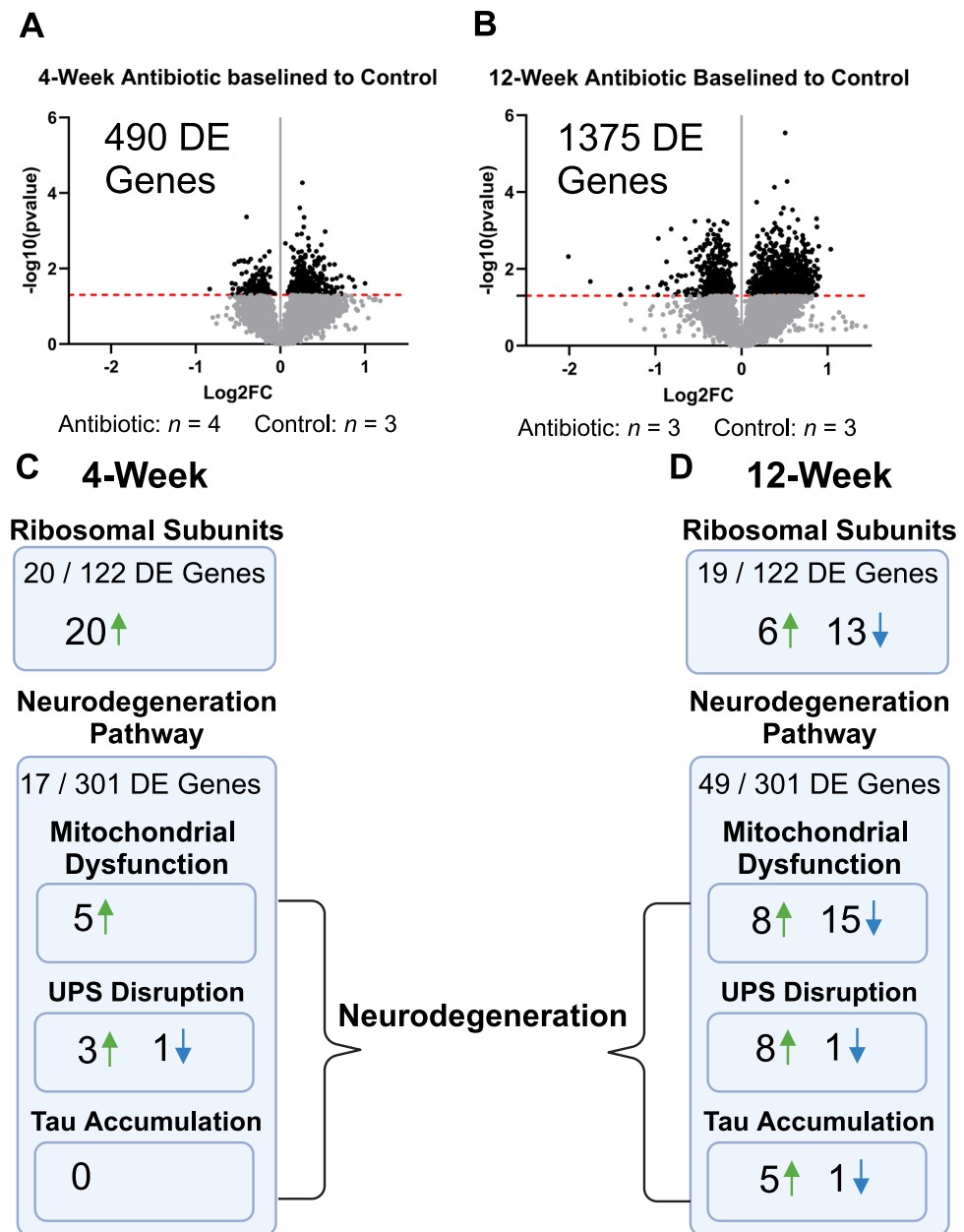

**Fig. 5 | Spatial transcriptomics reveals treatment- and time-dependent effects after implantation.** Transcriptomic data composing the full AOI of the implant site (within 270 μm from the implant site). Volcano plots are shown evaluating gene expression at 4- and 12-weeks post-implantation. Unadjusted p-values are plotted and shown. The black dotted line indicates significance ($p_{value} = 0.05$). Each point on the volcano plot indicates a singular gene. There were **A** 490 differentially expressed (DE) genes at the 4-week time point between antibiotic and control, which increased to **B** 1375 DE genes at the 12-week time point. Some pathways of note that were impacted by the antibiotic treatment at **C** 4-weeks post-implantation include the ribosomal subunit structure and neurodegeneration pathways, with changes occurring temporally as seen **D** in the 12-week time point. Each gene's raw data / count underwent Q3 normalization followed by statistical analysis using a custom MATLAB R2021a script to perform unpaired t-tests between samples. Unadjusted p-values were used for all further comparisons in the iPathways software suite. At 4-weeks post-implantation, $n = 4$ for antibiotic-treated and $n = 3$ for control, and at 12-weeks post-implantation, $n = 3$ for antibiotic-treated and $n = 3$ for control. Created in BioRender. Capadona, J. (2025) https://BioRender.com/p93j360.

expressed in the G protein-coupled receptor signaling pathway (GO: 0007186, GO $p = 1.000$), which grows to 238 out of 711 genes at 12-weeks post-implantation (227 upregulated, 11 downregulated, GO $p < 0.0001$). Nervous system processes (GO: 0050877) have only 30 out of 998 at 4-weeks post-implantation (majority downregulated, GO $p = 1.000$) and 278 out of 998 genes differentially expressed at 12-weeks post-implantation (majority upregulated, GO $p < 0.0001$). When looking at immune response processes at 4-weeks post-implantation, there are only 3 out of 51 genes differentially expressed for the regulation of production of molecular mediators of immune response

(GO: 0002700, GO $p = 0.593$) and 3 out of 37 differentially expressed for positive regulation of production of molecular mediators of immune response (GO: 0002702, GO $p = 0.380$). Contrast this to 12-weeks post-implantation where 16 out of 51 (GO $p = 0.007$) and 12 out of 37 (GO $p = 0.013$) differentially expressed genes, respectively. Additionally, at 12-weeks post-implantation, there are three out of five genes significantly expressed in both mucosal immune response (GO: 0002385, GO $p$-value 0.035) and organ-specific immune response (GO: 0002251, GO $p = 0.035$). Lastly, at 12-weeks post-implantation, the most widely impacted pathway is olfactory transduction, with 210 DE

genes out of 482 (KEGG: 04740, $p < 0.0001$). On top of their classical ability to provide odorant detection, recent literature has implicated olfactory receptors in inflammation, including presence on and activation of macrophages and monocytes[49]. While the connections between olfactory receptors and inflammation are evolving and less studied, there is evidence suggesting upregulation of several genes during inflammation, including *Olfr1014* and *Olfr65*, both of which were differentially expressed here[50]. Olfactory transduction at the 4-week time point is impacted much less, having only four DE genes out of the 482. Olfactory receptors may prove an interesting area for future studies to explore inflammation in the brain.

While it is difficult to draw conclusions based on pathway and GO term analysis, the 12-week time point has a more altered immune and neurological response than the 4-week time point, given the abundance of significantly impacted genes, pathways, and GO terms. It is possible that long-term antibiotic treatment and remodeling of the gut microbiome significantly influence the body's response to intracortical microelectrode implantation. Yet, it remains important to remember that the antibiotic treatment used in this study was not proposed to be a treatment to overcome the effects of microbiome invasion at the implant site. Antibiotic treatment was used as a standard method to alter the gut microbiome composition to determine if changes in gut microbiome composition correspond to changes in the composition of invading microbiota, impacting microelectrode performance. As there are thousands of genes, hundreds of pathways, and thousands of GO terms, not all were analyzed and discussed in this study. A file containing all gene data with their respective log2FC and p-values, as well as pathways and GO terms are included in the Supplementary Information for reader accessibility and transparency.

## Discussion

Intracortical microelectrodes are used for neuroscience research and clinical brain-machine interface systems, but the recording performance decreases over prolonged implantation periods[3]. A major factor in the degradation of implant performance is the neuroinflammatory response[4]. Degradation of BBB integrity is an appreciable consequence of microelectrode-mediated neuroinflammation and can allow previously restricted blood-borne components to enter the brain parenchyma and amplify the neuroinflammatory response[4,6,15,51,52].

During disease and injury, components of the gut microbiome can directly infiltrate the brain, causing a local inflammatory response[19]. Gut-resident microbiota activate and modulate neuroinflammatory processes implicated in schizophrenia, depression, anxiety, Alzheimer's, Parkinson's, and stroke[19,21]. Despite this link, there have been no reports examining the infiltration of gut-resident microbes into the brain following microelectrode implantation and associated device performance. Therefore, the principal hypothesis of the current study was that damage to the BBB caused by microelectrode implantation facilitates/permits the infiltration of gut-resident microbes into the brain, contributing to the chronic neuroinflammatory response and decreased performance of intracortical microelectrode arrays.

Recent studies have expanded the scope of the human microbiome to include the brain[19], yet the brain-biota hypothesis remains controversial and under-explored. Contamination is a major concern in microbial DNA research, especially from typically sterile sites like the brain. To address this, we utilized computational and statistical methods to distinguish between sample-derived microbial DNA and technical contaminants. Our results indicate that the microbial DNA extracted from implanted brain tissues does not align with random contamination and could indicate a non-random pattern of microbial infiltration linked to BBB disruption following microelectrode implantation.

Bacteria can enter the brain at various stages of device implantation during the surgical procedure, ranging from contamination of the initially sterile device to transport by blood to the implantation site[13,53]. However, we have previously shown that after two weeks, residual endotoxin contamination was unable to impact the neuroinflammatory response to intracortical microelectrodes[11]. All implants in this study, in both control and antibiotic-treated mice, followed our established protocols to limit the introduction of bacteria from the microelectrode. While it is unlikely that the responses observed here are due to implant contamination with viable bacteria or external factors entering the wound margins, it cannot be ruled out that the shifts in bacterial sequence composition observed here represent contamination from other body sites. However, these shifts would still indicate some sort of disruption of the host microbiome following intracortical microelectrode implantation, warranting further investigation.

Further, several studies in rodents and humans have shown that traumatic brain injuries are accompanied by increased intestinal permeability and intestinal barrier dysfunction[17,54,55]. Therefore, we were particularly interested in understanding the role that microbes that reside in the intestines may have on microelectrode performance if they invade the brain tissue following microelectrode implantation.

In this investigation, we have demonstrated that microbes, some of which are associated with the gut microbiome, can invade the brain tissue proximally to the microelectrode implantation site following microelectrode implantation (Fig. 1). To explore the link between the gut microbiome, damage to the BBB, microbial invasion of the brain, and device recording performance, we treated mice implanted with intracortical microelectrodes continuously with an antibiotic cocktail to limit the reservoir of microbes in the gut (Figs. 1–3, Supplementary Fig. S1). Depletion of the gut microbiome via systemic antibiotic treatment was associated with better microelectrode performance up to 5 weeks post-implantation, lower bacterial sequence diversity, lower abundance of distinct implanted-associated bacterial features, and lower abundance of pro-inflammatory microglia/macrophage proteins in the associated brain tissue as compared to background following implantation (Figs. 1–4). Long-term antibiotic treatment resulted in significantly decreased intracortical microelectrode performance at 12 weeks post-implantation, which could be associated with the substantial genetic changes (Fig. 5). After 12 weeks, the composition of microbial sequences extracted from the brain tissue was still distinct from that observed in the background, while the antibiotic-treated mice showed no distinct clustering (Fig. 2D, E). A lack of change across the antibiotic-treated group suggests that treatment may be keeping the bacterial composition closer to baseline levels, although this cannot be distinguished from background changes in the microbiome at this stage. It is important to recognize that even subtle changes in gut microbiota composition have been linked to changes in brain health[19,21].

The most abundant microbial sequences identified in background samples and implanted brain samples following microelectrode implantation were classified as Firmicutes and Bacteroidota, the dominant bacterial phyla in the gut of at least 60 mammalian species[56]. Additionally, bacterial sequences not identified in fecal matter or unimplanted brain tissue were found in microelectrode-implanted brain tissue (Fig. 1C, D). The brain is a distinct environment from the colon. Therefore, it is not without reason that invading species would be unable to thrive in the brain and are readily removed – dead or alive.

Together, our results suggest the host microbiome is disrupted following intracortical microelectrode implantation in mice and may result in the translocation of bacteria to the brain from potentially multiple sources. To theorize the origin of these distinct, implantation-associated bacterial features, we analyzed whether they were also detected in the gut (gut-resident bacteria) (Fig. 1D). While around ~20% of these features can also be found in the gut, ~30% originate from a location beside the gut, presenting a need to investigate other potential reservoirs of bacteria in the body. While estimates vary depending on the source, there is a consensus that bacteria at least

equal if not far outnumber the number of cells in the human body, but not all come from the gut microbiome[57]. In fact, different body sites such as the skin, oral cavity, lung, and nasal cavity each develop their own individual microbiomes[58–60]. In a paper published in Nature in 2022, Hosang et al. concluded that by altering the microbiome of the lung, it was possible to modulate immune signals in the brain by impacting microglia[61]. Combined with our results, the Hosang study and similar literature[62] highlight the importance of expanding our horizons beyond just the gut microbiome in future studies to understand how other bacterial sources in the body may influence the brain.

However, it is important to note that bacteria that may reside in the brain have been poorly characterized, leading to the possibility that there are bacteria in the brain after implantation that are unable to be matched to current databases. The results of the present study demonstrate that the bacterial environment of the host is significantly impacted after intracortical microelectrode implantation, and this may influence the type of bacteria present at the microelectrode-tissue interface. The modulation of the bacteria environment is associated with changes in the intracortical microelectrode recording performance and the neuroinflammatory response near the implant site, which has been identified as a major factor influencing microelectrode performance[3–5].

The acute improvements in microelectrode recording performance reported here, in combination with the alterations to bacterial sequence composition at 4-weeks, indicate a potential avenue for new therapeutics to improve brain implant function and mitigate neuroinflammation. While not designed to be a solution, the extreme antibiotic treatment utilized here represents proof of concept for designing a more tailored approach to target specific gut-derived bacteria strains that may be exacerbating the inflammatory response of the brain. For example, at 4-weeks post-implantation, differences in microbe composition and abundance between antibiotic-treated and untreated control groups were largely in the phylum Firmicutes. Specific strains of Firmicutes have been linked to neurodegenerative diseases such as multiple sclerosis, autism, depression, and schizophrenia[63]. Lower abundance of Firmicutes in the antibiotic-treated group may be contributing to the improvements in recording performance through a reduced neuroinflammatory response or even through secondary mechanisms. Recent studies have identified diverse microbiota-derived bioactive molecules that are implicated in inflammatory processes ranging from the gut to the brain[24]. In a pilot study, we examined the fecal matter of a human subject implanted with a brain-machine interface[64] and found the composition of microbes at the phylum level to be >90% consistent after human and mouse intracortical microelectrode implantation (Supplementary Fig. 9). Therefore, therapeutic approaches designed to provide an optimal balance of microbes such as Firmicutes may be beneficial for improving microelectrode recording performance and can be readily tested in mouse models due to the consistency between human and mouse gut microbiome.

Antibiotics are commonly used as part of post-surgery treatments and have been investigated acutely in mice, showing a beneficial reduction of glial encapsulation post-implantation of MEAs[65–67]. However, it is unlikely that regular antibiotic treatment throughout the duration of microelectrode implantation would represent a practical clinical solution to improved microelectrode performance. Long-term dosing of antibiotics is well known to be detrimental to overall health[68]. Chronic administration of antibiotics can lead to the selection of antibiotic resistant bacteria, as well as shift stable, healthy populations of bacteria in the local microbiome into unstable and/or unhealthy ones[69,70]. One strategy that is more commonly employed after prescribing antibiotics is pairing with probiotics[71]. Perhaps, a shorter duration of antibiotic treatment followed by a specific probiotic cocktail to promote the invasion of more benign or even neuroprotective bacteria can be possible with time. If bacteria do infiltrate the

brain, the application of antimicrobial coatings to the microelectrode substrate[72,73] to prevent the population of brain tissue adjacent to the microelectrodes with invasive microbes represents a promising materials-based approach to overcome the newly identified problem. Further investigation into the development of vaccines to regulate T-cell programming[74,75] towards specific strains of gut-derived microbes, such as Firmicutes, could provide a means to 'prime' the adaptive immune system prior to microelectrode implantation.

There were one control and two antibiotic animals at 4-weeks post-implantation that had similar bacterial compositions to that of the 12-week post-implantation brain. Such results may indicate that an animal's response to treatment and/or bacterial abundance may vary, possibly due to their individual immune response or due to the variability reported by many labs in the damage to the BBB following intracortical microelectrode implantation[6,36,76–78] – either merits further investigation.

It is important to note the limitations of 16S bacteria analysis and its associated assumptions. First, while the work outlined has demonstrated the presence of 16S sequences at the site of implantation and in the brain, 16S measurement does not confirm the presence of live bacteria and may indicate either dead or fragments of bacterial DNA. It is possible that the sequences detected are from fragmented DNA inside macrophages or other immune cells that ingested bacteria and then invaded the brain after implantation. The confirmation of live bacteria and the presence of a microbiome in the brain after implantation necessitates further work, including comprehensive live bacteria culture of implanted brain tissue and appropriate controls for excluding the contribution of background noise. Second, 16S sequencing results provide a view of bacterial composition and relative abundance, which is not indicative of total bacterial quantity. Third, additional experimental controls are needed to improve the resolution between background (e.g., technical contamination) and microbial signal from whole tissues with low microbial biomass, such as DNA extraction techniques which deplete host gDNA before extraction of bacterial DNA. Lastly, translocation of intestinal bacteria, particularly anaerobic bacteria, occurs infrequently[79], raising the question of whether the identified 16S bacteria here are metabolically active and growing. However, with the rise of 16S sequencing, previous teachings of anaerobic bacteria translocation may change. To better investigate the effect of bacteria infiltration and antibiotic treatment, we explored proteomic and transcriptomic analysis around the implant site.

Cell-specific spatial proteomic and spatial whole transcriptome analysis revealed that antibiotic treatment impacted dozens of proteins and hundreds of genes at both 4- and 12-weeks post-implantation. We postulate that the large number of downregulated proteins in the antibiotic-treated group at 4-weeks post-implantation may influence a more favorable environment for improved neural recording quality. Many proteins involved with macrophage and microglial response were downregulated (MerTK, CD40, CD68, TFEB), which may account for a reduced neuroinflammatory response and improved recording performance[6,80]. Additionally, NF-kB Inhibitor B, an inhibitor of the immune instigator, NF-kB, was significantly upregulated at 4-week post-implantation but not 12-weeks post-implantation. Such a gene may be an important marker for quantifying and understanding the immune response to implanted IMEs and could be a contributor to the differences in recording performance observed at 4-weeks. Transcriptomics also revealed overexpression of genes associated with ribosomal subunit structures at 4-weeks post-implantation in the antibiotic-treated group compared to the control, but a majority downregulated at 12-weeks post-implantation in antibiotic compared to the control. Ribosomal dysfunction is commonly associated with neurodegenerative diseases such as Parkinson's and Alzheimer's, which may be tied to the switch from significantly improved recordings to worse recordings at those respective time points[81,82]. Proteins

associated with autophagy and neural health were downregulated at 4-weeks post-implantation as well (ATG12, SYP, MAP2, NfL, NeuN). Such protein downregulation may be a precursor to the significant drop in implant function observed at week 7 and onward, as loss of autophagy and neural health are often associated with neurodegenerative diseases such as Parkinson's disease[83]. This is reflected both in the decline of implant function and uptick in differentially expressed genes of the neurodegenerative pathway at 12-weeks post-implantation (49 genes) vs 4-weeks post-implantation (17 genes), including higher dysfunction in mitochondria, UPS disruption for clearing misfolded proteins, and tau protein accumulation, all of which are common indicators of neurodegeneration and disease states[84–86]. The 12-week group also showed 19 DE genes involved in the NOD-like pathway, a regulator of the innate immune and inflammatory response. The upregulation of NOD-like genes could indicate increased inflammation in the antibiotic group compared to control, which may contribute to the sharp decline observed in recording performance at that time point. Furthermore, the olfactory transduction pathway was heavily impacted at 12 weeks point-implantation (210/482 DE genes) compared to 4 weeks post-implantation (4/482 DE genes), which may indicate further investigation into how olfactory receptors can influence the immune response in the brain. Overall, the 12-week timepoint had many more pathways and GO terms impacted compared to the 4-week time point, which may explain the drop in recording performance. In contrast, proteomics showed more significantly impacted proteins at 4-weeks point-implantation, which, in combination with the transcriptomics, we believe helps explain the improvements to recording performance. Such differences between analyses highlight the importance of using both proteomics and transcriptomics to understand the effects of treatment. Continuing to develop an in-depth understanding of changes in gene and protein expression following changes in microbiome composition could identify pathways for molecular or gene therapy[87] approaches to modulating the innate immune response following intracortical microelectrode implantation. It is important to note that a limiting factor of this study was the lack of healthy, non-implanted control animals to compare proteomics and transcriptomics. Even though we can conclude how antibiotic treatment can influence the implanted microenvironment with regards to no treatment, it is difficult to state how these changes influence inflammation and brain health compared to a healthy mouse. Future studies should employ healthy controls to help answer whether treatment can reduce inflammation and improve neural health back to baseline levels.

In conclusion, the current study reports the detection of bacterial DNA sequences from normally gut-resident microbes and microbes of a currently unknown origin in DNA extracted from brain tissues after intracortical microelectrode implantation which were not detected in background samples. Further, it is possible to modulate the neuroinflammatory response following implantation and microelectrode performance by depleting gut bacteria using an antibiotic cocktail, which was associated with a reduced abundance of distinct, implantation-associated bacterial sequences. Our findings suggest that alternative strategies could locally target invading bacteria rather than systemically manipulating all bacteria in the host, which may introduce complications such as weakened immunity, antibiotic resistance, and long-term health issues. The importance of microbes invading the brain extends far beyond device performance and tissue reaction alone and raises concerns about unintended consequences or ripple effects. Some microbial taxa identified from sequences in this initial study have been previously associated with neurodegenerative symptoms and diseases. This raises long-term concerns and requires the development of a comprehensive approach for the optimal integration of neuro-modulatory devices within the brain tissue. While the focus of the current study was solely intracortical microelectrodes arrays, devices with a larger footprint could presumably produce an even more pronounced effect if the nature and extent of BBB damage determines the microbial invasion of the brain. Future studies should further investigate both the mechanism of invasion and approaches to mitigate the invasion and colonization of the brain by foreign microbes.

## Methods

All procedures and animal care protocols were performed in compliance with Case Western Reserve University's Institutional Animal Care and Use Committee (IACUC) approved protocol 2013-0106. All mice were housed in a 12-h dark/light cycle, following ambient temperature of 68–79 °F and humidity of 30–70% outlined in the guide for the care and use of laboratory animals, 8th edition. All mice used were of the male sex to stay consistent with past studies for the sake of comparison and historical data. Past work in our lab has indicated that there are minimal sex-based differences[80], unless testing specific treatments or circumstances such as treatments that impact estrogen levels.

### Intracortical microelectrode array preparation

A 16-channel single-shank intracortical microelectrode array (A1x16-3 mm-50-177-Z16, iridium electrode sites, NeuroNexus Technologies, Ann Arbor, MI, USA) was used to record neural action potentials of the motor cortex (M1). Alternatively, a non-functional silicone implant of the same dimension was used to assess neuroinflammation and microbial composition. In a Faraday cage setup, each MEA to be implanted underwent EIS testing with a Gamry Interface 1010E Potentiostat (Gamry Instruments, Warminster, PA, USA) consisting of each electrode site as the working electrode, a platinum wire as a counter electrode, and an Ag|AgCl electrode stored in KCl reference electrode for measurements. EIS was performed in 1x PBS (pH = 7.4) over a range of 1 to $10^6$ Hz (12 points per decade) with an AC voltage of 50 mV. The impedance magnitude at 1 kHz was used to confirm functionality with expected values between 150 and 550 kHz. Following EIS verification, MEAs were cleaned using 70% ethanol and DI water to remove any residual 1x PBS and optically imaged using a Keyence Optical Microscope (Keyence Corporation, Osaka, Japan) at a magnification of 150x for visual inspection. Non-functional dummy implants were cleaned using the same protocol as functional implants. After cleaning, both implant types were sterilized using cold gas ethylene oxide.

### Intracortical microelectrode implantation

Male *C57BL/6* mice were obtained from Jackson Labs aged 8–10 weeks and separated to single housing before surgery. Each cohort of animals followed the experimental timeline outlined in Fig. 6A with end points of 4-weeks and 12-weeks post-implantation. All surgical procedures followed established protocols in our combined labs[88,89]. Briefly, mice were anesthetized in an isoflurane chamber (3.5% at 0.8 L/min $O_2$). Anesthetic plane was monitored via paw pinch and respiratory rate. Following anesthesia, the incision site was shaved, nails trimmed, and eye lube applied to prevent eyes from drying out. The mouse was then mounted to the stereotaxic frame (David Kopf Instruments, Tujunga, CA, USA) via bite bar and ear bars. Anesthesia was maintained at 0.5%–2.0% at 0.8 L/min $O_2$ via nose cone inhalation. Topical analgesic Lidocaine was applied to the surgical site[80]. Subcutaneous analgesics buprenorphine and meloxicam were administered before surgery. No systemic antibiotic was administered for any group for surgeries. Once mounted, the surgical site was cleaned and sterilized using betadine and 70% isopropyl alcohol in alternating scrubs. A one-inch incision was made along the midline of the scalp and skin retracted using alligator clips to expose the skull. A swab of hydrogen peroxide was applied to the skull to dry out and make cranial sutures more visible. A thin coat of Vetbond tissue adhesive (Catalog #70200742529, 3 M, Saint Paul, MN, USA) was applied to the skull to prepare for dental

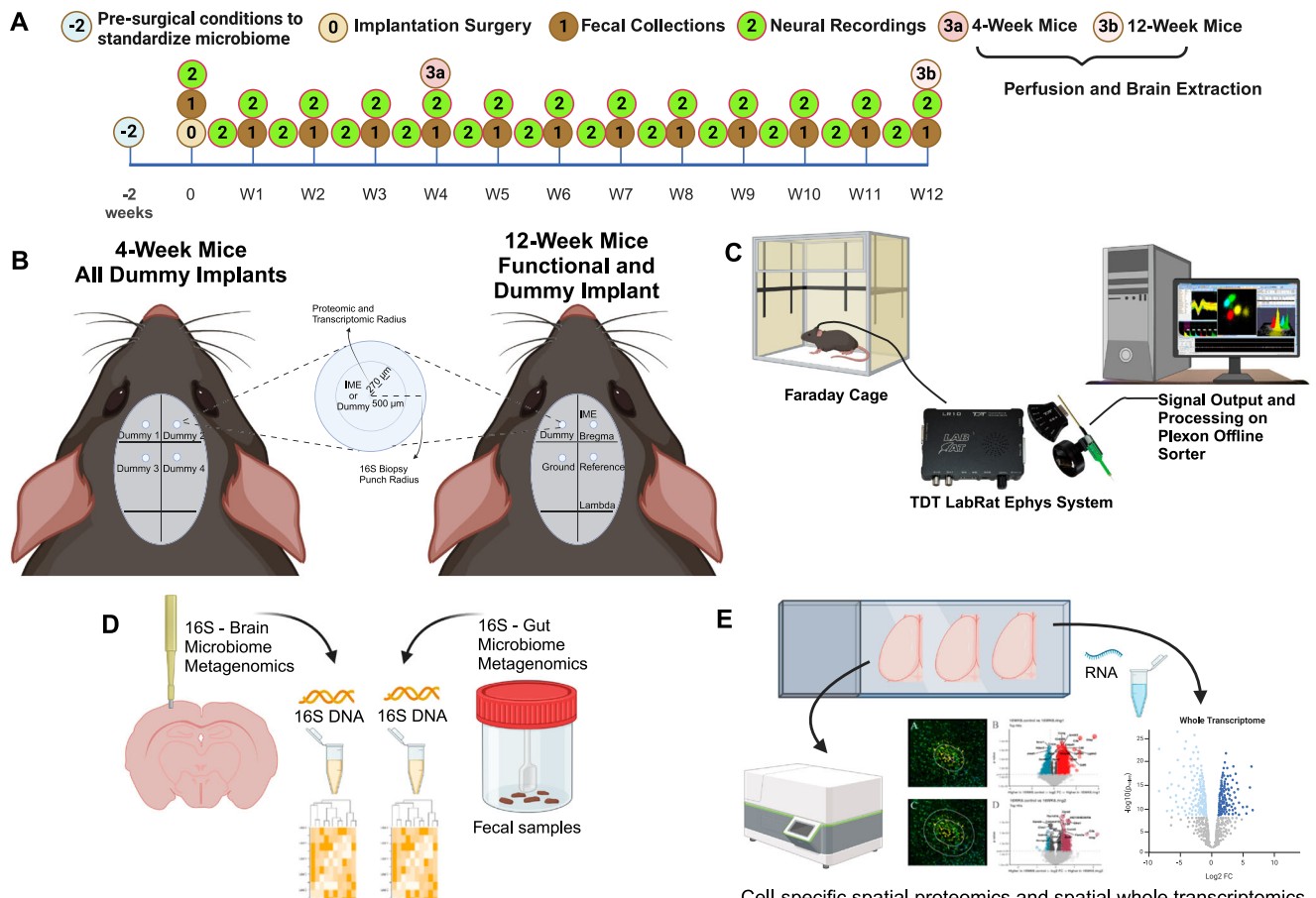

**Fig. 6 | Experimental design outlining the timeline for each cohort. A** The unimplanted mice were sacrificed two weeks after housing separation for analysis. The 4- and 12-week post-implantation animals undergo implantation, fecal collection, neural recordings, and perfusion at their endpoint. **B** The 4-week cohort received four non-functional dummy implants and the 12-week cohort received one non-functional dummy implant and one functional intracortical microelectrode (IME) implant with respective ground and reference wires. **C** 12-week functional implanted mice were recorded using a commutator hooked up to the TDT LabRat Ephys system. **D** 16S analysis was done on a biopsy of brain tissue around the implant site and on fresh fecal matter collected from each animal. **E** Cell-specific spatial proteomics and spatial transcriptomics were performed on various brain samples sectioned onto microscope slides. Created in BioRender. Capadona, J. (2025) https://BioRender.com/p93j360.

cement adhesion. Using a 1.35 mm drill bit attached to an electric drill, two craniotomies were drilled into the skull to implant, one for the non-functional implant and one for the functional intracortical microelectrodes; two additional craniotomies were made as well for insertion of the ground and reference wires. Using a dura pick, the dura was carefully removed before implantation to expose the implantation site. Mice were divided into two groups, one surviving 12-weeks and a second one 4-weeks post-implantation. The group which survived for 12-weeks post-implantation were implanted with the functional implant inserted 1 mm deep into the primary motor cortex (2 mm anterior to bregma, 2 mm dextral to midline) with reference (2 mm posterior to bregma, 2 mm dextral to midline) and ground wires (2 mm posterior to bregma, 2 mm sinistral to midline) inserted into the brain (Fig. 6B). The non-functional dummy implant was inserted at 2 mm anterior to bregma and 2 mm sinistral to midline. The mice which survived for 4-weeks post-implantation received four non-functional dummy implants inserted at each of the above four coordinate sites (Fig. 6B). Once a wire or implant were inserted, they were secured in place using Kwik-Sil silicone elastomer (World Precision Instruments, Sarasota, FL, USA) to close off the opening of the brain. Following which, Teets dental cement (A-M Systems, Sequim, WA, USA) was applied to anchor the wires and implants to the skull and prevent movement over the course of the study. Following surgery, 5–0 monofilament polypropylene sutures were used to close the surgical

site and promote healing of the skin and tissue. A daily dose of analgesic meloxicam and twice daily buprenorphine were administered for 72 h post-operation to manage pain.

### Treatment and preparation
To manipulate the composition of the gut microbiome we used a high-dose antibiotic mixture administered to the mice. A mixture of Ampicillin (Millipore Sigma, A5354), Clindamycin (Millipore Sigma, PHR1159), and Streptomycin (Millipore Sigma, S9137) were provided via sterile drinking water at a concentration of 0.33 mg/mL for each antibiotic. Such antibiotics were chosen based off previous literature to provide broad spectrum capacity and effect on the gut microbiome[90]. Animals drank *ad libitum* from the water and was replaced every 3 days. Control mice received normal food and water diets. All animals were singly housed in a reversed 12-hour light cycle.

### Neurophysiological recording and analysis
Electrophysiological recordings were taken from the functional intracortical microelectrode twice weekly beginning on day 0 of the implant and continuing throughout the duration of the 12-Week implants to assess device function. The data collector was blinded to the animal's group to eliminate any inherent bias in recording data collection. Similarly, all recordings were analyzed blindly to remove bias. To record, animals were briefly anesthetized using isoflurane at

3.5% and 0.8 L/min $O_2$. While anesthetized, animals were placed into an acrylic box surrounded by a Faraday cage and connected to the recording equipment (Fig. 6C). The functional intracortical microelectrode was connected to a 16-channel ZIF-Clip Headstage (Tucker-Davis Technologies Inc., Alachua, FL, USA) which was part of a 32-channel motorized commutator system (Catalog #ACO32, Tucker-Davis Technologies Inc) for free movement without damaging the wires. The commutator was then connected directly to a Lab Rat Ephys system (Tucker-Davis Technologies) and into a laptop for processing. Using the Synapse recording software (Tucker-Davis Technologies), recordings were taken at a sampling rate of 24414 Hz with a bandpass filter between 300 and 3000 Hz. Recording files were analyzed using Plexon Offline Sorter (Plexon Inc, Dallas, TX, USA) by first converting recordings to a usable.DDT format and importing into Plexon Offline Sorter for single unit analysis. Once imported, common median referencing was performed to reduce noise across channels. If any bad channels were known on the device, as observed during recordings for abnormal noise or activity levels, they were excluded to prevent interfering with the other channels. Once referenced, spikes were detected using settings of −4.00 standard deviation (σ) from the mean with waveform settings of 1720 μs for waveform length, a pre-threshold period of 410 μs, and a dead time of 1352 μs. To remove any possible artifacts that were not filtered out, amplitudes of ±500 μV were removed along with any identical spikes that were detected across 90% of the channels. If there were any particularly noisy portions of a recording (e.g., a wire getting caught or the animal interfering with the connection), the noisy intervals were removed using the interval selection tool. If high noise happened excessively during recording, all connections were checked between the device, headstage, commutator, and computer, and the recording was immediately redone. After filtering and detecting spikes, single unit sorting was performed using the K-Means scan algorithm in Plexon Offline Sorter to find between 1 and 4 units on each channel. From here, manual validation was performed on every channel to ensure that all units detected were correctly identified as single units. In many cases, units were deleted as they did not have typical characteristics for single units[91]. From this, the total number of active channels (channels picking up a single unit recording) for each recording was recorded to determine the % of active channels for each animal as a main outcome for recording performance. After manually checking for single units, files were exported and analyzed in MATLAB R2021a (Mathworks, Natick, MA, USA) to calculate peak-to-peak voltage ($V_{pp}$), noise levels, signal-to-noise ratio (SNR), spiking rate, and the number of single units detected per channel. $V_{pp}$ was calculated as the sum of the peak and trough signal of each waveform, noise was calculated at the root-mean-square of the channel after removing spikes, SNR was calculated by dividing $V_{pp}$ by the noise for each unit, and spiking rate was defined as the inverse of the median interspike interval per unit (from Plexon Offline Sorter). To summarize the data, the recording metrics for each individual intracortical microelectrode were averaged in their respective groups and time points. Recording data was binned into three distinct phases corresponding to the progression of neuroinflammation after implantation[92–94]: an acute phase (weeks 0–5), a sub-chronic phase (weeks 6–11), and a chronic phase defined as any time points after week 11. Sample size for the week-by-week Proportion of Active Electrodes was determined by summing the total number of electrodes multiplied by the number of animals in each group on a week-by-week basis. The sample size for acute, sub-chronic, and chronic Proportion of Active Electrodes was determined by summing the total number of electrodes multiplied by the number of weeks in each phase and the number of animals in each group. The sample size for the additional recording metrics was calculated by averaging the respective recording metric on a per-channel basis, and then summing up all unique channels across all animals within the same group and time point (e.g., if channel 1 of antibiotic animal 1 records an SNR on weeks 1,

2, and 3, then those values were averaged into a singular SNR value for channel 1 during the acute phase of antibiotic animal 1 to be used in further analysis).

Using Excel (Microsoft Corporation, Redmond, WA, US), a one-tailed proportions z-test was used for calculating statistical differences in the proportion of active electrodes within and across groups for the acute, sub-chronic, and chronic phases. Additional recording metrics were compared using R Studio 2022.7.1 + 554 (RStudio, PBC, Boston, MA, USA) and GraphPad Prism (Dotmatics, Boston, MA, USA) within and across acute, sub-chronic, and chronic neuroinflammatory phases using a Kruskal-Wallis test followed by a Benjamini–Krieger–Yekutieli test to adjust for multiple comparisons for non-normal distributions to increase statistical power and reduce type I errors. Statistical comparisons for antibiotic vs. control were only conducted within the same time point (acute antibiotic vs. acute control, sub-chronic antibiotic vs. sub-chronic control, chronic antibiotic vs. chronic control). No comparisons were made across time points and groups (acute antibiotic vs. chronic control, acute antibiotic vs. sub-chronic control, etc.) due to a lack of relevance concerning treatment effect. In all cases, statistical significance was defined at $p < 0.05$. For recording data box plots, whiskers represent minimum and maximum values, the box represents the first and third quartiles of the data, and the horizontal line indicates the median. All recorded numerical data were represented in the text as the mean ± SD.

## Fecal matter and brain sample isolation in mice
Weekly mouse fecal samples were taken from every singly housed mouse to provide samples to measure 16S bacteria of the gut throughout the duration of the study (Fig. 6A). On the day before fecal matter collection, each animal's housing was changed to fresh, sterile bedding. The next day a microcentrifuge tube of fecal matter was collected using sterile, disposable forceps before being stored in a −80 °C freezer until processing. At the end point of the study, animals were perfused to extract brain tissue (Fig. 6A). The researcher performing perfusion was blinded to the animal's group to eliminate any inherent bias in sample collection. Animals were injected with an IP anesthetic injection of Ketamine (100 mg/kg) and Xylazine (10 mg/kg). Sufficient anesthetic depth was determined by paw pinch before proceeding with perfusion. Once anesthetized, an incision was made along the abdomen just below the xyphoid process. A horizontal cut was made down the sides of the abdomen proceeded by two vertical cuts through the rib cage on both sides. The rib cage was held up using a pair of hemostats and diaphragm cut through to expose the heart and lungs. Once exposed, a butterfly needle was inserted into the left chamber of the heart and perfusate was pumped through the body. As soon as perfusate was turned on, a small cut was made on the right ventricle to allow for liquid to flow from the heart and prevent collapse of the heart. Approximately 15 mL of each solution was needed to perfuse the animal as indicated by a flushing of the liver. Following perfusion, the brain was extracted, and a biopsy punch was taken around an implant site for analysis. One implant site was biopsy punched to extract total DNA for 16S bacterial DNA analysis (Fig. 6B). The rest of the brain and remaining implant sites were left for proteomic and transcriptomic analysis by freezing in a mold containing Optimal Cutting Temperature (OCT, Sakura Finetek USA Inc, Torrance, CA, USA) and placed into a −80 °C freezer until processing (Fig. 6D).

## Human fecal matter collection
Human fecal matter was collected under an approved IRB protocol at University Hospitals Cleveland Medical Center in collaboration with the Reconnecting the Hand and Arm to the Brain (ReHAB) clinical trial (clincaltrials.gov #NCT03898804). At the time of sample collection, the study participant (coded RP1) was a 29-year-old male who had suffered spinal cord injury (C3/C4, AIS B), resulting in tetraplegia (motor-complete, sensory-incomplete), 8 years prior. His participation

in the ReHAB pilot clinical trial has been previously reported[64]. Briefly, RP1 received six 64-channel (8 × 8) Utah intracortical microelectrode arrays (Blackrock Microsystems, Salt Lake City, UT) implanted into various sensorimotor cortices for the purpose of restoring cortically-controlled movements of his paralyzed arm and hand, reanimated by functional electrical stimulation through composite flat interface nerve cuff electrodes. RP1 received the cortical implants 2 years and 6 months prior to fecal sample collection. Fecal matter was collected using standard procedures and transported in sterile and sealed containers, packaged in dry ice, for subsequent analysis. Samples were stored in a −80 °C freezer until processing until sequencing.

## 16S bacterial DNA sequencing

To analyze 16S bacterial DNA, all fecal matter and brain samples were sent to the Genomics Core on Case Western Reserve University's campus for DNA isolation and 16S sequencing. Samples were processed across four batches: (1) naïve unimplanted brains and their associated fecal samples, (2) 4-weeks and 12-weeks post-implantation brains, (3) fecal samples associated with 4-weeks post-implantation brains, and (4) fecal samples associated with 12-weeks post-implantation brains. At least one no-template library was sequenced in each run for a total of five sequencing blanks. Total DNA was isolated using the QIAamp PowerFecal Pro DNA Kit (Cat.No./ID:51804) and sequencing libraries prepared according to the 16S metagenomic sequencing library preparation protocol for the Illumina MiSeq system (Illumina Inc., San Diego, CA, USA)[95,96]. The V3-V4 region of the 16S rRNA small subunit (464 base pairs) was sequenced using an Illumina MiSeq (Illumina Inc., San Diego, CA, USA) with a paired-end 250-cycle run[95].

Raw paired-end ASVs were processed and assigned to an OTU using QIIME2 v2023.5[97]. Sequencing primers were trimmed from the reads and untrimmed sequences discarded using the Cutadapt plugin. Forward and reverse trimmed sequences were joined (minimum overlap $p = 4$ bases) and the merged sequences denoised with the DADA2 plugin[98]. Processed library depths ranged from 7292 to 16382 reads (median: 12100) for unimplanted brain samples, 26,661 to 129,331 reads (median: 40,028) for 4 weeks post-implantation brain samples, 17,351 to 37,695 reads (median: 27,160) reads for 12 weeks post-implantation brain samples, 1204 to 85,579 (median: 26,725) reads for pre-treatment and control group fecal samples, and 261 to 133,488 reads (median: 19,428) for antibiotic-treated fecal samples. Representative sequences were assigned to OTUs with a Naïve Bayes classifier trained on the V3-V4 region of 16S rRNA extracted from the SILVA v138.1 SSU (small subunit, 16S rRNA) Ref NR 99 reference sequences using the feature-classifier plugin[99].

Sequences were queried against the NCBI mouse (GRCm39 assembly, RefSeq accession GCF_000001635.27) and human genomes (GRCh38.p14 assembly, RefSeq accession GCF_000001405.39) as well as the prokaryote 16S rRNA sequences downloaded from the BLAST database (v5) using the rBLAST package v0.99.2[100]. Sequences with hits in the eukaryotic genomes (mouse, $n = 834$; human, $n = 2$) as well as sequences with no 16S BLAST hit and a low confidence OTU assignment (<10%) from QIIME2 ($n = 887$) were considered host gDNA contamination[100]. No sequences had hits in both the eukaryote and prokaryote databases. Technical contaminants ($n = 173$) were identified by sequencing batch and sample source (brain tissue, fecal) using the decontam package v1.22.0[28]. The prevalence and frequency methods were used for fecal samples and the prevalence method was used for brain tissue samples to identify likely contaminants with $p < 0.1$. Read counts for the remaining ASVs ($n = 4524$) were used for downstream analyses.

Microbiome data was managed using the microbiome package v1.20.0[101]. Genera detected in the implanted brain samples but not the unimplanted brain samples were classified as potentially invasive microbes with distinct, implantation-associated features. Samples were repeatedly rarified 500 times to the minimum read depth of 187

reads, and these datasets were used for both alpha and beta diversity analyses[29]. The Shannon Diversity Index and total observed features were calculated for each set of rarefied samples and differences in these metrics and in implantation-associated feature abundance by implantation status and treatment group were assessed via two-way ANOVA. The Benjamini-Hochberg method was used to adjust $p$-values for multiple testing of pairwise contrasts using Tukey's Honest Significant Differences.

An unrooted phylogenetic tree was produced by aligning representative sequences with the DECIPHER package v2.26.0 using the default parameters and de novo assembly with the phangorn package v2.11.1[102–104]. The best nucleotide substitution model, the transition model TIM1 + G(4) + I, was selected using the lowest Bayesian Information Criterion among available models and an unrooted tree inferred and optimized via maximum likelihood. Unweighted UniFrac distances were calculated for each rarified dataset using the phylogenetic tree. Samples were ordinated via principal components analysis and differences in the distances by implantation status assessed by PERMANOVA with the vegan package v2.6.4 for each treatment group[105,106].

Linear discriminant analysis Effect Size (LEfSe) was used to determine OTUs enriched in brain samples from each implantation status using the microbiomeMarker package v1.4.0, with taxa having a linear discriminant analysis score greater than 4.5 being considered enriched[30,107]. The differential abundance of OTUs agglomerated at the phylum through genus levels in antibiotic-treated fecal samples versus baseline, pre-treatment fecal samples after controlling for sequencing batch was assessed with ANCOMBC2 from the ANCOMBC package v2.0.3[108,109]. The Benjamini-Hochberg method was used to adjust $p$-values of pairwise comparisons for multiple testing.

All the analyses were performed in R 4.3.3 in Windows 10 ×64[110].

## Spatial proteomic analysis of the implant site

Since not all implanted brains were utilized for proteomics, a random selection of 3–4 brains were taken from each group. Frozen, non-fixed brains were first sectioned at 5 μm thickness using a cryostat and mounted onto microscope slides (SuperFrost Plus, FisherBrand, Hampton, NH). One section from the middle depth of the implant (~500 μm deep into the cortex) for each brain was taken and sectioned onto each slide. Doing so yielded slides containing one brain slice from each animal in the study. Once slides were prepared containing each brain slice, spatial proteomic analysis was done using the NanoString GeoMx and nCounter suite of equipment and reagents and following their established protocols (NanoString Technologies, Seattle, WA, USA) (Fig. 6E). For proteomic analysis, slides were first submerged in 10% neutral buffered formalin (NBF, Thermo Fisher Scientific, Waltham, MA, USA) for 12–16 h followed by 3x washes in 1x Tris-Buffered Saline with Triton (1x TBS-T, NanoString). Briefly, slides then undergo antigen retrieval with 1x Citrate buffer using the TintoRetriever Pressure Cooker (Bio SB, Item Number: BSB 7008) on high temperature and pressure settings for 15 min followed by blocking tissue for non-specific reaction to antibodies. Morphological antibodies for neurons (1:100 anti-NeuN, Alexa Fluor® 647 EPR12763, Item Number: ab190565) and astrocytes (1:40 anti-GFAP, Alexa Fluor® 532 GA-5, Item Number: NBP2-33184AF532) were then incubated in a humidity chamber overnight in a 4 °C refrigerator along with antibodies specific to NanoString mouse neural proteomics panel at 1:25 concentration (Table 1). The mouse neural proteomics panel consists of the Neural Cell Profiling Core (25 proteins, Item Number: 121300120) paired with the Glial Cell Subtyping Module (10 proteins, Item Number: 121300125) and the Autophagy Module (10 proteins, Item Number: 121300124). These modules contain antibodies that are bound by photocleavable cDNA sequences unique to each protein of interest. Following overnight primary antibody incubation, tissue was washed three times in 1x TBST for ten minutes each then postfixed with formalin for 30 min. Residual

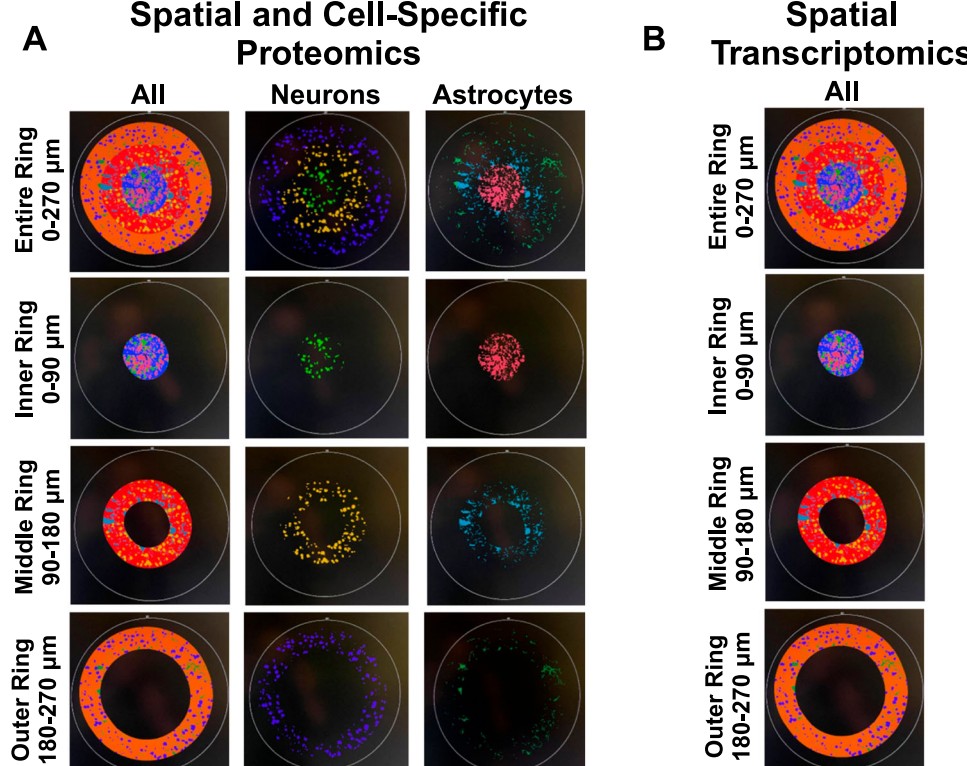

**Fig. 7 | Spatial and Cell-Specific Analysis of the Implant Site Using Proteomics and Transcriptomics. A** Proteomics analysis was performed on the entire implant ring (0–270 μm from the implant site), inner ring (0–90 μm from the implant site), middle ring (90–180 μm from the implant site), and outer ring (180–270 μm from the implant site) on a cell-specific basis for neurons, astrocytes, and all cells. **B** Transcriptomic analysis was not done using cell-specificity. Only spatial separation to analyze the implant regions was performed. Created in BioRender. Capadona, J. (2025) https://BioRender.com/p93j360.

formalin was washed off twice in 1x TBST for five minutes each before being stained with a nuclear stain (1:10 Syto13, NanoString Technologies, #121300303) before imaging. Using the NanoString GeoMx, tissue was imaged, and the implant site was identified. Regions of interest were then selected for protein extraction. Here, we extracted proteins from areas stained by either NeuN, GFAP or from the entire AOI in three regions: 0–90 μm from the implant site (inner region), 90–180 μm from the implant (middle region), or 180–270 μm from the implant (outer region) (Fig. 7A). The cDNA from each unique protein is then collected on a region and cell-specific basis utilizing the UV-cleaving process of NanoStringto to separate the photocleavable cDNA from the antibody.

Once cDNA was collected into the 96-well plate, the plates were dried overnight in the GeoMx at room temperature before being rehydrated in DNAse/RNAse-free water. After, GeoMx Hybridization Codes (NanoString Technologies, Item Number: 121300401) were added to each row A-H to distinguish between each row and allow for pooling of samples. Each column was then pooled into a final collection of 12 pooled sample solutions that were then loaded into the nCounter MAX/FLEX system (NanoString) for barcode analysis to obtain protein expression counts.

Raw proteomic counts from the neural proteomic panel were uploaded and analyzed using a custom MATLAB R2021a script, following previously established protocols[37]. First, the negative and positive spike-in proteins were removed from analysis. From here, all protein counts were normalized to the geometric mean of the housekeeping proteins. Housekeeping proteins were used for normalization due to their prevalence in all samples and accounts for the number of cells and proteins across varying runs. Housekeeping proteins were not included in the differential expression comparisons. The $\log_2$(fold change) (log2FC) for each protein was calculated for

each comparison. After normalization, unpaired t-tests were performed across respective groups for comparison. Unadjusted $p$ values were corrected using the Benjamini-Hochberg false discovery rate method to account for random significance. Data was visualized with volcano plots using GraphPad Prism Plus. All proteomic volcano plots show the $-\log_{10}(p_{adjusted})$ plotted against the log2FC. A dotted line indicates the significance threshold, as determined using the adjusted $p$-values calculated.

### Spatial transcriptomic analysis of the implant site

Since not all implanted brains were utilized for transcriptomics, a random selection of 3–4 brains were taken from each group. Like the proteomics analysis, frozen, non-fixed brains were first sectioned at 5 μm thickness using a cryostat and mounted onto microscope slides (SuperFrost Plus, FisherBrand, Hampton, NH). One section from the middle depth of the implant (~500 μm deep into the cortex) for each brain was taken and sectioned onto each slide. Doing so yielded slides containing one brain slice from each animal in the study. Once slides were prepared containing each brain slice, spatial transcriptomic analysis was done using the NanoString GeoMx and reagents following their established protocols (NanoString Technologies, Seattle, WA, USA) (Fig. 6E). Slides were again fixed overnight in 10% NBF followed by 3x washes in 1x PBS and sequential washes in 50% ethanol, 70% ethanol, and 100% ethanol. From here, antigen retrieval was performed using 1x Tris-Ethylenediaminetetraacetic acid (EDTA) (NanoString) for 20 min at 99 °C. RNA targets were then exposed using Proteinase K (NanoString) at a concentration of 1 μg/mL for 15 min at 37 °C before undergoing a postfix to preserve tissue morphology using NBF Stop Buffer. An overnight in situ hybridization step at 37 °C within a humidity chamber then occurs to bind the RNA probe mix to RNA targets on tissue. The probe mix used here contains the Whole

Transcriptome Atlas (WTA) probes from NanoString (NanoString Technologies, Item Number 121401103) for mouse tissue utilizing NanoString's barcode identification technology. Following hybridization, tissue was washed with a mixture of 100% formamide (NanoString) and 4x Saline Sodium Citrate buffer (4x SSC, NanoString) in a 37 °C water bath to remove any off-target probes. Slides were then placed in a humidity chamber and covered with Buffer W (NanoString Technologies, Item Number 100474) at room temperature for 30 min. Morphology markers for GFAP (1:40 anti-GFAP, Alexa Fluor® 532 GA-5, Item Number: NBP2-33184AF532) and NeuN (1:100 anti-NeuN, Alexa Fluor® 647 EPR12763, Item Number: ab190565) were then added along with SYTO 13 (NanoString Technologies, Item Number 121300303) for visualizing the implant site during imaging. After incubation for one hour at room temperature, excess morphology markers were washed off twice using 2X SSC for five minutes each. Slides were then loaded into the GeoMx Digital Spatial Profiler for imaging, region selection, and sample collection. GFAP and NeuN were used to visualize where the implant was in the brain to determine where to position the concentric rings for ROI selection. Regions of interest of 0–90 μm (inner), 90–180 μm (middle), and 180–270 μm (outer) around the implant site were selected for extracting target RNA; there was no cell-specific transcriptomics performed (Fig. 7B). Briefly, the NanoString barcode identification technology consists of binding a target complementary sequence to the target RNA. Attached to the target complementary sequence is a photocleavable linker with a Digital Signal Profiler (DSP) barcode at the end that corresponds to a specific gene of the mouse transcriptome. The DSP barcode is composed of a sequence of oligonucleotides that are unique to each gene of the mouse transcriptome, according to NanoStrings library. The target RNA is not actually collected during the collection process; instead, the photocleavable linker is detached and the DSP barcode is collected that corresponds to the target RNA / gene sequence. Once collected, the oligonucleotide DSP barcode is sent to the Case Western Reserve University genomics core for sequencing using the Illumina NextSeq 550. After sequencing, FASTQ files were loaded into NanoString's NGS pipeline software to convert into DCC before processing using the GeoMx software suite. Going through the NanoString NGS pipeline allows for converting the oligonucleotide sequences from the DSP barcode into the corresponding mouse genes composing the entire transcriptome. It is important to point out that the whole transcriptome RNA of the mouse is not being collected and sequenced here. The oligonucleotides of the NanoString DSP barcodes are being sequenced.

For processing the transcriptomic data, technical and biological quality control was performed to remove any outlier genes and genes with minimal expression detected. Filtering was also performed to remove any genes that did not show expression in at least 5% of the analyzed segments. Quality control and filtering parsed the data down from 20175 genes to 8272 genes for analysis. For measuring the entire implant region (All AOI), the inner, middle, and outer regions were summed together on a per sample basis before proceeding to normalization. Each gene underwent Q3 normalization followed by statistical analysis using a custom MATLAB R2021a script to perform unpaired t-tests between samples. Unadjusted p-values were used for all further comparisons in the iPathways software suite. In iPathways, individual genes as well as hundreds of pathways and thousands of gene ontology (GO) terms are evaluated based on gene levels and significance. P-values for pathways and GO terms use uncorrected p values. Pathways and GO terms of interest were discussed based on their relevance to neurological processes and inflammation. It is important to note that iPathways includes all biological processes and does not filter by organ. Therefore, it was up to user discretion to determine which pathways and GO terms were relevant when discussing impacts on the brain and microbiome. Biological processes involving neurological response, ribosomal function, immune response, cellular function, and metabolism were of key interest and evaluated here, to name a few. However, as there are hundreds of pathways and thousands of GO terms available for analysis, not all relevant areas were able to be discussed. A full Excel file of genes, with log2FC and p-values, and all pathways and GO terms with associated genes and p-values is included in the Supplementary Information for reader accessibility and transparency. Volcano plots were created using GraphPad Prism Plus 10 and include the -$\log_{10}$(unadjusted p-value) plotted against the log2FC for each gene. The dotted line indicates significance.

### Replication

Our hypothesis was to test whether changing the gut microbiome can impact neural recording performance and brain health after implantation. We performed conceptual replication by testing this hypothesis in four different experiments: 16S evaluation of the brain after implantation and treatment, neural recording performance, proteomic evaluation around the implant site, and transcriptomic evaluation around the implant site. There was no direct replication or systematic replication performed in this study. However, future studies will be utilized to perform both direct and systematic replication of these results.

### Ethics

Every experiment involving animals, human participants, or clinical samples have been carried out following a protocol approved by an ethical commission. Each participant gave informed written consent. All contributors to this study who met the authorship criteria required by Nature Portfolio journals AND Case Western Reserve University have been listed as authors, given that their involvement was crucial for both the study's design and execution. No implicit or explicit biases were used when considering collaborators and contributors for this project. The team is diverse and represents the local experts related to each stage of this project. The human fecal data collected was ethically collected from a single subject from a separate study focused on brain-controlled movement and sensory restoration neuroprostheses for individuals living with chronic paralysis. The FDA approved Investigational Device Exemption (IDE) is focused on the efficacy and safety of the regulated implanted devices. The fecal sample collection is an added procedure to the protocol specifically intended, as articulated in the manuscript, to understand the gut microbiome concentration in a chronically implanted study participant. Collection of the fecal sample in no way interferes with the ongoing clinical trial. The study participant signed a consent form, which was approved by both federal regulatory bodies and the local institutional review board and the clinical trial can be found under #NCT03898804.

### Reporting summary

Further information on research design is available in the Nature Portfolio Reporting Summary linked to this article.

## Data availability

All data supporting the findings of this study are available within the article and its supplementary files. Any additional requests for information can be directed to, and will be fulfilled by, the corresponding authors. Source data are provided with this paper. The 16S, recording, proteomics, and transcriptomics data generated in this study have been made freely and publicly available in the OSF database under accession code https://doi.org/10.17605/OSF.IO/JKG27. Source data are provided with this paper.

## Code availability

All custom codes are available within the study's OSF repository under accession code https://doi.org/10.17605/OSF.IO/JKG27.

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

## Acknowledgements

This study was supported in part by the Merit Review Award GRANT12418820 (JRC) and Senior Research Career Scientist Award # GRANT12635707 (JRC) from the United States (US) Department of Veterans Affairs Rehabilitation Research and Development Service. Additionally, this work was also supported in part by the National Institute of Health, National Institute of Neurological Disorders and Stroke GRANT12635723 (JRC) and diversity supplement AGH) and NS131502 (JRC), the Congressionally Directed Medical Research Program (CDMRP) – Spinal Cord Injury Research Program (SCIRP), administered through the Department of Defense Award # SC180308 (ABA) and the National Institute for Biomedical Imaging and Bioengineering, T32EB004314, provided support for both GFH and GB (JRC). Microbiome analyses were partially supported by the junior faculty's startup funding from the CWRU School of Medicine, BGT630267 (LZ). Finally, partial funding was provided from discretionary funding from the Donnell Institute Professorship endowment (JRC) and the Case School of Engineering Research Incentive Program (JRC). The authors would also like to acknowledge Professors Joseph Pancrazio and Stuart Cogan from the University of Texas at Dallas for scientific conversations related to the direction and scope of the project.

## Author contributions

J.R.C. and H.A.vR. contributed to the conception and design of the work. G.F.H., S.E.G., L.N.D., J.L.D., G.B., G.R.W., A.H.L., C.H., M.B., H.O., T.B., J.C., L.L., C.D., G.J., H.A.H., A.P.A., A.B.A., A.G.H., A.J.S., L.Z., and J.R.C. contributed to the methodology, software analysis, validation, formal analysis, investigation, and data curation. W.M., J.S., and A.B.A. provided access to human subjects for fecal sample acquisition. E.R.C. and L.Z. guided the statistical analysis of the work. G.F.H., S.E.G., and J.R.C. wrote the original draft. G.F.H., S.E.G., and J.R.C. prepared figures. All authors edited the final manuscript. J.R.C. provided overall project management and the funding and resources to conduct the study.

## Competing interests

The contents do not represent the views of the U.S. Department of Veterans Affairs, the National Institutes of Health, or the United States Government. The authors declare no conflict of interest.

## Additional information

George F. Hoeferlin ®[1,2], Sarah E. Grabinski[3], Lindsey N. Druschel ®[1,2], Jonathan L. Duncan[1,2], Grace Burkhart ®[1], Gwendolyn R. Weagraff ®[2,4], Alice H. Lee[1,2], Christopher Hong[1,2], Meera Bambroo[1,2], Hannah Olivares[1,2], Tejas Bajwa[1,2], Jennifer Coleman ®[1], Longshun Li ®[1,2], William Memberg ®[1,5], Jennifer Sweet[5,6], Hoda Amani Hamedani ®[2,7], Abhinav P. Acharya ®[1], Ana G. Hernandez-Reynoso[1,8], Curtis Donskey[5,9], George Jaskiw ®[5,10], E. Ricky Chan ®[11], Andrew J. Shoffstall ®[1,2], A. Bolu Ajiboye[1,5], Horst A. von Recum ®[1,2], Liangliang Zhang ®[3,12] ✉ & Jeffrey R. Capadona ®[1,2] ✉

[1]Department of Biomedical Engineering, Case Western Reserve University, Cleveland, OH, USA. [2]Advanced Platform Technology Center, Louis Stokes Cleveland Department of Veterans Affairs Medical Center, Cleveland, OH, USA. [3]Department of Population and Quantitative Health Sciences, Case Western Reserve University, Cleveland, OH, USA. [4]Department of Biology, University of Florida, Gainesville, FL, USA. [5]Louis Stokes Cleveland Department of Veterans Affairs Medical Center, Cleveland, OH, USA. [6]Department of Neurological Surgery, University Hospitals Case Medical Center, Cleveland, OH, USA. [7]Department of Materials Science and Engineering, Case Western Reserve University, Cleveland, OH, USA. [8]Department of Bioengineering, The University of Texas at Dallas, Richardson, TX, USA. [9]Division of Infectious Diseases & HIV Medicine in the Department of Medicine, Case Western Reserve University School of Medicine, Cleveland, OH, USA. [10]Department of Psychiatry, Case Western Reserve University, Cleveland, OH, USA. [11]Cleveland Institute for Computational Biology, Case Western Reserve University, Cleveland, OH, USA. [12]Case Comprehensive Cancer Center, Case Western Reserve University, Cleveland, OH, USA. ✉e-mail: lxz716@case.edu; jrc35@case.edu

