## [Peer Review file · Nature Communications]

Bacteria Invade the Brain Following Intracortical Microelectrode Implantation, Inducing Gut-Brain Axis Disruption and Contributing to Reduced Microelectrode Performance

Corresponding Author: Professor Jeffrey Capadona

Version 0:

Reviewer comments:

Reviewer #1

(Remarks to the Author)

In this work entitled "Bacteria Invade the Brain Following Sterile Intracortical Microelectrode Implantation" by George F. Hoefler et al. the authors addressed the impact of microelectrode implantation in the brain on brain microbiome and the microbiome-gut-brain axis. The authors used state of the art strategy: intracortical microelectrodes recordings, spatial proteomics and spatial transcriptomics to follow neuroinflammation and 16S sequencing to identified bacteria phyla. They concluded that intracortical microelectrode promote brain colonization – or at least brain modification of the microbiome – associated with lower quality and stability of electrode performance.

Although the question is interesting, I have serious doubts about the methodological approach. Many results are over-interpreted and sometimes the literature is misinterpreted or over-interpreted with the purpose to give strength a somewhat weak level of evidence, which sometimes borders on the unethical. I'm not convinced at this stage by this work, which is too preliminary to be recommended for publication.

- Abstract: "resulted in differential expression of bacteria in the brain" ; what do you mean ?
- Abstract: "Fecal microbiome composition was similar between implanted mice and an implanted human", this is not a result from this work. Please remove.
- Page 2: "Degradation of blood-brain barrier (BBB) integrity is an appreciable consequence of microelectrode-mediated neuroinflammation and increases the entry of blood-borne components into the brain, where they could amplify and extend the neuroinflammatory response^{3,4,6,15-18}". None of this reference are related to microelectrode, and poorly to neuroinflammation.
- "The integrity of the mucosal lining of the intestines is dysfunctional following brain injury¹⁹". This reference is related to Traumatic Brain Injury (TBI). Quite often authors make the link between TBI and microelectrode implantation, which is not the same thing. This should be rephrased and/or explained with evidences that implantation and TBI are related.
- "Despite such links, there are no reports of how brain responses to gut microbiome infiltration following microelectrode implantation". This sentence is an assumption that infiltration following microelectrode implantation has been demonstrated. There is no evidence for this, please rephrase.
- Page 3: "Our results suggest that modulating the invasion of microbes into the brain may impact microelectrode performance to improve quality and stability". Over-interpretation, please rephrase.
- Page 3, figure 1:
While I do not doubt in bacteria contamination of the brain after electrode implantation I have several concern for the Brain-biota hypothesis since there is very few studies searching for bacteria in the brain. Here I am not convinced either. Since data are normalized in fig1, I would like to have access to coverage and exact number of reads. This should be dramatically different between unimplanted and 4 weeks after implantation. Moreover, the background of this experiment was not taken into account. What is noise and what is results (which means that probably you can't distinguish between the noise and the signal before implantation).
A minor concern is the absence of a paragraph in the introduction discussing the microbiota of the healthy brain.
- Page 5 : The use of this mix of antibiotic is not clear. Page 5 the authors claimed that "mice was treated with antibiotics to

deplete fecal microbiota". In the mat and methods the authors claimed that "Such antibiotics were chosen based off previous literature to provide broad spectrum capacity and effect on the gut microbiome"

In fact, Ampicillin, Clindamycin, and Streptomycin clearly promote dysbiosis (for instance : <https://molecularbrain.biomedcentral.com/articles/10.1186/s13041-021-00759-w>) causing inflammation and some brain dysfunction. Therefore, my question is: What was the purpose of using this mix ? It's not clearly mentioned in the manuscript. My guess is that the use of this mix is likely to cause biases that do not seem to have been anticipated by the authors. And given that this mix induces dysbiosis, what conclusion can we draw from this experiment?

- Page 6/7 what is the meaning of having 55.8% of reads of unknown origin ? Are these reads not linked to any phyla? Or is the anatomical origin of these phyla unknown ? Please rephrase and give information on these reads.

- Page 9, fig2 should be fig3

- Figure 3: I do not understand the use of Acute and chronic for implantation of electrodes.

- Figure 3 C-F: What is the relevance of such a difference?

- Page 11: "Across all comparisons, 28 of the 39 possible proteins" do you mean 39 tested proteins ? Why these 39 proteins have been chosen ? Are they related to inflammation ? For instance there is marker of M2 macrophages but not that of M1 macrophage. Therefore, the authors cannot conclude on this point (page 12).

- Discussion,

- Page 17: "Therefore, the principal hypothesis of the current study was that damage to the BBB caused by microelectrode implantation would amplify dysregulation of the microbiome-gut-brain axis". This is far out of the results of this work.

- "Further, several studies in rodents and humans have shown that traumatic brain injuries are accompanied by increased intestinal permeability and intestinal barrier dysfunction^{19,47,48}. Therefore, we were particularly interested in understanding the role that microbes that reside in the intestines may have on microelectrode performance if they invade the brain tissue following microelectrode implantation" Comparison of TBI and electrode implantation should not be used.

- "we have demonstrated that microbes associated with the gut microbiome invade the brain tissue". I fully disagree.

- "The differential composition and abundance of invasive microbes were associated with significant temporal changes in the recording performance of intracortical microelectrodes". I am not convinced. There is not statistical evidence for this.

- "It is important to recognize that even subtle changes in the microbe composition in the brain have been linked to changes in brain health^{20,22,23} ." Ref are inappropriate.

Ref 20: quote: "Here I argue that the evidence for the presence of microbes in diseased brains is quite strong, but a compelling demonstration of resident microbes in the healthy human brain remains to be done"

Ref 22 talk about gut microbiota that have an effect on the BBB

Ref 23 talk about microglia inflammation, not brain microbes

- Conclusion:

- "Further, it is possible to modulate the neuroinflammatory response following implantation and microelectrode performance by altering the composition and abundance of invasive microbes". This is overinterpretation.

- "Some of the microbial strains identified in brain tissue in this initial study have been previously associated with neurodegenerative symptoms and diseases". I doubt that this study has identified the strains.

Reviewer #2

(Remarks to the Author)

This study makes a convincing case that intracortical microelectrode implantation alters the distribution of microbial sequences in the brain, and antibiotic treatment can influence the microbial sequence changes induced by electrode implantation. With the growing use of implants in people, these findings have clear clinical relevance. However, there are some technical issues:

1). There is an ongoing concern in the field that bacterial sequences recovered from mouse (or human) tissue are artifactual (e.g., introduced during library preparation or sequencing). The alteration of the bacterial population after implantation makes a good case that the bacterial sequences are "real", particularly if all samples were processed in parallel. It is therefore important for the authors to give more details about sequencing library construction and whether all 16S reads were sequences in the same Illumina run.

2). An important distinction is whether bacterial sequences found in brain tissue are actually in the brain parenchyma, or in the associated blood vessels. (There is abundant evidence for microbial sequences in human blood.) Although the investigators perfused their mice before tissue sampling, it is unclear if this completely depletes bacterial sequences that reside in the blood. Do the authors have any evidence that the detected bacterial sequences are actually in the brain parenchyma?

3) The spatial transcriptomics is not well described, presented, or analyzed. This reviewer was confused about whether the differentially-expressed genes identified are solely between implanted animals with and without antibiotics, or are being compared to unimplanted mice. If it is the former, the investigators cannot use the data to follow the trajectory of inflammation caused by the implantation. This is unfortunate, because it makes the interpretation of +/- antibiotics more difficult (e.g., it is not known what gene changes occur just because the animals have gotten older). In terms of presentation, it is incumbent on the authors to present all the differential gene changes in a supplementary table, so readers can do their own interpretation. In terms of analysis, I would have expected a typical gene ontology analysis, with P values for the enriched categories, rather than the cherry-picked categories presented. Glaringly missing is any indication of changes in gene categories associated with immune response or inflammatory genes. If these genes do not change, then this should be explicitly pointed out by the authors.

4). The authors likely have a valuable data source that they should tap. There is a high probability that there are microbial sequences (particularly bacterial rRNA reads) in the RNA-seq data they generated. (How many depend on how whether their RNA-seq libraries were generated from total or polyA-selected RNA, which they need to detail in their Methods). There are multiple pipelines available (e.g., <https://pubmed.ncbi.nlm.nih.gov/33914880/>) to identify microbial sequences in human or mouse RNA-seq data. Identifying these reads would enable the investigators to confirm their 16S data as well as identify novel microbes. In theory, they might be able to associate specific microbes with specific regions of interest, or specific cell types.

Minor points:

1) line 94: presumably the investigators did not extract bacterial DNA from biopsy punches; they extracted total DNA from which they amplified 16S sequences.

2). Results text: It would be helpful give the N of mice used for the various conditions, rather than forcing the readers to count this up in the Figures.

3) An important result of this study is that the majority of microbial sequences in the implant brain do not appear to come from the gut. Potential alternative sources could be the oral microbiome or the nasal cavity. Perhaps the authors should discuss the possible sources a little bit more and describe follow-up approaches.

Version 1:

Reviewer comments:

Reviewer #1

(Remarks to the Author)

The authors have significantly modified their manuscript, So I've taken my analysis from the start. However, you will first find below a response to the authors for two remaining concerns.

“3) Page 2: “Degradation of blood-brain barrier (BBB) integrity is an appreciable consequence of microelectrode-mediated neuroinflammation and increases the entry of blood-borne components into the brain, where they could amplify and extend the neuroinflammatory response^{3,4,6,15-18}”. None of this reference are related to microelectrode, and poorly to neuroinflammation.

We disagree with the reviewer’s interpretation of the literature citations we used. Below, we show the full reference and call out specific examples of text from each reference, which demonstrate why it is an appropriate citation for the text. No changes to the manuscript were made in response to this reviewer comment.”

I agree that my comment was not clear. However, the sentence is still strongly misleading. References did not support that microelectrodes increase the entry of blood-borne components into the brain, where they could amplify and extend the neuroinflammatory response

Ref 3 is dedicated to microelectrode, failure is used for “recording failure” : “Most failures (56%) occurred within a year of implantation, with acute mechanical failures the most common class (48%), largely because of connector issues (83%). Among grossly observable biological failures (24%), a progressive meningeal reaction that separated the array from the parenchyma was most prevalent (14.5%).”

Ref 4 is a review that only acknowledge the role of microelectrode in the maintenance of inflammation with maybe an impact on the BBB.

Same for Ref 6, same for ref 15 (DAMPs are not associated to bacteria in this paper), same for reference 16-18

“Comment 8) Page 5 “

It's still not clear to me whether the authors took into account, in their analysis, the fact that this antibiotic cocktail can cause dysbiosis resulting in neuroinflammation.

“Comment # of reviewer 2”

While I am convinced now that there were no blood contamination in the collected brain, I am not convinced by Reviewer Response Figure 2 and 3. There is so many bacteria detected in the sham control. If there were so many bacteria, they would undoubtedly be detected on agar and the 16S would be very clear in the event count.

Review, September 2024

The authors have greatly modified their manuscripts and clarified many points. The results obtained on inflammation seem solid to me. However, I am not at all convinced by the hypothesis of a microbiota in the naive brain.

I have noted in this work that the authors have also called this hypothesis into question or presented data that call it into question.

Line 169 : “These ratios suggest that microbial biomass is low or non-existent in the unimplanted brain.”

Line 651 : “16S measurement does not confirm the presence of live bacteria and may indicate either dead or fragments of 651 bacterial DNA.”

The impact of antibiotics on the results of 16S sequencing in the brain suggests that these results are background noise. In my opinion, it is important, in order to avoid any misinterpretation by the community, to clearly state that in the current state of the results, the 16S results obtained in the brain before implantation or in the presence of antibiotics cannot be differentiated from the background noise. I suggest that authors rewrite the manuscript taking into account this general comment. With this in mind, here are my comments on this point (This list is not exhaustive):

- Line 185. You cannot compare the “composition of microbes present (...) in the brain tissue of naive unimplanted mice”. This suggests that the composition of a brain microbiota was described in this manuscript. I suggest to rephrase the sentence. You can compare results of 16S sequencing but not microbes.
 - line 197. You cannot conclude that “45 genera are detected in the unimplanted brain”
 - Concerning the paragraph line 273/280 and the sentence : “we would not expect them to vary in response to antibiotic-treatment of the animal”. I disagree. If there is any contamination from bacteria living in mice, this contamination should be different after antibiotic treatment, because the whole bacteria population is different. Therefore, the 16S amplicon background arising from this “whole microbiota” will differ between treated and non-treated mice. This argument that 16S results in naive brains are not artefactual doesn't stack up in my opinion and should be removed.
- Moreover, line 278, you cannot mention the possibility of a naïve brain microbiota if we consider your conclusion line 169.
- Line 556/558, thank you for specifying that this concerns the post-implantation brain.
 - Line 584: “in microbiota composition”. Do you mean microbiota in the gut ? If yes please add "gut" before microbiota.
 - Finally, could you exclude the possibility that immune cells that have collected bacteria debris may be present in the brain or its surrounding ? As for the trojan horse hypothesis proposed for listeria monocytogenes ?

Other concerns:

- Ampicillin, Clindamycin, and Streptomycin induce a modification of the gut microbiota and this may have an impact on brain inflammation. It might be useful to clarify this.
- I do not understand the link between line 604 and 605.

Minor concerns:

- Line 31 and 68, what are « constituents » ? Or “components”, line 58. Do you mean PAMPs ?
- Line 63/64: “Bacteria entry through the damaged BBB is likely the dominant entry point for bacteria residing in or on the host”. Why not clearly state that bacteria pass through the bloodstream? What do you mean by dominant? What other routes do the authors consider that do not involve passage through the bloodstream?
- Line 108: “in-tact”. Probably “intact” ?
- Figure 1C, it seems that the y axis is wrong, 1000 should have been 100

Reviewer #2

(Remarks to the Author)

The authors have done an admirable job addressing my initial concerns, and the manuscript has been significantly improved. Now that the full differential gene expression data has been included, I note that the most obvious differential gene expression in the 12 month antibiotic-treated mice is a global upregulation of olfactory receptors. This is curious for multiple reasons, and it would be appropriate for the authors to comment on this (as anyone who looks at the DEG data will be similarly curious).

Reviewer #3

(Remarks to the Author)

Great work with pretty impressive data.

Pretty difficult to intervene at this stage of the review process, but reviewer 1 is 100% right – there is no data in this manuscript that firmly show bacteria in the naive brain. The post-surgery increase is, to me, what really matters and what this manuscript is about. So I suggest that the authors make 100% clear that the presence of brain-associated bacteria remains unknown and challenging to study.

Version 2:

Reviewer comments:

Reviewer #1

(Remarks to the Author)

The authors modified their manuscript extensively throughout the review process.

The result is very satisfactory and I congratulate the authors for having taken the reviewers' suggestions into account in a constructive way.

Reviewer #3

(Remarks to the Author)

A point-by-point response to reviews:

We thank the reviewers for their time and feedback. Our additional analysis performed as a result of the questions and suggested experiments greatly improved the resolution of the data and our ability to support our conclusions – thank you!

The reviewer's comments are shown in **BOLD** font below. Our response to the comment is shown in “normal font.” We have made all requested changes to the manuscript, which are noted both below and in the manuscript with **BLUE FONT**. Text deleted from the manuscript has not been indicated. Minor grammatical changes made during our additional reads have also not been noted here.

Please note that we have updated the title to better reflect the updated data and interpretation of the data. Additionally, three new authors were included in the revision because of their contributions to the histology included in the response to reviewer #2.

Additional changes that were not in direct response but tangential responses to the requested edits/clarification are listed at the end of this document as “**Additional changes.**”

Reviewer #1 (Remarks to the Author):

In this work entitled “Bacteria Invade the Brain Following Sterile Intracortical Microelectrode Implantation” by George F. Hoferlin et al. the authors addressed the impact of microelectrode implantation in the brain on brain microbiome and the microbiome-gut-brain axis. The authors used state of the art strategy: intracortical microelectrodes recordings, spatial proteomics and spatial transcriptomics to follow neuroinflammation and 16S sequencing to identified bacteria phyla. They concluded that intracortical microelectrode promote brain colonization – or at least brain modification of the microbiome – associated with lower quality and stability of electrode performance.

Although the question is interesting, I have serious doubts about the methodological approach. Many results are over-interpreted and sometimes the literature is misinterpreted or over-interpreted with the purpose to give strength a somewhat weak level of evidence, which sometimes borders on the unethical. I'm not convinced at this stage by this work, which is too preliminary to be recommended for publication.

We thank the reviewer for their transparent and blunt opinion. However, we respectfully disagree with their interpretation of our methodological approach, misinterpretations of results and literature, and their perceived “unethical” intentions. We will address specific comments below. Additionally, the revised analysis has strengthened the statistical significance of our findings and the strength of our conclusions.

1) Abstract: “resulted in differential expression of bacteria in the brain”; what do you mean?

Line 33: Changed to “**Systemic antibiotic treatment of mice implanted with microelectrodes to suppress gut bacteria resulted in decreased abundance of invading microbes in the brain tissue...**” to be more specific.

2) Abstract: “Fecal microbiome composition was similar between implanted mice and an implanted human”, this is not a result from this work. Please remove.

This result was moved from the manuscript to the supplemental on final read. We regretfully forgot to remove the statement from the abstract. *It has been removed.*

3) Page 2: “Degradation of blood-brain barrier (BBB) integrity is an appreciable consequence of microelectrode-mediated neuroinflammation and increases the entry of blood-borne components into the brain, where they could amplify and extend the neuroinflammatory response^{3,4,6,15-18}”. None of this reference are related to microelectrode, and poorly to neuroinflammation.

We disagree with the reviewer's interpretation of the literature citations we used. Below, we show the full reference and call out specific examples of text from each reference, which demonstrate why it is an appropriate citation for the text.

No changes to the manuscript were made in response to this reviewer comment.

Reference 3: Barrese, J. C. et al. Failure mode analysis of silicon-based intracortical microelectrode arrays in non-human primates. *J Neural Eng* 10, 066014 (2013). <https://doi.org:10.1088/1741-2560/10/6/066014>

- This is a seminal paper in intracortical microelectrode research and theorizes that the biological response is a crucial factor in failure.
- Section 4.2: “The mean time to failure of all 62 failed implants was 332 days (median = 133)”
- Section 4.4.1: “There were six acute biological failures (9.7%) with a mean time to failure of 36 days (median = 30).”
- Section 4.5.1: “There were nine (14.5%) chronic biological failures with a mean time to failure of 160 days (median = 163).”

Reference 4: Jorfi, M., Skousen, J. L., Weder, C. & Capadona, J. R. Progress towards biocompatible intracortical microelectrodes for neural interfacing applications. *J Neural Eng* 12, 011001 (2015). <https://doi.org:10.1088/1741-2560/12/1/011001>

- Section 3.4: “There is increasing consensus that the neuro-inflammatory response to intracortical microelectrodes is a primary hurdle preventing microelectrode-driven BMIs from reaching their full potential. Therefore, improving the understanding of the neuro-inflammatory response that develops following microelectrode implantation in the brain, and developing strategies to reduce its impact are critical to achieving the promise of BMIs and to enable longer recording durations for basic science experiments.”
- Section 3.4.4: “Multiple studies have shown that activated microglia and macrophages release a plethora of proinflammatory/cytotoxic soluble factors that can damage healthy bystander cells and the surrounding tissue [120124].”
- “Of the plethora of soluble factors within a macrophage’s available palette, previous work from Biran et al has shown that adherent cells retrieved from explanted devices secrete both tumor necrosis factor-alpha (TNF- α) and monocyte chemoattractant protein-1 (MCP-1) [107]. TNF- α can have direct toxic effects on neurons and oligodendrocytes, while MCP-1 is a chemokine involved in opening the blood-brain barrier (BBB) and recruiting new macrophages to sites of injury and inflammation [120, 121, 124, 128–134].”
- 3.4.8: “As observed in many neurodegenerative disorders, it was found that local blood BBB integrity is compromised in the tissue immediately surrounding implanted microwires and Michigan-style microelectrodes [78, 117].”
- “Additional recent data further highlights the potential role of BBB dysfunction in connection with poor recording performance. Findings from the Bellamkonda group with Michigan-style and microwire electrodes [162] have shown that recording performance correlates with markers of BBB dysfunction such as extravasated immunoglobulin G (IgG) or labeled albumin.”

Reference 6: Ravikumar, M. et al. The Roles of Blood-derived Macrophages and Resident Microglia in the Neuroinflammatory Response to Implanted Intracortical Microelectrodes. *Biomaterials* S0142-9612, 8049-8064 (2014).

- Section 3.3: “To date, most approaches evaluating the neuroinflammatory response have focused on analyzing neuronal densities, total population of microglia/macrophage (IBA1+ immunoreactivity), activated population of microglia/macrophage (CD68+ immunoreactivity), astrocyte population (GFAP+ immunoreactivity), and blood–brain barrier integrity (IgG+ immunoreactivity) [29,30,33–37].”
- Section 3.5.2: “We observed the highest levels of CD68+ immunoreactivity at two weeks post implantation, with significantly lower CD68+ immunoreactivity at chronic time points (four, eight, and sixteen weeks) (Fig. 10A–E).”

Reference 15: Bedell, H. W. et al. Understanding the Effects of Both CD14-Mediated Innate Immunity and Device/Tissue Mechanical Mismatch in the Neuroinflammatory Response to Intracortical Microelectrodes. *Front Neurosci* 12, 772 (2018). <https://doi.org:10.3389/fnins.2018.00772>. eCollection 2018.

- Section Titled “BBB Disruption”: “Similar to the glial scar and activated microglia/macrophages, BBB disruption (IgG expression) was found to be greatest at the probe-tissue interface and decreased in intensity as distance from probe-tissue interface increased (Figure 5).”
- Discussion: “The current study explores how a softening thiol-ene probe and/or targeting the TLR/CD14 innate immune pathway affects inflammation and neuronal density at both 2 and 16 weeks post-implantation of intracortical microelectrode probes”
- “The current study further highlights the importance of CD14 for microglial and macrophage responses to DAMPs. As previously reported, CD14 is central for microglial responses to damage signals in the brain (Janova et al., 2016). In this current study, Cd14-/- resulted in higher neuronal density (Figures 2A,C) and decreased glial scar (Figures 3A,C) at 2 weeks post-implantation, further revealing the importance of CD14 as a molecular target to reduce neuroinflammation.”

Reference 16: Saxena, T. et al. The impact of chronic blood-brain barrier breach on intracortical electrode function. *Biomaterials* 34, 4703-4713 (2013). <https://doi.org/10.1016/j.biomaterials.2013.03.007> S0142-9612(13)00289-5 [pii]

- Section 3.1: “In order to non-invasively assess the state of the BBB around intracortical electrode implants, animals were administered equal intravenous doses (15 nmoles) of Albumin-Cy7 and accumulation of the fluorophore around electrode implant sites in the barrel cortex was quantified using fluorescence molecular tomography (FMT). Albumin extravasation is a well-established indicator of compromised BBB in a variety of pathophysiological conditions [14], [15], [16], [17].”
- “However, at 16WPI (Fig. 1b,d, Fig. S1, Movie S1), significant differences were observed between implant groups. The Michigan electrodes exhibited a significantly higher breach than the microwire electrodes ($p < 0.01$). FMT imaging-based BBB quantification was validated by immunohistochemical staining of tissue sections with antibodies against the rat serum proteins albumin and IgG. The fluorescence intensity, as a function of distance from shank sites, was quantified and integrated [9], [12]. Immunoreactivity for the rat serum proteins, albumin (Fig. 2a,b) and IgG (Fig. 2c) was seen in parenchyma at or around the injured sites of all electrode-implanted animals. Immunofluorescence quantification confirmed FMT results and significant differences for both albumin (Fig. 2d,e) and IgG (Fig. 2f) extravasation were observed, wherein Michigan electrodes exhibited a significantly higher breach in comparison to microwire electrodes ($p < 0.01$) at 16WPI.”
- Section 3.2: “Results from the electrophysiological evoked recordings showed that the Michigan electrodes, which also exhibited a significantly higher BBB breach, failed to record within a period of 10 days (Fig. 3b) when compared to microwire electrodes, which provided stable recordings for a period of up to 84 days (Fig. 3b). The microwire electrodes also induced a significantly lower breach of the BBB.”

Reference 17: Wellman, S. M., Li, L., Yaxiaer, Y., McNamara, I. & Kozai, T. D. Y. Revealing Spatial and Temporal Patterns of Cell Death, Glial Proliferation, and Blood-Brain Barrier Dysfunction Around Implanted Intracortical Neural Interfaces. *Front Neurosci* 13, 493 (2019). <https://doi.org/10.3389/fnins.2019.00493>.

- From Section Titled “Pattern of Glial Cell Immunoreactivity and Proliferation Around Implanted Microelectrode Arrays”:
 - “Glial reactivity to implanted intracortical devices was evaluated by staining for histological markers specific for microglia (Iba-1), astrocytes (GFAP), and NG2 glia (NG2) at 1, 3, 7, and 28 days post-insertion (Figure 3A). Iba-1 immunoreactivity was highest proximal to the implant for all time points, decreasing steadily to normalized control levels with distance from the probe hole (Figure 3B). When observing directly at the 50 μm region at tissue-device interface, Iba-1 expression increased steadily over time; however, there was no significant difference in fluorescence intensity ($p < 0.05$) (Figure 3C). Beginning 3 days after implantation, GFAP fluorescence intensity increased 2–4 fold with proximity to the probe hole showing preferential expression around the site of device implantation (Figure 3D). GFAP expression was minimal at 1 day post-insertion and the average GFAP fluorescence was significantly increased within the 50 μm region surrounding the device at 3, 7, and 28 days post-implantation ($p < 0.05$) (Figure 3E and Supplementary Table S6). GFAP intensities reached a maximum at 7 days, and was significantly increased from expression levels at 3 days post-insertion ($p < 0.05$). Interestingly,

NG2 expression increased adjacent to the implant at 3 days post-insertion; however, it remained reduced at 1 and 7 days following implantation (Figure 3F). Similar to Iba-1 expression, NG2 expression was highest at 28 days post-insertion and was significantly higher within 0–50 μm compared to NG2 fluorescence intensity at 1 and 7 days post-insertion ($p < 0.05$) (Figure 3G and Supplementary Table S7).”

- From Section Titled “PDGFR β Immunoreactivity and Pattern of Blood-Brain Barrier Leakage Around Implanted Microelectrode Arrays”:
 - “Coincidentally, analysis of BBB leakage via immunoglobulin G (IgG) staining revealed a temporal pattern of vascular disruption similar to the observed pericyte reactivity (Figure 6D). IgG fluorescence intensity was increased with proximity to the device for all time points, up to 50 μm away from the probe hole at 1, 3, and 7 days post-insertion, and up to 100–150 μm away from the probe hole at 28 days post-insertion (Figure 6E). Within the 50 μm region directly adjacent to the probe hole, IgG intensity was highest at 28 days post-insertion, significantly more increased than at 1, 3, and 7 days post-insertion ($p < 0.05$) (Figure 6F and Supplementary Table S14).”

Reference 18: Bennett, C. et al. Blood-brain barrier (BBB)-disruption in intracortical silicon microelectrode implants. *Biomaterials* 164, 1-10 (2018).
<https://doi.org/10.1016/j.biomaterials.2018.02.036>

- Section 3.1: “We used qRT-PCR to determine the relative mRNA expression of specific cytokines that are involved in pro- and anti-inflammatory signaling at multiple acute time-points (6-hr, 24-hr, 48-hr, and 72-hr) after electrode implant surgery. Fig. 2 shows the gene regulation for the pro-inflammatory cytokines TNF α , IL6, IL1 α , and IL1 β and the anti-inflammatory cytokines IL1Rn and Bcl2 at the various acute time-points. We found that the expression of these pro-inflammatory cytokines was most upregulated right after electrode implantation at 6-hr (Fig. 2A). At all time-points, the mRNA transcript expression levels were upregulated for the cytokines TNF α , IL6, IL1 α (>5-fold relative to unimplanted controls) whereas the cytokines IL1 β , Cxcl1, and Cxcl2 showed expression values greater than 20-fold (Fig. 2, Fig. 3B) indicating electrode implant-induced injury.”
- Section 3.3: “The BBB is known for its highly regulated protein structure [29]; therefore, minute changes in expression levels relative to control values were considered as significant. Upon initial injury due to implant, claudin-5 and occludin were slightly downregulated relative to unimplanted animals where baseline expression level was 1 (Fig. 4). The initial downregulation of these genes regulating the TJ proteins is indicative of the acute damage to the BBB structural integrity as a result of implant injury that increases the BBB-permeability and leads to barrier dysfunction. At the 24-hr time point, all the TJ genes, except for occludin, increased above baseline values after the initial injury, suggesting the BBB trying to restore itself after an injury. However, due to the presence of an electrode array in the brain tissue, which leads to persistent disruption of the BBB, we observed downregulation of all the TJ genes at later time-points except Zo-2 (Fig. 4). We also monitored the expression of cadherin family of genes that regulate the AJ proteins. Cadherin-5 (Cdh-5) also known as vascular endothelial (VE)-cadherin, showed an increase in expression values upon injury peaking at 48-hr and significantly decreasing ($p < 0.03$) at 72-hr post-implant (Fig. 5). Cadherin-1 (Cdh1), also known as epithelial (E)-cadherin, remained downregulated at all the time-points tested.”

4) “The integrity of the mucosal lining of the intestines is dysfunctional following brain injury¹⁹”. This reference is related to Traumatic Brain Injury (TBI). Quite often authors make the link between TBI and microelectrode implantation, which is not the same thing. This should be rephrased and/or explained with evidences that implantation and TBI are related.

We have revised the text by adding an additional clarifying statement and reference. We now cite Prof Rob Rennakar’s 2005 study, which suggests similarities between injury characteristics with TBI and microelectrode implantation caused by the compression of brain tissue during implantation.

Prof Rennakar is an expert in **BOTH** TBI and intracortical microelectrode implantation and characterization. See below his Google Scholar profile indicating that he self-identifies expertise in TBI and that his most cited publication is on microelectrode recording performance. He is among the most qualified in the world to make this connection.

However, our revised analysis indicates that “... 18.8% \pm 0.01% of 16S reads at 4 weeks post-implantation and 6.1% \pm 0.01% at 12 weeks post-implantation were from invasive genera found in the gut. 28.1% \pm

0.3% of 16S reads at 4 weeks post-implantation and 2.5% ± 0.01% at 12 weeks post-implantation were from invasive genera of unknown anatomical origin.” Therefore, we have added more context that the dominant mechanism is likely bacteria entering through the damaged BBB, from various origins.

The revised text now reads:

The integrity of the mucosal lining of the intestines is dysfunctional following traumatic brain injury (TBI)¹⁹. Upon microelectrode implantation, there is compression of brain tissue of 1-3 mm, which Rennaker et al. suggest shares similar injury characteristics with TBI due to the compression of brain tissue upon insertion²⁰. Therefore, a potential pathway for gut-derived bacteria to enter the brain following the trauma associated with microelectrode implantation may exist.

Robert Rennaker

Other names ▶

Neuroscience, UT Dallas

Verified email at vulintus.com - Homepage

Neuroscience Biomedical Engineering Stroke Traumatic Brain Injury Spinal Cord Injury

TITLE	CITED BY	YEAR
Long-term neural recording characteristics of wire microelectrode arrays implanted in cerebral cortex JC Williams, RL Rennaker, DR Kipke Brain research protocols 4 (3), 303-313	528	1999

	All	Since 2019
Citations	6850	4094
h-index	44	35
i10-index	88	75

5) “Despite such links, there are no reports of how brain responses to gut microbiome infiltration following microelectrode implantation”. This sentence is an assumption that infiltration following microelectrode implantation has been demonstrated. There is no evidence for this, please rephrase.

The reviewer’s comment is confusing. The statement that the reviewer references is from the introduction, in the portion of the paper where we are setting up the scientific premise and proposing our hypothesis. Asking for evidence at that point is premature. Further, with the revision of Figures 1-2, we have shown

additional evidence demonstrating the bacteria isolated in the brain tissue adjacent to the microelectrode arrays contain bacteria that are found in the fecal sample.

Therefore, we do not feel that this statement requires rephrasing.

6) Page 3: “Our results suggest that modulating the invasion of microbes into the brain may impact microelectrode performance to improve quality and stability”. Over-interpretation, please rephrase.

We appreciate the reviewer’s concerns that our wording suggested that we demonstrated this finding in the current study when, in fact, our results to this effect require further exploration. Therefore, we more clearly wrote the statement to suggest future studies that could be done.

The text now reads: *Our initial findings suggest that future studies could explore the connection between the endogenous (example: gut) microbiome and microelectrode performance by modulating the (gut) microbiome or implementing strategies to manipulate the neuroinflammatory response to invading bacteria. While the source of bacteria not resident to the gut or naïve brain was not identified in this study, future studies could also investigate sources such as the nasal cavity or oral microbiome, and/or that some portion of the 16S DNA is brought into the brain by infiltrating macrophages that had phagocytosed the bacteria before entering the brain during the inflammatory response to the implanted microelectrodes.*

7) Page 3, figure 1:

While I do not doubt in bacteria contamination of the brain after electrode implantation I have several concern for the Brain-biota hypothesis since there is very few studies searching for bacteria in the brain.

Here I am not convinced either.

- a) Since data are normalized in fig1, I would like to have access to coverage and exact number of reads. This should be dramatically different between unimplanted and 4 weeks after implantation. Moreover, the background of this experiment was not taken into account. What is noise and what is results (which means that probably you can’t distinguish between the noise and the signal before implantation).**

Although 16S read counts cannot be used to quantify microbial loads without additional experimental controls directly, the read counts are higher for the four-week post-implantation brains than in the unimplanted or 12-week post-implantation brains. We have added the following to the methods section:

Processed library depths ranged from 7292 to 16382 reads (median: 12100) for unimplanted brain samples, 26,661 to 129,331 reads (median: 40,028) for 4 weeks post-implantation brain samples, 17,351 to 37,695 reads (median: 27,160) reads for 12 weeks post-implantation brain samples, 1,204 to 85,579 (median: 26,725) reads for pre-treatment and control group fecal samples, and 261 to 133,488 reads (median: 19,428) for antibiotic-treated fecal samples.

We are grateful for the reviewer’s suggestion that we more appropriately address the problem of background noise in microbiome studies, as we feel this has improved the overall quality of the work. We performed statistical identification of contaminant sequences as described in Reference 31, and we reprocessed the results with those sequences removed. Contamination accounted for <30% of the reads in many brain tissues 4 weeks post-implantation, which was not significantly different from what was observed in either the antibiotic-treated or baseline fecal samples, suggesting that a similar proportion of 16S amplicons were sample-derived. The section “Bacteria invade the brain after intracortical microelectrode implantation” and corresponding Figure 1A-C have been added to the results to address the removal of background noise (i.e. contamination):

The change is over two pages of text, so it is not listed here, but identified with blue font in the revised manuscript.

***Because of the revised analysis, Figure 2 and the corresponding text was also revised – as indicated with blue font in the manuscript.*

Lastly, despite the high proportion of reads derived from contaminants in the unimplanted and 12 weeks post-implantation brain tissues (Fig 1C), we saw significant differences in alpha diversity between antibiotic-treated and untreated control brains regardless of implantation status after the removal of these contaminants from the analysis (Fig 2B-C). If these remaining 16S sequences retrieved from brain tissues were solely artifactual or due to residual contamination, we would not expect diversity to vary between the two treatment groups, suggesting that these are sample-derived microbes and constitute “true” signal. We feel this validates the reviewer’s assertion that we had not properly accounted for background in our initial analysis, and we thank them for their contribution.

Reference 31:

Davis, N. M., Proctor, D. M., Holmes, S. P., Relman, D. A. & Callahan, B. J. Simple statistical identification and removal of contaminant sequences in marker-gene and metagenomics data. *Microbiome* **6**, 226 (2018). <https://doi.org/doi:10.1186/s40168-018-0605-2>.

b) A minor concern is the absence of a paragraph in the introduction discussing the microbiota of the healthy brain.

We recognize that the existence of a brain-specific microbiome in the healthy brain is a controversial topic. The goal of this study was not to argue for or against the existence of a healthy brain microbiome. That concept/hypothesis is beyond our study's scope and detracts from this study's main finding. Therefore, we have added the following statement to the second paragraph of the **Introduction**:

“The existence of a brain-specific microbiome is beyond the scope of this study. However, publicly presented data,…”

We also added following paragraph to the discussion section:

Recent studies have expanded the scope of the human microbiome to include the brain, yet the "Brain-biota hypothesis" remains controversial and under-explored. Contamination is a major concern in microbial DNA research, especially from typically sterile sites like the brain. To address this, we utilized computational and statistical methods to distinguish between genuine microbial DNA and potential contaminants. Our results indicate that the microbial DNA found in implanted brain tissues does not align with random contamination but rather suggests a non-random pattern of microbial infiltration, likely linked to blood-brain barrier disruption following microelectrode implantation. Although the idea of bacteria residing in the brain is contentious due to the limited studies in this area, we believe our rigorous approach effectively mitigates the risk of contamination, supporting the presence of microbes in situ within the brain tissue.

Additionally, we revised the second to last line of the **Abstract** to state our focus on invading bacteria more clearly.

A significant portion of invading bacteria was not observed in the gut or naïve unimplanted brain.

8) Page 5 : The use of this mix of antibiotic is not clear. Page 5 the authors claimed that “mice was treated with antibiotics to deplete fecal microbiota”. In the mat and methods the authors claimed that “Such antibiotics were chosen based off previous literature to provide broad spectrum capacity and effect on the gut microbiome”

In fact, Ampicillin, Clindamycin, and Streptomycin clearly promote dysbiosis (for instance:The causing inflammation and some brain dysfunction. Therefore, my question is: What was the purpose of using this mix ? It’s not clearly mentioned in the manuscript. My guess is that the use of this mix is likely to cause biases that do not seem to have been anticipated by the authors. And given that this mix induces dysbiosis, what conclusion can we draw from this experiment?

The goal of using this mixture was to deplete bacterial populations (induce intestinal dysbacteriosis) in the gut microbiome to observe whether changes to gut microbiome composition impact brain health and microelectrode function and to measure the possibility of bacteria invasion to the brain. We expected gut dysbiosis to occur because of treatment. We chose this mixture to confidently say that bacteria composition is different to characterize how gut dysbiosis can influence microelectrode function, neuroinflammation, and the possibility of bacterial invasion post-implantation. We have attempted to make this clearer in the revised manuscript.

Within the results section, a section under the sub-heading “Antibiotic Treatment Mitigates Invasive Microbiome Diversity in the Brain, the text now begins with:

“An additional cohort of mice was treated with antibiotics to deplete fecal (gut) microbiota, characterize differences in brain microbial composition as compared to untreated mice, and identify any association between microbial composition and microelectrode recording performance. Antibiotic-treated mice were provided with an antibiotic cocktail of Ampicillin, Clindamycin, and Streptomycin in their drinking water following established protocols³⁰. Antibiotic-treated mice displayed significant alterations to the gut microbiome as early as one week after the start of treatment, which continued throughout the study (Supplemental Fig. S1A-C).“

9) Page 6/7 what is the meaning of having 55.8% of reads of unknown origin ? Are these reads not linked to any phyla? Or is the anatomical origin of these phyla unknown ? Please rephrase and give information on these reads.

Thank you for the important clarification. Throughout the revised manuscript, the term “unknown origin” was removed and/or replaced with “unknown anatomical origin”.

10) Page 9, fig2 should be fig3

Thank you. We have updated/corrected figure numbers and in-text references throughout the revised manuscript.

11) Figure 3: I do not understand the use of Acute and chronic for implantation of electrodes.

To help with this clarification of terminology, we added the following text and reference to the methods section, which describes our neural recording strategies:

“Recording data was binned into three distinct phases corresponding to the progression of neuroinflammation after implantation¹¹⁵⁻¹¹⁷: an acute phase (weeks 0 – 5), a sub-chronic phase (weeks 6 – 11), and a chronic phase defined as any time points after week 11.”

12) Figure 3 C-F: What is the relevance of such a difference?

To help with this clarification, we added to end of IME performance results section:

“Although there were no significant differences in AEY at the sub-chronic time point, the larger amplitude and spiking rate may indicate healthier and more active neurons in the antibiotic group at that time point.”

13) Page 11: “Across all comparisons, 28 of the 39 possible proteins“ do you mean 39 tested proteins ? Why these 39 proteins have been chosen ? Are they related to inflammation ? For instance there is

marker of M2 macrophages but not that of M1 macrophage. Therefore, the authors cannot conclude on this point (page 12).

Correct, this should have read “tested” instead of possible. This was corrected in text.

The 39 proteins that were examined in the current manuscript were a neuroinflammation panel provided by the vendor, NanoString Technologies. All proteins assembled in this panel are listed in Table 1. While there are other targets that could have been included in this analysis, it is among the most comprehensive assessment of the neuroinflammatory response to intracortical microelectrodes arrays ever reported – second only to one of our manuscripts that was accepted while this manuscript was under revision.

Additionally, to place the results in context, for those that may read the methods section separate from the results section, the following text was added to the appropriate part of the results section that discusses the proteomic analysis:

“Employing panels for neural health from NanoString, proteins were measured pertaining to neural cell profiling (25 proteins), glial cell subtyping for identification of specific cell types (10 proteins), and autophagy processes (10 proteins) (Table 1).”

Additionally, the following text was added to the end of the same section of the manuscript to provide more clarity to the presence or absence of protein markers for inflammation, including M1 vs M2 microglia and macrophage phenotypes.

There are several proteomic markers for microglia activation between their pro-inflammatory M1 phenotype (CD14, CD16, CD32, CD40, CD86, MHCII) and their anti-inflammatory M2 phenotype (CD163 and CD206), a few of which were measured in this study⁴³. Consistently across comparisons, the antibiotic group shows a downregulation in CD40, indicating reduced M1 microglial activity. Additionally, CD68 is a common protein marker for microglia and macrophages⁴⁴, showing consistent downregulation across the comparisons above. Notably, CD163 is also downregulated, but only in a single comparison above.

14) Discussion, - Page 17: “Therefore, the principal hypothesis of the current study was that damage to the BBB caused by microelectrode implantation would amplify dysregulation of the microbiome-gut-brain axis”. This is far out of the results of this work.

We believe that the hypothesis comes before the experiments are designed and performed, and the results are interpreted. We did not change our hypothesis retroactively to fit the data that we obtained from the study that we carried out. Our presentation of the data is honest and unbiased, and our discussion and conclusion are supported by the actual data, not preconceived hypotheses. Therefore, the hypothesis remains. However, to try to articulate better, the text was revised to:

“Therefore, the principal hypothesis of the current study was that damage to the BBB caused by microelectrode implantation facilitates/permits the infiltration of gut-resident microbes into the brain, contributing to the chronic neuroinflammatory response and decreased performance of intracortical microelectrode arrays.”

15) “Further, several studies in rodents and humans have shown that traumatic brain injuries are accompanied by increased intestinal permeability and intestinal barrier dysfunction^{19,47,48}. Therefore, we were particularly interested in understanding the role that microbes that reside in the intestines may have on microelectrode performance if they invade the brain tissue following microelectrode implantation” Comparison of TBI and electrode implantation should not be used.

See the response to point #4 above.

16) “we have demonstrated that microbes associated with the gut microbiome invade the brain tissue”. I fully disagree.

Unfortunately, the vague nature of this comment makes it difficult to respond. However, we feel that the additional analysis completed and reported in the revised **Figure 1-2**, related text in the manuscript, and supporting information strengthen and clarify our data, supporting this claim.

17) “The differential composition and abundance of invasive microbes were associated with significant temporal changes in the recording performance of intracortical microelectrodes”. I am not convinced. There is not statistical evidence for this.

We have revised the questioned statement in the fourth paragraph of the Discussion section of the manuscript. The sentence in question has been removed completely and replaced with”

“Depletion of the gut microbiome via systemic antibiotic treatment was associated with better microelectrode performance up to 5 weeks post-implantation, lower microbial beta diversity, lower abundance of gut-resident microbial taxa, and lower abundance of pro-inflammatory microglia/macrophage proteins in the associated brain tissue as compared to untreated controls following implantation (Fig. 1-4).”

18) “It is important to recognize that even subtle changes in the microbe composition in the brain have been linked to changes in brain health^{20,22,23} .” Ref are inappropriate. Ref 20: quote: “Here I argue that the evidence for the presence of microbes in diseased brains is quite strong, but a compelling demonstration of resident microbes in the healthy human brain remains to be done”

We appreciate the sensitive nature of the idea of a native brain microbiome. We feel that the idea of a native brain microbiome is outside the scope of this publication, and we do not wish to entangle that hypothesis with the current study. The intent of the statement that the reviewer calls into question was to point out that in disease states, bacteria not believed to be resident in the brain are suspected of influencing the brain's health when found in the brain tissue. Therefore, we have done **three** things to soften the language:

The word “native” as a prefix to microbiome was removed from all three locations in the text.

We rewrote the sentence to now read:

“It is important to recognize that even subtle changes in microbiota composition have been linked to changes in brain health^{21,23}.”

We removed reference #23 from the manuscript (three places), which was the third reference in the original manuscript. Note, new references have been added before this statement, so the numbering from the original submission to the revision is different – yet we have removed the reference noted by the reviewer.

19) Conclusion:

19a) “Further, it is possible to modulate the neuroinflammatory response following implantation and microelectrode performance by altering the composition and abundance of invasive microbes”. This is overinterpretation.

We appreciate the reviewer’s concerns that the conclusion statement could have extrapolated further than they were comfortable with. Therefore, we rewrote that statement to now read:

“Further, it is possible to modulate the neuroinflammatory response following implantation and microelectrode performance by **depleting gut bacteria using an antibiotic cocktail, reducing the abundance of gut-resident bacteria in the brain. Our findings suggest that alternative strategies could locally target invading bacteria rather than systemically manipulating all bacteria in the host.**”

19b) “Some of the microbial strains identified in brain tissue in this initial study have been previously associated with neurodegenerative symptoms and diseases”. I doubt that this study has identified the strains.

We appreciate the reviewer’s attention to detail. We have corrected our mistake to now read “taxa” rather than “strains.”

Reviewer #2 (Remarks to the Author):

This study makes a convincing case that intracortical microelectrode implantation alters the distribution of microbial sequences in the brain, and antibiotic treatment can influence the microbial sequence changes induced by electrode implantation. With the growing use of implants in people, these findings have clear clinical relevance. However, there are some technical issues:

1). There is an ongoing concern in the field that bacterial sequences recovered from mouse (or human) tissue are artifactual (e.g., introduced during library preparation or sequencing). The alteration of the bacterial population after implantation makes a good case that the bacterial sequences are "real", particularly if all samples were processed in parallel. It is therefore important for the authors to give more details about sequencing library construction and whether all 16S reads were sequences in the same Illumina run.

We thank the reviewer for identifying these concerns regarding background signal (i.e. contamination) and batch effects in microbiome studies, as we feel addressing them has improved the overall quality of the work. We performed statistical identification of contaminant sequences as described in Reference 31, and we reprocessed the results with those sequences removed. Contamination accounted for <30% of the reads in many brain tissues 4 weeks post-implantation, which was not significantly different from what was observed in either the antibiotic-treated or baseline fecal samples, suggesting that a similar proportion of 16S amplicons were sample-derived. The section “Bacteria invade the brain after intracortical microelectrode implantation,” and corresponding Figure 1A-C have been added to the results to address the removal of background noise (i.e., contamination).

The change is over two pages of text, so it is not listed here, but identified with blue font in the revised manuscript.

***Because of the revised analysis, Figure 2 and the corresponding text was also revised – as indicated with blue font in the manuscript.*

Reference 31:

Davis, N. M., Proctor, D. M., Holmes, S. P., Relman, D. A. & Callahan, B. J. Simple statistical identification and removal of contaminant sequences in marker-gene and metagenomics data. *Microbiome* 6, 226 (2018). <https://doi.org/doi: 10.1186/s40168-018-0605-2>.

We have added the following to the methods to address batch effects in library construction:

Samples were processed across four batches: (1) naïve unimplanted brains and their associated fecal samples, (2) 4-weeks and 12-weeks post-implantation brains, (3) fecal samples associated with 4-weeks post-implantation brains, and (4) fecal samples associated with 12-weeks post-implantation brains. At least one no-template library was sequenced in each run for a total of five sequencing blanks.

Because the implanted and unimplanted brains were sequenced separately, we have added the following statement to the second paragraph in the “Antibiotic Treatment Reduces Invasive Microbe Abundance” section to acknowledge some conclusions may be sensitive to a batch effect:

Although we recognize that differences in the observed features between implanted and unimplanted brains could be attributed to their being sequenced separately, for convenience's sake in this analysis, any taxa that were not originally detected in the unimplanted brain were considered potentially "invasive" bacteria in the implanted brains. At the same time, any taxa observed in the unimplanted brain were referred to as "non-invasive" bacteria in the implanted brains.

We also added the following paragraph into discussion (paragraph 3).

Recent studies have expanded the scope of the human microbiome to include the brain, yet the "Brain-biota hypothesis" remains both controversial and under-explored. Contamination is a major concern in research involving microbial DNA, especially from typically sterile sites like the brain. To address this, we utilized computational and statistical methods to distinguish between genuine microbial DNA and potential contaminants. Our results indicate that the microbial DNA found in implanted brain tissues does not align with random contamination but rather suggests a non-random pattern of microbial infiltration, likely linked to blood-brain barrier disruption following microelectrode implantation. Although the idea of bacteria residing in the brain is contentious due to the limited studies in this area, we believe our rigorous approach effectively mitigates the risk of contamination, supporting the presence of microbes in situ within the brain tissue.

In response to the concern, we removed the analysis of differential abundance in the brain tissues with ANCOM-BC from the manuscript (original Fig 2A). The unimplanted brain served as a baseline for the model, and after considering the reviewer's point regarding batch effects, we no longer felt this analysis was reliable. The ANCOM-BC model for differential abundance in the fecal samples controlled for sequencing batch (Supplemental Fig 1C), so this analysis was retained.

We do feel that our primary conclusions stand despite the limitations of our experimental design. Regarding the invasion of bacteria into the brain, Figure 1C shows that there is greater abundance of sample-derived microbial DNA in the 4 week post-implantation brain versus the 12 week post-implantation brains, which were processed and sequenced in parallel, suggesting a transient population of microbes that cannot be explained by a batch effect. Regarding the effect of systemic antibiotic treatment, all antibiotic-treated samples were processed and sequenced in parallel with their corresponding untreated control cohort, so the reduced abundance of invasive microbes (Fig 1E) and reduced alpha diversity (Fig 2B-C) in antibiotic-treated animals also cannot be explained by a batch effect. Lastly, although the significant difference in beta diversity by implant status in untreated control animals could be due to the separate sequencing of unimplanted brains (Fig 2D), the meaningful overlap between implanted and unimplanted samples in the antibiotic-treated cohort suggests these samples are compositionally similar despite having been processed separately (Fig 2E). We have added the following statements to address this:

Lines 213-215: "All implanted brains were processed and sequenced in parallel, so the variation in invasive microbial abundance between the acute and chronic time points cannot be explained by a batch effect."

Lines 223-225: "All antibiotic-treated brains were processed and sequenced in parallel with the corresponding control cohort, so the variation in invasive microbial abundance between treatment groups cannot be explained by a batch effect."

Lines 263-264: "The implanted brains were processed in parallel and sequenced together, so the variation in alpha diversity between the two time points in implanted brains cannot be explained by a batch effect."

Lines 273-280: "All antibiotic-treated samples were processed in parallel and sequenced with the corresponding untreated controls, so the variation in alpha diversity between the two treatment groups cannot be explained by a batch effect. In addition, were these 16S amplicons extracted from whole brain tissues and analyzed here solely artifactual (e.g. residual contaminants persisting beyond quality control), we would not expect to them to vary in response to antibiotic-treatment of the animal. The variation seen

here supports the plausibility that we were successful in retrieving sample-derived microbial sequences from whole brain tissues, even naïve unimplanted brains, and the composition of these sample-derived microbes varies in response to systemic antibiotic treatment.”

Lines 294-301: “Of note, there was little overlap between the implanted and unimplanted brain tissue, which indicated either a batch effect from sequencing unimplanted and implanted brains separately or that the brain environment had still not returned to a baseline state after 12 weeks. However, in the antibiotic-treated cohort, the unimplanted, acute, and chronic brain tissues did not form distinct clusters (Fig. 2E). The considerable overlap between the three groups, despite the unimplanted brains being sequenced separately, suggested that systemic antibiotic treatment may mitigate invasive microbe diversity and promote a return to or maintenance of the baseline, unimplanted brain environment in response to IME.”

2). An important distinction is whether bacterial sequences found in brain tissue are actually in the brain parenchyma, or in the associated blood vessels. (There is abundant evidence for microbial sequences in human blood.) Although the investigators perfused their mice before tissue sampling, it is unclear if this completely depletes bacterial sequences that reside in the blood. Do the authors have any evidence that the detected bacterial sequences are actually in the brain parenchyma?

We thank the reviewer for the interesting question. Our initial assumption was that the perfusion was complete, and nothing was left in the blood because the perfusate ran clear before the perfusion procedure was terminated.

To be as comprehensive in our response as possible, we performed two experiments on the tissue we have remaining from the original study. (1) We examined the tissue for evidence of blood products left in the vasculature using an iron-detecting stain. (2) We examined the tissue for both gram-positive and gram-negative bacteria. Because tissue was only available from limited animals in limited groups, our analysis was qualitative and not quantitative. Therefore, we only include the figures in the response to reviewers and have not included the images in the manuscript.

Our iron stain (**Reviewer Response Figure 1**) showed no detectable signal in either the control or antibiotic-treated implanted animals, suggesting no presence of blood in the brain. This result demonstrates the effective removal of any blood (and blood products) during the perfusion.

(A) Iron Stain 4-week Antibiotic

(B) Iron Stain 12-week Control

(C) Iron Stain 12-week Antibiotic

Ferric deposits: blue

Nuclei: red

Background tissue elements: pink

Reviewer Response Figure 1: The Iron Stain Kit (Agilent DAKO, Part: AR15892-2) stains blue for any ferric deposits, red for nuclei, and pink for background tissue. Frozen brain tissue collected from mice at 4- and 12-weeks post-implantation of both treatment groups and cryosectioned at 5 μm was stained and imaged using a light microscope. The 4-week implanted antibiotic-treated mice (A) along with the 12-week control (B) and antibiotic-treated mice (C) all showed a lack of signal detected. Due to a lack of histological markers to identify the stain such as neurons or astrocytes, along with the minimal, if any, signal detected, it was impossible to locate or estimate the location of the implant site. One of the biopsy punch locations used for other tissue processing was identified in (A) (outlined circle), however no detectable iron staining was measured at that site, only sparse nuclei. There does not appear to be any iron detected across both time points and treatment groups. Unfortunately, no 4-week control mice brains were able to be stained for iron due to lack of tissue remaining from the study. However, given the extreme lack of any iron deposits detected in any of the stains, it is very unlikely that the 4-week control brain would show any different result. It appears from these stains that the perfusion process leaves no blood detected in the brain. Scale bars are shown on each image.

Also, a gram-negative bacteria stain on the above groups (**Reviewer Response Figure 2 and 3**) showed bacteria (dark blue stain) within the brain around the estimated location of the implant site. It was difficult to identify the implant site due to a lack of classical histological stains, such as astrocytes and neurons. However given our implant coordinates, we could estimate the location. Results from this stain suggest that bacteria are present in the brain and not isolated solely to vasculature. We cannot completely rule out contamination during the staining procedure or facility processing.

Gram-positive organisms: dark blue | Gram-negative organisms: light pink to magenta | Background: yellow

Reviewer Response Figure 2: The Gram Yellow Stain Kit (Agilent DAKO, Part: AR30692-5) stains dark blue for any gram-positive organisms, light pink to magenta for any gram-negative organisms, and yellow for background. Frozen brain tissue collected from mice at 4- and 12-weeks post-implantation of both treatment groups and cryosectioned at 5 μm was stained and imaged using a light microscope. Gram-negative bacteria appear to be consistently present and distributed in 4-week and 12-week mice of both treatment groups. A zoomed-out image (Left) and zoomed-in image (Right) of the implant site were taken to give an idea of distribution throughout the brain and directly around the implant with corresponding scale bars shown in the bottom right of each image. The 4-week implanted untreated control mice (A) and 4-week implanted antibiotic-treated mice (B) showed a consistent distribution of bacteria with no discernible qualitative difference between the groups. A similar distribution is seen in the 12-week untreated control mice (C) and the 12-week antibiotic-treated mice (D). Black bars were drawn over the estimated location of the implant site. Due to a lack of histological markers to identify the stain such as neurons or astrocytes, the location of the implant site is estimated based on stereotaxic coordinates used during surgery.

Gram-positive organisms: dark blue | Gram-negative organisms: light pink to magenta | Background: yellow

Reviewer Response Figure 3: The Gram Yellow Stain Kit (Agilent DAKO, Part: AR30692-5) stains dark blue for any gram-positive organisms, light pink to magenta for any gram-negative organisms, and yellow for background. Frozen brain tissue was collected from non-implanted sham mice (A and B) and cryosectioned at 5 μm was stained and imaged using a light microscope. Gram-negative bacteria appear to be consistently present and distributed within both brains.

3) The spatial transcriptomics is not well described, presented, or analyzed. This reviewer was confused about whether the differentially-expressed genes identified are solely between implanted animals with and without antibiotics, or are being compared to unimplanted mice. If it is the former, the investigators cannot use the data to follow the trajectory of inflammation caused by the implantation. This is unfortunate, because it makes the interpretation of +/- antibiotics more difficult (e.g., it is not known what gene changes occur just because the animals have gotten older). In terms of presentation, it is incumbent on the authors to present all the differential gene changes in a supplementary table, so readers can do their own interpretation. In terms of analysis, I would have expected a typical gene ontology analysis, with P values for the enriched categories, rather than the cherry-picked categories presented. Glaringly missing is any indication of changes in gene categories associated with immune response or inflammatory genes. If these genes do not change, then this should be explicitly pointed out by the authors.

3a) “The spatial transcriptomics is not well described, presented, or analyzed. This reviewer was confused about whether the differentially-expressed genes identified are solely between implanted animals with and without antibiotics, or are being compared to unimplanted mice. If it is the former, the investigators cannot use the data to follow the trajectory of inflammation caused by the implantation. This is unfortunate, because it makes the interpretation of +/- antibiotics more difficult (e.g., it is not known what gene changes occur just because the animals have gotten older).”

We thank the reviewer for bringing up a valid criticism of a lack of unimplanted animals for comparison. However, our main goal was not to compare antibiotic treatment to unimplanted brains, but to measure whether an antibiotic treatment would impact the brain and inflammatory response compared to an untreated, implanted animal. However, your point is well taken. The study would be stronger had we included a comparison to sham animals to get an idea of what a “baseline” transcriptomic and proteomic response would look like. We have ongoing studies that are looking into tracking inflammation using transcriptomics for implanted vs non-implanted animals. In this study, both control and antibiotic animals are of the same age, so any alterations to genes should be solely from antibiotic treatment.

We have added the below text to the section entitled “Antibiotic Treatment Impacts the Neuroinflammatory Response to Intracortical Microelectrodes”, in paragraph 9.

“Mouse age and implant status were controlled to allow us to conclude transcriptomic changes based solely on antibiotic treatment versus no treatment control. Of note, comparisons were not made between implanted (antibiotic and control) and non-implanted, healthy mice. Future studies could incorporate healthy mouse comparisons to bolster analysis and comparisons across groups.”

3b) “In terms of presentation, it is incumbent on the authors to present all the differential gene changes in a supplementary table, so readers can do their own interpretation.”

We thank the reviewer for the crucial suggestion to include a data sheet containing all transcriptomic analysis data. Extensive information was added in the “Antibiotic Treatment Impacts the Neuroinflammatory Response to Intracortical Microelectrodes” results section regarding additional pathways and the inclusion of GO terms to explore immune response and highlight the inability to analyze all genes, pathways, and GO terms. Further methodological clarification was added to explain our selection of analysis and include a comprehensive sheet containing all genes, pathways, and GO terms analyzed. An additional supplementary data sheet is now available (“Full Gene Data.xlsx”) containing all gene, GO term, and KEGG pathway p-values and $\log_2(\text{fold change})$'s across the 4- and 12-week time points.

3c) “In terms of analysis, I would have expected a typical gene ontology analysis, with P values for the enriched categories, rather than the cherry-picked categories presented. Glaringly missing is any indication of changes in gene categories associated with immune response or inflammatory genes. If these genes do not change, then this should be explicitly pointed out by the authors.”

We thank the reviewer for revealing a hole in our results explanation regarding a lack of specific p-values, gene ontology, and expansion on inflammatory pathways. We have revisited our analysis and included far more details on each of these points. These changes can be found in the section titled “Antibiotic Treatment Impacts the Neuroinflammatory Response to Intracortical Microelectrodes”

1. Expanding on additional inflammatory pathways

“Some pathways discussed were not identified in iPathways as being significantly altered. However, since there are still many differentially expressed genes implicated in the pathway, it was deemed important to discuss the possibility that the altered genes could influence the functions of said pathways.”

“Ribosomal genes are involved in regulating immune responses, such as Rps3, which has multiple functions including regulating the production of inflammatory marker, NF-K β , and is upregulated at the 4-week time point (Log₂(FC) = 0.277, p-value = 0.0387.”

“Inflammation and the immune response are crucial when characterizing the body's response to an intracortical microelectrode. NOD-like receptors are key regulators of inflammation, especially the innate immune response ^{ref}. At 12-weeks post-implantation, 19 of 69 total genes associated with NOD-like receptor signaling (KEGG: 04621, p-value = 0.019) were differentially expressed (10 upregulated, nine downregulated). Conversely, at 4-weeks post-implantation, there only two genes were upregulated in the NOD-like receptor signaling pathway (p-value = 0.354). Interestingly, the COVID-19 disease pathway (KEGG: 05171) was also impacted at 4-weeks post-implantation where 18 out of 126 genes were differentially expressed (17 upregulated, p-value < 0.0001). Of the effected genes, there was an upregulation of NF-kB Inhibitor B (Nfkbib, Log₂(FC) =0.387, p-value = 0.026), which inhibits the activity of the common immune and inflammatory initiator, NF-kB ^{ref}. It is also worth noting that 16 of the 18 DE genes are from the ribosomal dysfunction group, which has previously been noted in Figure 5C and reinforces their involvement in the inflammatory process. By 12-weeks post-implantation, the number of DE genes in the COVID-19 pathway increases to 23, however without significance (p-value = 0.635), including a lack of significance of Nfkbib (Log₂(FC) = 0.224, p-value = 0.373). Like the 4-week timepoint, most genes impacted are from the ribosomal dysfunction group.”

2. Expanding on the transcriptomic analysis by performing and discussing Gene Ontology (GO) analysis.

“To further explore the changes to inflammation and neurological processes from antibiotic treatment, gene ontology (GO) of biological processes were analyzed. GO terms are sets of genes that contribute to a specific biological, cellular, or molecular function. There were 156 significantly impacted GO terms at 4-weeks post-implantation, 227 at 12-weeks post-implantation, and only one GO term was significant and shared across both groups (Cytoplasmic translation, GO: 0002181). G protein-coupled receptor signaling is responsible for a wide range of cellular functions including neurotransmission, cell metabolism, and immune response ^{ref}. At 4-weeks post-implantation, there are only 12 out of 711 genes differentially expressed in the G protein-coupled receptor signaling pathway (GO: 0007186, GO p-value = 1.000), which grows to 238 out of 711 genes at 12-weeks post-implantation (227 upregulated, 11 downregulated, GO p-value < 0.0001). Nervous system processes (GO: 0050877) have only 30 out of 998 at 4-weeks post-implantation (majority downregulated, GO p-value = 1.000) and 278 out of 998 genes differentially expressed at 12-weeks post-implantation (majority upregulated, GO p-value < 0.0001). When looking at immune response processes at 4-weeks post-implantation, there are only 3 out of 51 genes differentially expressed for the regulation of production of molecular mediators of immune response (GO: 0002700, GO p-value = 0.593) and 3 out of 37 differentially expressed for positive regulation of production of molecular mediators of immune response (GO: 0002702, GO p-value = 0.380).

Contrast this to 12-weeks post-implantation where 16 out of 51 (GO p-values = 0.007) and 12 out of 37 (GO p-value = 0.013) differentially expressed genes, respectively. Additionally, at 12-weeks post-implantation, there are three out of five genes significantly expressed in both mucosal immune response (GO: 0002385, GO p-value 0.035) and organ-specific immune response (GO: 0002251, GO p-value = 0.035).

While it is difficult to draw conclusions based on pathway and GO term analysis, the 12-week time point has a more altered immune and neurological response than the 4-week time point, given the abundance of significantly impacted genes, pathways, and GO terms. It is possible that long-term antibiotic treatment and remodeling of the gut microbiome significantly influence the body's response to intracortical microelectrode implantation. Yet, it remains important to remember that the antibiotic treatment used in this study was not proposed to be a treatment to overcome the effects of microbiome invasion at the implant site. Antibiotic treatment was used as a standard method to alter the gut microbiome composition to determine if changes in gut microbiome composition would correspond to changes in the composition of invading microbiota, impacting microelectrode performance. As there are thousands of genes, hundreds of pathways, and thousands of GO terms, not all were analyzed and discussed in this study. A file containing all gene data with their respective log₂FC and p-values, as well as pathways and GO terms are included in the **Supplemental Information** for reader accessibility and transparency.”

In the discussion, line 693-701: The 12-week group also showed 19 DE genes involved in the NOD-like pathway, a regulator of the innate immune and inflammatory response. The upregulation of NOD-like genes could indicate increased inflammation in the antibiotic group compared to control, which may contribute to the sharp decline observed in recording performance at that time point. Overall, the 12-week timepoint had many more pathways and GO terms impacted compared to the 4-week time point, which may explain the drop off in recording performance. In contrast, proteomics showed more significantly impacted proteins at 4-weeks point-implantation, which, in combination with the transcriptomics, we believe helps explain the improvements to recording performance. Such differences between analyses highlight the importance of using both proteomics and transcriptomics to understand the effects of treatment.

4). The authors likely have a valuable data source that they should tap. There is a high probability that there are microbial sequences (particularly bacterial rRNA reads) in the RNA-seq data they generated. (How many depend on how whether their RNA-seq libraries were generated from total or polyA-selected RNA, which they need to detail in their Methods). There are multiple pipelines available (e.g., <https://pubmed.ncbi.nlm.nih.gov/33914880/>) to identify microbial sequences in human or mouse RNA-seq data. Identifying these reads would enable the investigators to confirm their 16S data as well as identify novel microbes. In theory, they might be able to associate specific microbes with specific regions of interest, or specific cell types.

Additions to the spatial transcriptomics methods portion have been made to clarify the sequencing and analysis procedure to highlight the unique methodology of NanoString's spatial transcriptomics process. Unfortunately, we do not have RNA sequences collected from the brain due to the DSP barcode technology, which is now better explained in the methods. Therefore, it is impossible for us to extract information regarding bacterial sequences from our transcriptomics data for further validation. While details were added throughout the section, a bulk of what the review is asking for clarification could be found in the below-revised text.

Briefly, the NanoString barcode identification technology consists of binding a target complementary sequence to the target RNA. Attached to the target complementary sequence is a photocleavable linker with a Digital Signal Profiler (DSP) barcode at the end that corresponds to a specific gene of the mouse transcriptome. The DSP barcode is composed of a sequence of oligonucleotides that are unique to each gene of the mouse transcriptome, according to NanoString's library. The target RNA is not actually collected during the collection process; instead, the photocleavable linker is detached and the DSP barcode is collected that corresponds to the target RNA / gene sequence. Once collected, the

oligonucleotide DSP barcode is sent to the Case Western Reserve University genomics core for sequencing using the Illumina NextSeq 550. After sequencing, FASTQ files were loaded into NanoString's NGS pipeline software to convert into DCC before processing using the GeoMx software suite. Going through the NanoString NGS pipeline allows for converting the oligonucleotide sequences from the DSP barcode into the corresponding mouse genes composing the entire transcriptome. It is important to point out that the whole transcriptome RNA of the mouse is not being collected and sequenced here. The oligonucleotides of the NanoString DSP barcodes are being sequenced.

Minor points:

1) line 94: presumable the investigators did not extract bacterial DNA from biopsy punches; they extracted total DNA from which they amplified 16S sequences.

Now line 102, this has been clarified under the section titled "Bacteria Invade the Brain After Intracortical Microelectrode Implantation"

"Here, the V3-V4 region of the gene for the 16S rRNA small subunit was sequenced using total DNA extracted from brain biopsy punches and pre-treatment baseline and weekly fecal samples...."

On line 938-940, this has also been clarified under the Methods in the section "16S Bacterial DNA Sequencing"

"Total DNA was isolated using the QIAamp PowerFecal Pro DNA Kit (Cat.No. / ID:51804) and sequencing libraries prepared according to the 16S metagenomic sequencing library preparation protocol for the Illumina MiSeq system (Illumina Inc., San Diego, CA, USA)."

2). Results text: It would be helpful give the N of mice used for the various conditions, rather than forcing the readers to count this up in the Figures.

We added these to the opening paragraphs of the results section.

Lines 102-107: "Here, the V3-V4 region of the gene for the 16S rRNA small subunit was sequenced using total DNA extracted from brain biopsy punches and pre-treatment baseline and weekly fecal samples from unimplanted, untreated control mice 2 weeks after housing separation (n = 5) and intracortical microelectrode-implanted, untreated control mice from the acute (n = 6) and chronic time points (n = 7), 4- and 12-weeks post-implantation respectively."

Lines 110-111: "To determine the impact, if any, of depleting the fecal (gut) microbiota on bacteria in the brain, an additional cohort of mice were treated with antibiotics: unimplanted (n = 5), acute (n = 5), and chronic (n = 6)."

The sample size for the proteomic and transcriptomic analysis is already present under the section titled "Antibiotic Treatment Impacts the Neuroinflammatory Response to Intracortical Microelectrodes", in paragraph two.

We added the numbers again in paragraph 9 of the same section to help remind readers as they make it further through the section of dense text.

"Of the 8259 genes included in our analysis, 490 were differentially expressed at 4-weeks post-implantation (n = 4 for antibiotic, n = 3 for control), and 1375 genes were differentially expressed at 12-weeks post-implantation (n = 3 for antibiotic, n = 3 for control)"

Additionally, the sample size was added to both the proteomic (Figure 4) and transcriptomic (Figure 5) figures and captions.

3) An important result of this study is that the majority of microbial sequences in the implant brain do

not appear to come from the gut. Potential alternative sources could be the oral microbiome or the nasal cavity. Perhaps the authors should discuss the possible sources a little bit more and describe follow-up approaches.

The first two sections of the Results portion of the manuscript have been dramatically rewritten. Based on the reviewers' comments and suggestions, we have refined our protocols further. Both figures 1 and 2 have been replaced. Our results now more conclusively identify the proportion of the microbiome found in the brain that was also found in the feces.

To specifically address the reviewer's comments regarding the potential that unidentified microbiome could be from oral or nasal sources, the last paragraph of the introduction which summarized the results was edited to discuss possible sources.

“Our study explores the role of **the microbiome-gut-brain axis and** neuroinflammatory response following intracortical microelectrode implantation in a mouse model. The current study was designed to test the hypothesis that microelectrode implantation could **disrupt** the microbiome-gut-brain axis **via changes to the composition of the gut microbiome and/or infiltration of the brain by gut-resident microbes**. Utilizing 16S rRNA gene sequencing, we have **identified a transient population of bacteria previously observed in the gut but not the naïve brain** and bacteria of an undefined **anatomical** origin **observed in either the gut or naïve brain** following microelectrode implantation. We also demonstrated that systemic antibiotic treatment altered **microbes' abundance and composition** in feces and **implanted** brain tissue. Manipulations of the **gut** microbiome with antibiotic treatment were associated with changes in single-unit recordings using intracortical microelectrodes **and** temporal changes in the neuroinflammatory response as indicated through spatial proteomics and spatial transcriptomics. **Our initial findings suggest that future studies could explore the connection between the gut microbiome and microelectrode performance by modulating the gut microbiome or implementing strategies to manipulate the neuroinflammatory response to invading gut-derived bacteria. While the source of bacteria not resident to the gut or naïve brain was not identified in this study, future studies could also investigate sources such as the nasal cavity or oral microbiome, and/or that some portion of the 16S DNA is brought into the brain by infiltrating macrophages that had phagocytosed the bacteria before entering the brain during the inflammatory response to the implanted microelectrodes.**”

Additionally, in the discussion, lines 596-603, we added:

While estimates vary depending on the source, there is a consensus that bacteria at least equal if not far outnumber the number of cells in the human body, but not all come from the gut microbiome⁵⁹. In fact, different body sites such as the skin, oral cavity, lung, and nasal cavity each develop their own individual microbiomes⁶⁰⁻⁶². In a paper published in Nature in 2022, Hosang et al. concluded that by altering the microbiome of the lung, it was possible to modulate immune signals in the brain by impacting microglia⁶³. Combined with our results, the Hosang study and similar literature⁶⁴ highlight the importance of expanding our horizons beyond just the gut microbiome in future studies to understand how other bacterial sources in the body may influence the brain.

As far as follow-up approaches, we have introduced the idea of implementing antimicrobial coatings on the surface of the microelectrodes. We have completed a study doing this since the original submission of this manuscript, and received funding to continue that work. Therefore, we are comfortable proposing that plan but are less comfortable including conversation/discussion about our less-developed ideas.

“Additional changes.”

Minor changes throughout to support major revisions requested by the reviewers. All of the minor changes are highlighted in blue font, to the best of our ability.

The title was changed to: “Bacteria Invade the Brain Following Intracortical Microelectrode Implantation, Inducing Gut-Brain Axis Disruption and Contributing to Reduced Microelectrode Performance”

Added authors who participated in histological experiments required to respond to the reviews.

Jennifer Coleman¹, Longshun Li^{1,2}, and Andrew J. Shoffstall^{1,2}

A point-by-point response to reviews:

We thank the reviewers for their additional time and feedback. We appreciate that the reviews were positive regarding our initial revisions. As in the last round, our additional analysis performed because of the questions and suggested revisions greatly improved the resolution of the data and our ability to support our conclusions – thank you!

Below, the reviewer's comments are shown in **BOLD** font. Our response to the comment is shown in “normal font.” Changes to the manuscript are noted both below and in the manuscript with **RED FONT**. Text deleted from the manuscript has not been indicated. Minor grammatical changes made during our additional reads have also not been noted here. Edits that were denoted in blue from the first revision have been converted back to black text.

Please note that we did our best to indicate Line numbers for changes, but it was a moving target as lines shifted above and below due to accepting track changed additions and deletions. The Line numbers indicated should be within a few lines, if not correct.

In addition to reviewer-specific responses, we also edited the Methods portion of the manuscript to ensure that the Nature Communications Reporting Summary is satisfied.

“All studies must disclose on these points even when the disclosure is negative”

Replication

Methods, Lines 1149-1155:

“Replication

Our hypothesis was to test whether changing the gut microbiome can impact neural recording performance and brain health after implantation. We performed conceptual replication by testing this hypothesis in four different experiments: 16S evaluation of the brain after implantation and treatment, neural recording performance, proteomic evaluation around the implant site, and transcriptomic evaluation around the implant site. There was no direct replication or systematic replication performed in this study. However, future studies will be utilized to perform both direct and systematic replication of these results.”

Randomization

Methods, Lines 1083-1084: “Since not all implanted brains were utilized for proteomics, a random selection of 3-4 brains were taken from each group.”

Blinding

Methods, Lines 886-887: “The data collector was blinded to the animal’s group to eliminate any inherent bias in recording data collection. Similarly, all recordings were analyzed blindly to remove bias.”

Methods, Lines 950-951: “The researcher performing perfusion was blinded to the animal’s group to eliminate any inherent bias in sample collection.”

Reviewer #1 (Remarks to the Author):

The authors have significantly modified their manuscript, So I've taken my analysis from the start. However, you will first find below a response to the authors for two remaining concerns.

“3) Page 2: “Degradation of blood-brain barrier (BBB) integrity is an appreciable consequence of microelectrode-mediated neuroinflammation and increases the entry of blood-borne components into the brain, where they could amplify and extend the neuroinflammatory response^{3,4,6,15-18}”. None of this reference are related to microelectrode, and poorly to neuroinflammation.

We disagree with the reviewer’s interpretation of the literature citations we used. Below, we show the full reference and call out specific examples of text from each reference, which demonstrate why it is an appropriate citation for the text.

No changes to the manuscript were made in response to this reviewer comment.”

I agree that my comment was not clear. However, the sentence is still strongly misleading. References

did not support that microelectrodes increase the entry of blood-borne components into the brain, where they could amplify and extend the neuroinflammatory response.

Ref 3 is dedicated to microelectrode, failure is used for “recording failure”: “Most failures (56%) occurred within a year of implantation, with acute mechanical failures the most common class (48%), largely because of connector issues (83%). Among grossly observable biological failures (24%), a progressive meningeal reaction that separated the array from the parenchyma was most prevalent (14.5%).”

Ref 4 is a review that only acknowledge the role of microelectrode in the maintenance of inflammation with maybe an impact on the BBB.

Same for Ref 6, same for ref 15 (DAMPs are not associated to bacteria in this paper), same for reference 16-18

We thank the reviewer for clarifying their concerns regarding our introductory claim. As there is no current literature detailing introduction of bacteria through the BBB after microelectrode implantation, we felt that establishing the invasion of other blood-borne components and their effect on inflammation was vital to our rationale. In this sense, many studies, including those referenced, talk about the neuroinflammatory response triggered by invading macrophages and DAMPs.

It was our goal then to suggest a connection and use of logic. If the door is opened, and leaves blow into the house, it is not unreasonable to expect that **if** a bug were near the open door when the leaf blew in, that the bug may be able to enter the house at the same time the leaves entered.

Rationale and data to support this premise are first established by showing that the BBB is opened, and blood components (macrophages and serum proteins / leaves) enter the brain (house).

We revised our statement to be more direct in our suggested hypothesis, and removed several of the reference, leaving only three that in fact show blood derived products in the brain tissue following microelectrode implantation.

- 1) Ravikumar et al, Biomaterials 2014
- 2) Saxena et al, Biomaterials 2013
- 3) Bennett et al Biomaterials 2018

Line 59-62: “Degradation of blood-brain barrier (BBB) integrity is an appreciable consequence of microelectrode-mediated neuroinflammation and increases the entry of blood-borne components into the brain^{6,15,16}. **Therefore, it is not inconceivable that other factors circulating in the blood at the time of, or after, microelectrode implantation could also enter the brain through the permeable domain.**”

“Comment 8) Page 5 “It’s still not clear to me whether the authors took into account, in their analysis, the fact that this antibiotic cocktail can cause dysbiosis resulting in neuroinflammation.

We thank the reviewer for highlighting an important point.

In the original manuscript, we did not compare the neuroinflammatory profiles of unimplanted mice treated with antibiotics to untreated naïve control mice. However, we did consider the effect of dysbiosis on acute and chronic changes in the neuroinflammatory response to implanted microelectrodes.

At the acute time point, treatment with antibiotics reduced the neuroinflammatory response (**Figure 4**) – thus confirming that the antibiotics could not be simultaneously decreasing and increasing neuroinflammation. At the chronic timepoint the effect wore off, but did not reverse (**Figure S3**). The diminishing of an effect does not suggest a new mechanism in which the antibiotic is inducing dysbiosis leading to increased inflammation, but more likely the diminished effect of the original mechanism over time, as the biology compensates. While important to consider, the lack of evidence to the effect and evidence against the effect do not suggest the hypothesis to be examined fully enough for inclusion in the manuscript.

No change to the manuscript.

“Comment # of reviewer 2”

While I am convinced now that there were no blood contamination in the collected brain, I am not convinced by Reviewer Response Figure 2 and 3. There is so many bacteria detected in the sham control. If there were so many bacteria, they would undoubtedly be detected on agar and the 16S would be very clear in the event count.

We thank the reviewer for raising this concern. As mentioned in our reviewer response for Figure 2 and 3, we had no way of ruling out contamination for the tissue during processing. Although this tissue was kept frozen in a -80° C freezer, the staining process done here could have introduced bacteria to the tissue. However, the goal of this response was not to determine the origin of bacteria or measure contamination. The goal was to observe whether bacteria measured are coming from blood vessels and to show that our perfusion was sufficient in clearing out any blood. We believe that the outlined results demonstrated clear perfusion and that the bacteria measured in the rest of the study were not from blood vessels.

As to the first half of the last sentence, “If there were so many bacteria they would undoubtedly be detected on agar...” we state in the 13th paragraph of the Discussion (**Line 694**), that work of this nature still needs to be done: “The confirmation of live bacteria and the presence of a microbiome in the brain after implantation necessitates further work, including comprehensive live bacteria culture of implanted brain tissue **and appropriate controls for excluding the contribution of background noise.**”

While a positive agar plate test would be wonderful, a negative culture would not convincingly indicate a lack of bacteria. Although we are better at culturing human-associated microbes than environmental, specifically those of the gastrointestinal tract, sequencing studies suggest many remain uncultured, possibly due to the difficulty of culturing obligate anaerobes (Dominant and diet-responsive groups of bacteria within the human colonic microbiota, Walker et al 2011) (A human gut microbial gene catalogue established by metagenomic sequencing, Qin et al 2010). Uncultured microbes exist in other bodily environments, such as blood or tumors.

Additionally, “A high proportion of the most abundant 16S rRNA phylotypes that we detected here corresponded to cultured bacteria, with 66% of the 50 phylotypes that accounted for 40.5% of sequences having close cultured relatives. This suggests that the limited coverage of the human gut microbiota through cultivation may be because of insufficient anaerobic isolation work, rather than to intrinsic non-culturability of human colonic bacteria.” “Overall, however, 33.4% of phylotypes showed <98% identity with cultured bacteria in this study, reflecting the relatively poor coverage of the less abundant bacterial groups by cultured strains.” (Dominant and diet-responsive groups of bacteria within the human colonic microbiota, Walker et al 2011)

“Only 31.0–48.8% of the reads from the two previous studies and the present study could be aligned to 194 public human gut bacterial genomes (Supplementary Table 5), and 7.6–21.2% to the bacterial genomes deposited in GenBank (Fig. 1). This indicates that the reference gene set obtained by sequencing genomes of isolated bacterial strains is still of a limited scale.” (A human gut microbial gene catalogue established by metagenomic sequencing, Qin et al 2010).

As for the second half of the last sentence, “...and the 16S would be very clear in the event count,” Figure 1-C and the raw read counts to the Methods Section, second paragraph of the 16S Bacterial DNA Sequencing section, were added as part of this reviewer’s request for the raw read counts during the last review cycle, although the response clarified that 16S read counts alone are not a valid indicator of microbial loads and should not be used to do so.

In addition, for this experiment, total DNA was extracted from brain biopsy punches rather than specifically isolating bacterial DNA, meaning an appreciable portion of the DNA extracted was from mouse cells. Although the primers in this protocol are designed to specifically amplify bacterial 16S rRNA, a large proportion of the reads from brain samples were the result from non-specific, off-target amplicons from mouse gDNA. As such, we believe strong conclusions regarding bacterial biomass should not be drawn from these read counts.

Review, September 2024

The authors have greatly modified their manuscripts and clarified many points. The results obtained on inflammation seem solid to me. However, I am not at all convinced by the hypothesis of a microbiota in the naive brain. I have noted in this work that the authors have also called this hypothesis into question or presented data that call it into question.

We thank the reviewer for bringing up this concern. Our intentions here were to avoid claiming that the naïve brain has a native microbiome. We have addressed the following concerns and strengthened our language (and edited labels on figures) to ensure that the reader does not believe we are arguing for a native brain microbiome, as that was out of the scope of this study and an aim of future work. These edits are numerous and can be found throughout the entire manuscript.

More detailed information regarding our edits are provided below in response to more specific comments/questions. But we would also like to point out that we have edited Figures 1 and 2 and the resulting captions to remove “naïve brains” and replace with “background.”

Line 169 (now Line 180): “These ratios suggest that microbial biomass is low or non-existent in the unimplanted brain.”

Line 651 (now 692): “16S measurement does not confirm the presence of live bacteria and may indicate either dead or fragments of 651 bacterial DNA.”

The impact of antibiotics on the results of 16S sequencing in the brain suggests that these results are background noise.

In my opinion, it is important, in order to avoid any misinterpretation by the community, to clearly state that in the current state of the results, the 16S results obtained in the brain before implantation or in the presence of antibiotics cannot be differentiated from the background noise. I suggest that authors rewrite the manuscript taking into account this general comment. With this in mind, here are my comments on this point (This list is not exhaustive):

Thank you for your clear direction. We feel that “rewrite” was meant to read “edit” and have thus significantly “edited” the manuscript to more carefully articulate the main findings and not leave room for misinterpretation of background data that was not extensively analyzed.

Our primary revision regarding these points are as follows:

- **Line 180-181:** Now reads “These ratios suggest that microbial biomass is low or non-existent in the unimplanted brain and **is considered background for the sake of this analysis.**”
- **Lines 195-203:**
Antibiotic Treatment Reduces Abundance of Distinct, Implant-Associated Bacterial Features
Having removed contaminants and **finding evidence of host-derived bacterial DNA** in the implanted brain **samples**, our first step was to identify and compare the composition of **bacterial sequences** present in both fecal matter and **at background** in the **samples from unimplanted mice to those seen in samples from** microelectrode-implanted mice. **Given how little bacterial DNA was extracted in the naïve unimplanted brain samples, these samples from both control and antibiotic-treated mice will be referred to as background for their respective cohorts, so as not to imply that these results provide evidence of a native brain microbiome.** Analyses were conducted on read counts for both unique amplicon sequence variants (ASVs) and operational taxonomic units (OTUs) of similar sequences.
- **Lines 242-250:** “**Although** antibiotic-treated brains were processed and sequenced in parallel with the corresponding control cohort, **this study did not experimentally validate the origin of these bacterial sequences. Therefore, we cannot rule out that these differences are not due to differential contamination of the brain sample with host-derived bacterial DNA from other body sites, rather than indicative of the presence of live bacteria in the brain. If implantation disrupts the microbiome at other body sites, the contamination of samples from implanted brains would differ from the contamination in background samples. Similarly, systemic antibiotic treatment may have altered the microbiome in those mice, potentially contributing distinct contaminants. Any such differences in contamination would likely be reflected in the results, underscoring the need for further experimental validation to clarify the true origin of the bacterial sequences.**”
- **Lines 264-287:** “unimplanted brain” to “**background samples**”; “microbes” to “**sequences**”; “invading the brain...” to “**the distinct, implantation-associated features...**”

- **Added Lines 302-304:** “However, we again cannot rule out contamination of the brain biopsy samples by bacterial DNA from other body sites, so these results could represent disruptions to the microbiome which do not occur in the brain.”
- **Lines 601-604:** Removed “~~Although the idea of bacteria residing in the brain is contentious due to the limited studies in this area, we believe our rigorous approach effectively mitigates the risk of contamination, supporting the presence of microbes in situ within the brain tissue.~~”
- **Line 692-694:** “...16S measurement does not confirm the presence of live bacteria and may indicate either dead or fragments of bacterial DNA.” Is now immediately followed by, “It is possible that the sequences detected are from fragmented DNA inside macrophages or other immune cells that ingested bacteria and then invaded the brain after implantation.”

Line 185 (now Line 197). You cannot compare the “composition of microbes present (...) in the brain tissue of naive unimplanted mice”. This suggest that the composition of a brain microbiota was described in this manuscript. I suggest to rephrase the sentence. You can compare results of 16S sequencing but not microbes.

We agree that this framing of the results would be more appropriate at this stage. This line (now 196-201) was changed to read, “Having removed contaminants and finding evidence of host-derived bacterial DNA in the implanted brain samples, our first step was to identify and compare the composition of bacterial sequences present in both fecal matter and at background in the samples from unimplanted mice to those seen in samples from microelectrode-implanted mice. Given how little bacterial DNA was extracted in the naïve unimplanted brain samples, these samples from both control and antibiotic-treated mice will be referred to as background for their respective cohorts, so as not to imply that these results provide evidence of a native brain microbiome.”

We have also updated the text in the below places to reflect this change. Any additions made in response to reviewer comments incorporated this phrasing as well.

- **Line 33:** removed “in health”
- **Lines 34-35, 89-90, 103, 120, 226:** “bacteria” to “bacterial sequences”
- **Line 38:** “invading microbes” to “implantation-associated features”
- **Line 42:** “invading bacteria” to “implantation-associated bacterial sequences...”
- **Line 91:** “naïve brain” to “background”
- **Lines 91-92, 206:** “microbes” to “bacterial feature(s)”
- **Line 97-99:** “~~Although the composition of bacterial sequences varied by timepoint and treatment group relative to background in DNA extracted from implanted brain tissue, it is important to indicate that these results do not confirm the presence of live bacteria in the brain.~~”
- **Line 102:** “While the source of bacterial sequences not resident in the gut or seen at background was not identified in this study, future studies could...”
- **Line 116-118:** “We compared within-sample diversity, between-sample diversity, and differential abundance in the implanted brain tissues to samples from the naïve, in-tact brain as a background condition to characterize any changes in sequence composition following BBB disruption.”
- **Line 119-120:** “To determine the impact, if any, of depleting the fecal (gut) microbiota on bacterial sequences extracted from the brain,...”
- **Lines 126-127, 150:** “microbes” to “bacterial DNA”
- **Line 152:** “microbial biomass” to “microbial DNA”
- **Lines 168, 173, 181, 195:** “bacteria” to “bacterial DNA”
- **Line 179-180:** “total bacterial loads in the brain” to “proportion of host-derived bacterial DNA”
- **Line 195:** “Invasive Microbe Abundance” to “Distinct, implant-associated bacterial features”
- **Line 206:** “microbes” to “bacterial sequences”
- **Line 208, 210:** “unimplanted brain” to “background”
- **Line 212:** “non-invasive” bacteria to “common background features”
- **Line 213:** “unimplanted brains” to “background samples”
- **Line 215-216:** “unique invasive” to “distinct, implantation-associated”
- **Line 217:** “invading” to “distinct”
- **Line 218:** “genera found in the unimplanted brains...” to “common background features...”
- **Line 219:** “...of the original genera found in the unimplanted brains...” to “background genus observed...”

- **Line 220:** “invasive distinct, implantation-associated genera” to “**distinct, implantation-associated genera**”
- **Line 222-223:** “bacterial invasion” to “**distinct, implantation-associated features**”
- **Line 223-224:** “non-invasive” to “**common features**”
- **Lines 224, 228-229, 230-231, 232, 234-235, 235-236, 238-239, 255, 266, 330-331, 639, 759-760:** “invasive genera” / “invasive microbial” / “invasive bacteria” to “**distinct, implantation-associated features**”
- **Line 284-287:** “unimplanted control brains” to “**background samples**”
- **Line 300:** removed “naïve unimplanted brains...”

line 197. You cannot conclude that “45 genera are detected in the unimplanted brain”

- **Now Line 213:** “brain tissue from unimplanted mice” changed to “**background samples**”

Concerning the paragraph line 273/280 (now 299-300) and the sentence: “we would not expect them to vary in response to antibiotic-treatment of the animal”. I disagree. If there is any contamination from bacteria living in mice, this contamination should be different after antibiotic treatment, because the whole bacteria population is different. Therefore, the 16S amplicon background arising from this “whole microbiota” will differ between treated and non-treated mice. This argument that 16S results in naive brains are not artefactual doesn't stack up in my opinion and should be removed. Moreover, line 278, you cannot mention the possibility of a naïve brain microbiota if we consider your conclusion line 169.

Thank you for the suggestion. We have adjusted the text to reflect the meaning we were trying to convey. When we bring up contamination, we are referring to 16S present in the brain samples as a result of processing in the lab and extraction of DNA. As a result, these bacteria should not be changing due to our antibiotic treatment, as they would be influenced after sacrifice and extraction.

Removed “~~The implanted brains were processed in parallel and sequenced together, so the variation in alpha diversity between the two time points in implanted brains cannot be explained by a batch effect.~~”
From what was lines 287-289.

- **Line 556/558, thank you for specifying that this concerns the post-implantation brain.**

You're welcome.

- **Line 584 (now Line 625): “in microbiota composition”. Do you mean microbiota in the gut ? If yes please add "gut" before microbiota.**

Yes, we added the word “gut” – now reads: “It is important to recognize that even subtle changes in **gut** microbiota composition have been linked to changes in brain health.”

- **Finally, could you exclude the possibility that immune cells that have collected bacteria debris may be present in the brain or its surrounding? As for the trojan horse hypothesis proposed for listeria monocytogenes?**

We thank the reviewer for bringing this point forward. We added the following to the discussion to address it: **Line 693-694: “It is possible that the sequences detected are from fragmented DNA inside macrophages or other immune cells that ingested bacteria and then invaded the brain after implantation.”**

Other concerns:

- **Ampicillin, Clindamycin, and Streptomycin induce a modification of the gut microbiota and this may have an impact on brain inflammation. It might be useful to clarify this.**

Please see the above response regarding this same topic.

- **I do not understand the link between line 604 and 605.**

Line 645-647: The sentence has been revised to now read: “However, it is important to note that bacteria **that may reside in the brain have been poorly characterized, leading to the possibility that there are bacteria in the brain after implantation that are unable to be matched to current databases.”**

The intent of this statement is to say that some of the bacteria could be resident from the brain, but we can't detect that accurately yet to draw a conclusion one way or the other.

Minor concerns:

- Line 31 and 68, what are « constituents » ? Or “components”, line 58. Do you mean PAMPs ?

Both terms are synonyms for “parts” or “things that reside within” and are aligned with the intended meaning.

- Line 63/64: “Bacteria entry through the damaged BBB is likely the dominant entry point for bacteria residing in or on the host”. Why not clearly state that bacteria pass through the bloodstream? What do you mean by dominant? What other routes do the authors consider that do not involve passage through the bloodstream?

Line 66-67: This statement was revised to now read: “Bacteria could enter from the bloodstream through the damaged BBB.”

- Line 108: “in-tact”. Probably “intact” ?

Line 117: Corrected.

- Figure 1C, it seems that the y axis is wrong, 1000 should have been 100

Corrected

Reviewer #2 (Remarks to the Author):

The authors have done an admirable job addressing my initial concerns, and the manuscript as been significantly improved. Now that the full differential gene expression data has been included, I note that the most obvious differential gene expression in the 12-month antibiotic-treated mice is a global upregulation of olfactory receptors. This is curious for multiple reasons, and it would be appropriate for the authors to comment on this (as anyone who looks at the DEG data will be similarly curious).

We thank the reviewer for their encouragement to include description of the olfactory receptors in the transcriptomic data. We have added the following paragraph to the Results section titled “Antibiotic Treatment Impacts the Neuroinflammatory Response to Intracortical Microelectrodes”

Lines 543-551: “Lastly, at 12-weeks post-implantation, the most widely impacted pathway is olfactory transduction, with 210 DE genes out of 482 (KEGG: 04740, p-value < 0.0001). On top of their classical ability to provide odorant detection, recent literature has implicated olfactory receptors in inflammation, including presence on and activation of macrophages and monocytes⁵¹. While the connections between olfactory receptors and inflammation are new and less studied, there is evidence suggesting upregulation of several genes during inflammation, including Olfr1014 and Olfr65, both of which were differentially expressed here⁵². Olfactory transduction at the 4-week time point is impacted much less, having only four DE genes out of the 482. Olfactory receptors may prove an interesting area for future studies to explore inflammation in the brain.”

Lines 731-734: “Furthermore, the olfactory transduction pathway was heavily impacted at 12 weeks point-implantation (210/482 DE genes) compared to 4 weeks post-implantation (4/482 DE genes), which may indicate further investigation into how olfactory receptors can influence the immune response in the brain.”

Reviewer #3 (Remarks to the Author):

Great work with pretty impressive data.

Pretty difficult to intervene at this stage of the review process, but reviewer 1 is 100% right – there is no data in this manuscript that firmly show bacteria in the naive brain. The post-surgery increase is, to me, what really matter and what this manuscript is about. So I suggest that the authors make 100% clear that the presence of brain-associated bacteria remains unknown and challenging to study.

Thank you! We did not intend to make this claim and have revised the entire manuscript to more carefully reflect that we are not claiming there is a resident brain microbiome, and that bacteria detected in unimplanted brains were considered background. Specific details of these edits can be found above in response to reviewer 1.

Notably, **Lines** such as ~~“Although the idea of bacteria residing in the brain is contentious due to the limited studies in this area, we believe our rigorous approach effectively mitigates the risk of contamination, supporting the presence of microbes in situ within the brain tissue.”~~ have been **removed** from the manuscript.

*Note that additional edits were made throughout the revised manuscript to remain consistent with the reviewers' feedback regarding inconsistencies with our description of the brain resident microbiome. These edits were always denoted with **RED** font. For example, please see **Lines 299-330, and figure captions for Figure 1 and Figure 2.**

In addition, the first line of the conclusion was rewritten to now read: “The current study **reports the detection of bacterial DNA sequences from** normally gut-resident microbes and microbes of a currently unknown origin **in DNA extracted from brain tissues** after intracortical microelectrode implantation **which were not detected in background samples.**”